# Linking uncertainty in simulated Arctic ozone loss to uncertainties in modelled tropical stratospheric water vapour

Laura Thölix[1], Alexey Karpechko[2], Leif Backman[1], and Rigel Kivi[3]

[1]Climate Research, Finnish Meteorological Institute, Helsinki, Finland
[2]Arctic Research, Finnish Meteorological Institute, Helsinki, Finland
[3]Arctic Research, Finnish Meteorological Institute, Sodankylä, Finland

*Correspondence to:* L. Thölix (laura.tholix@fmi.fi)

**Abstract.** Stratospheric water vapour influences the chemical ozone loss in the polar stratosphere via controlling the polar stratospheric cloud formation. The amount of water vapour entering the stratosphere through the tropical tropopause differs substantially between simulations from chemistry–climate models (CCM). This is because the present-day models, e.g. CCMs, have difficulties in capturing the whole complexity of processes that control the water transport across the tropopause. As a result there are large differences in the stratospheric water vapour between the models.

In this study we investigate the sensitivity of simulated Arctic ozone loss to the simulated amount of water vapour that enters the stratosphere through the tropical tropopause. We used a chemical transport model, FinROSE-CTM, forced by ERA-Interim meteorology. The water vapour concentration in the tropical tropopause was varied between 0.5 and 1.6 times the concentration in ERA-Interim, which is similar to the range seen in chemistry climate models. The water vapour changes in the tropical tropopause led to about $1.5\,\mathrm{ppmv}$ less and $2\,\mathrm{ppmv}$ more water vapour in the Arctic polar vortex compared to the ERA-Interim, respectively. The change induced in the water vapour concentration in the tropical tropopause region was seen as a nearly one-to-one change in the Arctic polar vortex.

We found that the impact of water vapour changes on ozone loss in the Arctic polar vortex depends on the meteorological conditions. The strongest effect was in intermediately cold stratospheric winters, such as 2013/14 winter, when added water vapour resulted in 2–7 % more ozone loss due to the additional formation of polar stratospheric clouds (PSC) and associated chlorine activation on their surface, leading to ozone loss. The effect was less pronounced in cold winters such as the 2010/11 winter because cold conditions persisted long enough for a nearly complete chlorine activation even in simulations with prescribed stratospheric water vapour amount corresponding to the observed values. In this case addition of water vapour to the stratosphere led to increased areas of ICE PSC but it did not increase the chlorine activation and ozone destruction significantly. In the warm winter 2012/13 the impact of water vapour concentration on ozone loss was small, because the ozone loss was mainly $NO_x$ induced. The results show that the simulated water vapour concentration in the tropical tropopause has a significant impact on the Arctic ozone loss and therefore needs to be well simulated in order to improve future projections of the recovery of the ozone layer.

# 1 Introduction

Water vapour in the stratosphere is a minor constituent with typical mixing ratios of 3–6 ppmv (e.g., Randel et al., 2004). It plays, however, an important role in radiative and chemical processes, especially in the upper troposphere/lower stratosphere (UTLS) where changes in the water vapour concentration result in significant changes in radiative forcing of the troposphere (Riese et al., 2012). Dessler et al. (2013) suggested that, as a result of global warming, tropopause temperature will increase leading to an increase in stratospheric water vapour (SWV). Since water vapour is a greenhouse gas, future increases in SWV will provide a positive feedback further warming the troposphere below. However, tropospheric warming may also lead to a significant cooling near the tropopause in connection with deep convection (Kim et al., 2018), so that the link between warming climate and tropopause temperature is not straightforward. Photodissociation of water vapour is an important source of odd hydrogen $HO_x$ ($H+OH+HO_2$). Catalytic cycles involving $HO_x$ contribute to chemical ozone loss in the stratosphere (Dvortsov and Solomon, 2001). Water vapour contributes to the formation of stratospheric aerosols including polar stratospheric clouds (PSCs), i.e. liquid and solid particles in combination with $H_2SO_4$ and $HNO_3$, or ice particles (e.g., Hamill et al., 1997).

Heterogeneous reactions in or on PSC particles can lead to massive ozone depletion inside the polar vortices when atmospheric concentration of halogens is sufficiently high (Solomon et al., 1986; Wohltmann et al., 2013). Since the formation of PSCs requires very low temperatures (below about 195 K) that are rarely reached in the Arctic, significant polar ozone depletion takes place there only occasionally (Rex et al., 2006; Manney et al., 2011; Müller et al., 2008; Chipperfield et al., 2015), while it has been a yearly phenomenon in the Antarctic since about the mid 1980s (e.g., Dameris et al., 2014). The stratospheric abundance of chlorine will remain elevated for decades, and polar ozone losses will therefore be seen also in the future. Equivalent effective stratospheric chlorine (EESC) is a proxy that is frequently used to describe the combined effect of chlorine and bromine on stratospheric ozone. The level of EESC in 1980 is commonly taken as a level that needs to be reached to achieve a recovery of stratospheric ozone. According to a recent study by Engel et al. (2018) the EESC would return to the 1980 level in 2077 for a mean age of 5.5 years, which is representative for polar winter conditions. Several studies have discussed Antarctic ozone depletion and its recovery (see e.g. Eyring et al. (2010); Dameris et al. (2014); Solomon et al. (2016); Chipperfield et al. (2017); Kuttippurath and Nair (2017); Strahan and Douglass (2018)). Kuttippurath and Nair (2017) showed based on ozone balloon soundings and total ozone data from satellite instruments that Antarctic ozone has begun to recover. Based on data from the Aura Microwave Limb Sounder (MLS) of HCl and ozone Strahan and Douglass (2018) could show a decline in lower stratosphere chlorine and a corresponding decline in ozone depletion for the period 2013–2016 compared to the period 2004–2007. However, a recovery of ozone to 1980 ozone levels is projected not to occur until around 2025–2043 in the Arctic and 2055–2066 in the Antarctic (Dhomse et al., 2018). Both colder air and increased SWV can increase the formation of PSCs, which could release more active chlorine and cause severe ozone depletion although future chlorine loadings will be smaller. All these suggest that SWV is a critical factor affecting ozone chemistry.

The majority of the previous studies addressing impacts of SWV on Arctic ozone depletion considered the effects of observed and projected increases in SWV concentrations (Eyring et al., 2007; Kirk-Davidoff et al., 1999; Dvortsov and Solomon, 2001; MacKenzie and Harwood, 2004; Stenke and Grewe, 2005; Feck et al., 2008; Vogel et al., 2011). For example Vogel et al. (2011)

used a chemistry–transport model (CTM) and studied the effect of increased SWV on Arctic ozone loss for meteorological conditions from the cold Arctic winter 2004/05. They found that increasing SWV by 0.58 ppmv, which is a typical amount simulated by chemistry climate models (CCMs) by the mid-21 century (Eyring et al., 2007), would lead to an additional 6 DU of ozone loss under cold winter conditions. Sinnhuber et al. (2011) used a CTM driven by meteorological conditions for the cold Arctic winter 2010/11 and assumed a uniform increase of SWV of 1 ppmv. For such conditions they reported a 25 DU increase in ozone loss, i.e. about 20 % of their simulated total ozone loss for that winter.

Smalley et al. (2017) studied future trends in the tropical lower stratospheric water vapour and provided a regression model for analysing the factors driving the trends and variability in the 21st-century. They found that warming of the troposphere causes a long term increasing trend in the water vapour entering the stratosphere, which can be partially offset by an increase of the Brewer–Dobson circulation with accompanied cooling of the tropical tropopause. MacKenzie and Harwood (2004) studied the effect of increasing SWV due to future increase in tropospheric methane on ozone. They simulated the year 2060 under the Intergovernmental Panel on Climate Change Special Report on Emission Scenarios (SRES) B2 scenario, where $CH_4$ lies approximately midway between the extremes of the SRES scenarios. They found an increase in the occurrence of PSCs, with about 20 to 25 % due to increase in the water vapour. The rest is from radiative cooling of the middle atmosphere due to changes in the concentration of several trace gases. In the simulations by MacKenzie and Harwood (2004) the increased SWV due to projected methane increases caused a 15 % (about 0.5 ppmv at 465 K level) deeper Arctic ozone loss in 2060. However, cooling of the stratosphere could at least partially offset the effect of the increased PSCs by slowing down the second-order reactions in ozone loss cycles (Rosenfield et al., 2002; Revell et al., 2012). This effect is mainly seen in $NO_x$ and $HO_x$ induced loss outside the polar vortex, while the effect on PSCs from temperature is seen within the vortex.

Revell et al. (2016) also studied the effect of future methane changes on SWV under different RCP-scenarios. The contribution of methane to the SWV was found to be highly dependent on the projected methane concentration, altitude and latitude. Under RCP 6.0 between 1960 and 2100 the SWV was projected to increase by approximately 1 ppmv throughout most of the stratosphere, excluding the Antarctic lower stratosphere. The largest increase was seen following the RCP 8.5, with 60 % additional water vapour in the extratropical upper stratosphere, and ca. 35 % in the Arctic lower stratosphere. The largest contribution from methane to the SWV change was about 50 % under RCP 8.5, which assumes a rather extreme methane increase scenario, and the smallest was about 4 % under RCP 2.6.

Recently Sagi et al. (2017) studied Arctic ozone losses between years 2002 and 2013 using data assimilation of Odin/Sub-Millimetre Radiometer (SMR) observations. They found that the largest ozone losses were caused either by halogens or by the $NO_x$-family, and the dominating process for ozone destruction is determined mostly by the temperatures inside the polar vortex. The very stable and cold polar vortex in the Arctic winter 2010/11 led to remarkable halogen driven ozone loss with 2.1 ppmv ozone destroyed at the 450 K level. In the winter 2012/13 the polar vortex was more unstable and a vortex split occurred in early January due to a sudden stratospheric warming (SSW), and $NO_x$ rich air from the mesosphere descended to the upper stratosphere and led to ozone loss there. Thus the effect on Arctic ozone depletion from changes in SWV depend on the meteorological conditions, and the dynamical stability in a given winter.

The main source of SWV is the upward transport from the troposphere through the tropical tropopause in the upwelling branch of the Brewer–Dobson circulation. The concentration of SWV is controlled by the coldest temperature met by the ascending air parcels (i.e. cold point temperature). Gettleman et al. (2010) analysed 16 state-of-the-art CCMs and demonstrated large discrepancies between simulated SWV in these models, which were closely related to the simulated cold point temperatures. The "entry" value of SWV in these models ranged between 2 and 6 ppmv, compared to the observed value of 3–4 ppmv. These intermodel differences by far exceed the magnitude of the projected water vapour increases in the 21 century used so far in the studies of ozone loss sensitivities to SWV. One may wonder what the implications of these discrepancies for stratospheric ozone losses simulated by CCMs are. This question is difficult to address by analyzing CCM outputs because there are other differences between the models which affect simulated ozone losses, such as differences in simulated transport. Therefore a more controlled experiment is needed in order to assess impact of these SWV changes on ozone loss.

In this study we address the question of what the implications of the differences in simulated tropical stratospheric water vapour between chemistry–climate models are for the simulated Arctic ozone loss. Similar to Vogel et al. (2011) and Sinnhuber et al. (2011) we address this question by performing CTM simulations using different SWV concentrations. The principal differences in our methodology from the previous studies are (1) the boundary conditions of perturbed water vapour experiments resulting in a different spatial pattern of SWV anomalies and (2) the magnitude of SWV perturbation, which is larger than in the Vogel et al. (2011) and Sinnhuber et al. (2011) studies, but in the range of Revell et al. (2016) and MacKenzie and Harwood (2004). We also analysed seven different winters, whose dynamical circumstances such as the evolution of the temperature and polar vortex were different (see Section 3 for more details).

## 2   Modelling and data

A global off-line chemistry–transport model for the middle-atmosphere, FinROSE-CTM, was used for simulating the effect of the SWV on Arctic ozone depletion. The FinROSE-CTM is described in detail in Damski et al. (2007b). In this study, the model has a horizontal resolution of $3° × 6°$ (latitude × longitude). It has 40 hybrid-sigma levels up to 0.1 hPa (about 65 km). The temperature, winds and surface pressure are from the European Centre for Medium-range Weather Forecasts (ECMWF) ERA-Interim reanalyses (Dee et al., 2011).

The model transport is computed using a flux-form semi-Lagrangian transport code (Lin and Rood, 1996). The chemistry scheme of the model comprises 36 species and includes about 150 reactions. In addition to gas-phase chemistry, the model includes a PSC scheme with liquid binary aerosols (LBA), supercooled ternary solution of sulfuric acid, nitric acid and water (STS, type Ib), solid nitric acid trihydrate (NAT, type Ia) and ice (ICE, type II) PSCs. The heterogeneous chemistry includes altogether 30 reactions and is based on the calculation of the composition and volume of sulphate aerosols and PSCs, as well as the partitioning of species between gas phase and condensed phase. The number density profile is prescribed for each PSC type (Damski et al., 2007b) and the sulphuric acid distribution is based on 2-D model data (Bekki and Pyle, 1992). Absorption cross-sections and rate coefficients follow the recommendations by Sander et al. (2011) and for some heterogeneous reactions the recommendations by Atkinson et al. (2007) are used, see details in Damski et al. (2007b). The reaction rates on NAT

and ICE PSCs are not directly affected by the water vapour concentration except through the available surface area, i.e. the uptake coefficients are constant. Water vapour and $HNO_3$ condenses onto the binary aerosol and STS droplets making them more dilute, which increases the uptake coefficients of some reactions, i.e the hydrolysis reactions of $ClONO_2$, $BrONO_2$ and $N_2O_5$, as well as the reaction of HCl between $ClONO_2$ and HOCl (Sander et al., 2011). The sedimentation of PSC particles, which can lead to denitrification and dehydration, is calculated based on the settling velocity, which takes into account the PSC particle size.

Look-up-tables of photodissociation coefficients were pre-calculated using the PHODIS model (Kylling et al., 1995). Within PHODIS the radiative transfer equation is solved by the discrete ordinate algorithm (Stamnes et al., 1988). This algorithm has been modified to account for the spherical shape of the atmosphere using the pseudo-spherical approximation (Dahlback et al., 1991).

The tropospheric concentrations of the chemical species are prescribed via model boundary conditions. The boundary conditions of water vapour and ozone are taken from the ECMWF ERA-Interim reanalysis (Dee et al., 2011) except for the water vapour boundary conditions in the sensitivity experiments which are described below. The concentration of tropospheric methane ($CH_4$) is from Global view-data (http://www.esrl.noaa.gov/gmd/ccgg/globalview/ch4), nitrous oxide ($N_2O$) concentration is from Agage data (Prinn et al., 2000), and halogens concentrations in the troposphere ($Cly$ and $Bry$) are from Montzka et al. (1999) updated data. The carbon dioxide ($CO_2$) concentration is based on global annual mean trend data (ftp://aftp.cmdl.noaa.gov/products/trends/co2). At the upper boundary ($0.1\,hPa$) climatological values of water vapour and ozone averaged over 2005–2013 from MLS data were used.

The FinROSE-CTM has previously been used to study the impact of meteorological conditions on water vapour trends (Thölix et al., 2016), ozone and $NO_x$ chemistry in the mesosphere (Salmi et al., 2011), Arctic polar ozone loss (Karpechko et al., 2013) and the impact of the driver data on the model transport (Thölix et al., 2010). Long term trends of Arctic and Antarctic ozone losses, past and future, have been investigated by using driving data from a chemistry–climate model (Damski et al., 2007a). The model results showed good agreement with satellite and ground based observations. The FinROSE water vapour was compared to observations of water vapour profiles from the Microwave Limb Sounder (MLS) and frost point hygrometer soundings from Sodankylä (Thölix et al., 2016). The extent of ICE PSCs simulated by FinROSE was compared to Cloud-Aerosol Lidar and Infrared Path finder Satellite Observation (CALIPSO) data in Thölix et al. (2016).The total ozone distribution was compared to data from Total Ozone Mapping Spectrometer (TOMS) and the Ozone Monitoring Instrument (OMI) satellite instruments in Damski et al. (2007a), Thölix et al. (2010) and Karpechko et al. (2013). Salmi et al. (2011) compared the $NO_x$ and ozone profiles in FinROSE to data from the tmospheric Chemistry Experiment Fourier Transform Spectrometer (ACE-FTS) instrument.

For this study, three simulations covering the Arctic winters between 2009/2010 and 2015/2016 were performed. The simulations differed from each other by the prescribed water vapour concentration in the tropical tropopause region (stratosphere between $21°\,S$–$21°\,N$, below $80\,hPa$), where it was prescribed as follows: (1) water vapour taken from ERA-Interim (Interim simulation), (2) increased water vapour (Max simulation), and (3) decreased water vapour (Min simulation). Specifically, the SWV lower boundary conditions for Min and Max simulations were obtained by multiplying values from ERA-Interim be-

tween tropopause and 80 hPa, and between 21° S–21° N by monthly coefficients ranging between 1.46–1.7 (Max) and between 0.5–0.63 (min), so that they approximately correspond to the driest and wettest CCM, as determined by SWV values at the tropical tropopause, across models analysed by Gettleman et al. (2010). This construction allows us to isolate the influence of the tropical water vapour on stratospheric chemistry while keeping all other factors fixed, and thus to estimate the contribution of processes controlling tropical water vapour entry values to Arctic ozone loss. Eight simulated years before 2009 are considered as spin-up and were not analysed. Ozone was initialised with ERA-Interim ozone in every year, in the beginning of December. The water vapour concentration was not adjusted and allowed to evolve freely through the whole period of integrations. Ozone and water vapour observations from MLS (Lambert et al., 2007) were used to validate the reference simulation. MLS data is shown as 5 day averages because of the small amount of data covering the polar vortex in some cases.

## 3 Results

Model simulations were made for seven winters (2009–2016), but only four of them are discussed here. The four selected Arctic winters, 2010/11, 2012/13, 2013/14 and 2015/16 differ from each other with respect to the stratospheric temperatures and polar vortex strength. They provide examples of different role of SWV in ozone loss in mild (2012/13) cold (2010/11, 2015/16) and intermediate (2013/14) stratospheric winter conditions.

### 3.1 Temperature and water vapour

The boundary condition at the tropical tropopause for the reference simulation was evaluated by comparing simulated water vapour concentrations with observed ones from MLS. The top panels in Fig. 1 show daily mean water vapour at 80 hPa averaged between 21° S and 21° N for two representative years 2013 and 2014. The temperature for the same region is shown in the lower panels. The cold point, where SWV boundary conditions were prescribed, is just below the 80 hPa level. The temperature shows the typical annual cycle with minimum temperature in northern hemisphere (NH) winter and maximum temperature in NH summer. The temperature in the TTL controls how much water vapour enters the stratosphere by freeze drying the upwelling air (e.g., Fueglistaler et al., 2005). As a result the maximum water vapour concentration occurs in the NH autumn and minimum in early NH spring. The effect of interannual variability and shorter term variations in the temperature on stratospheric water vapour can also be seen, e.g. the low temperature in early 2013 results in 0.5–1 ppmv less water vapour than during the same time in 2014.

The Interim simulation produces water vapour concentrations comparable to the amount seen by MLS (Fig. 1), which shows that the boundary condition is reasonable. However, Interim variability is ahead that of MLS by 3–4 weeks. The reason for the time lag between Interim and MLS is not clear although it could at least partly be associated with a too fast Brewer–Dobson circulation in ERA-Interim which is responsible for upward transport of the water vapour anomalies in the tropics (Schoeberl et al., 2012). The Max simulation has 2–3 ppmv more water vapour in the tropics than the Interim simulation, while the Min simulation is about 1.5 ppmv drier than the Interim simulation. These differences correspond to the ratio between Max/Interim

of approximately 1.55–1.6 and about 0.55–0.6 between Min/Interim, i.e. they are consistent with the prescribed boundary conditions.

We next describe the meteorological condition in the Arctic stratosphere during the analysed winters. Figure 2 shows the daily average temperature in the Arctic polar vortex in winters 2010/11, 2012/13, 2013/14 and 2015/16. The polar vortex
was identified using the modified potential vorticity (mPV) (Lait, 1994), with the 475K potential temperature as reference level. Here the polar vortex is defined as the area enclosed by the 36 PVU isoline separately for every model level. The 36 PVU contour approximately corresponds to the region of maximum PV gradient, i.e the polar vortex edge (Rex et al., 1999; Streibel et al., 2006). The winter 2010/11 represents a cold winter with vortex average temperatures below 200 K and minimum temperatures below 195 K, sufficient for the formation of NAT/STS PSCs, throughout most of the winter, from December to
the beginning of April with only a brief interruption by a warming in early January. Minimum temperatures in the vortex were record cold and below 190 K even in the end of March (Manney et al., 2011). The winter 2012/13 is an example of a warm Arctic stratospheric winter. Vortex average temperatures below 195 K were seen for only a few days in December in the ERA-Interim data, and the minimum temperature was below 195 K until mid January. A SSW occurred in early January followed by a weakening and a break up of the polar vortex in the lower stratosphere already in February. The winter 2013/14 was
intermediate with average temperatures inside the polar vortex being close to the long-term climatological mean through most of the winter, until late March when a final SSW occurred. There were only a few days in late December when the average temperature was below 195 K. The 2015/16 winter was as cold, or even colder, as the 2010/11 winter during December–February with minimum vortex average temperatures below 195 K. However, a minor SSW occurred in early February and the final warming came in early March, ending the cold period and reducing ozone depletion potential much earlier than in the
2010/11 winter.

Figure 3 shows the five day running mean concentration of water vapour at 55 hPa averaged over the Arctic polar vortex for winters 2010/11, 2012/13, 2013/14 and 2015/16. The gaps in the 2012/13 MLS curve are due to an undefined vortex (or a too small vortex with only few observations) after the SSW. The water vapour concentration in the Interim simulation is comparable to the MLS data. However, the variability in water vapour is smaller in FinROSE than in the MLS data. Typically,
there is a stronger increase in water vapour towards spring in the MLS observations compared to the FinRose simulations. This is most evident in winter 2013/14 when MLS concentrations increased by more than 1 ppmv between November and April while the simulated increase was only 0.3 ppmv. Although an increase by spring is expected due to downward transport of air with higher SWV concentration by the Brewer–Dobson-circulation the increase seen in MLS observations in 2014 is unusual. For example the observed increase in January 2013 after the SSW associated with downward transport of water-rich air from
above was about 0.3 ppmv and that increase was reasonably well reproduced by FinROSE. Note that the MLS observations within the polar vortex are sparse, which adds some noise to the MLS vortex average. Also note that FinROSE vortex-mean values are calculated using all data points inside the vortex even if MLS data are not available for each point. This approach increases the robustness of model estimates but at the same time complicates direct comparison with MLS. Interestingly, when looking at the 60–90° N average, which includes also air from outside the polar vortex, there is no similar spring increase
in MLS data as can be seen in Fig. 3, and the agreement between FinROSE and MLS improves (not shown). In all winters,

the Max simulation has about 2 ppmv more water vapour in the Arctic polar vortex than the Interim simulation, and the Min simulation is about 1.5 ppmv drier than the Interim simulation. This indicates that the simulated differences in the polar vortex water vapour are about the same as the differences in the boundary conditions for the tropical tropopause (Fig. 1), despite the average increase in SSW between the TTL and the polar vortex of about 1.5 ppmv in each run.

There are also several SWV decreases seen in Fig. 3 which are due to the formation of ICE PSCs and possibly also to dehydration due to sedimentation of ICE particles. The most pronounced one is in the winter 2015/16 when, during a very cold period (Fig. 2), the observed concentrations decreased from 5.2 ppmv to 4.7 ppmv and remained low until late February. A relatively small decrease of only about 0.2 ppmv was simulated in the Interim run. This decrease corresponds to the formation of ICE PSCs in the model (see Section 3.2 for discussion of PSC results) and therefore at least a part of the decrease could be explained by sedimentation. A much larger decrease of about 1 ppmv was seen in the Max simulation starting from late December, which is consistent with larger amounts of ICE PSCs simulated in this run. Another, much smaller, decrease of about 0.2 ppmv can be seen in the MLS observations during mid-January 2011 corresponding to a cold period. The decrease is almost undistinguishable in the Interim simulation, but is pronounced in the Max simulation, which is a result of a larger amount of ICE PSCs.

## 3.2 Polar stratospheric clouds

**Table 1.** Sum of the ICE, NAT and STS PSC areas ($10^6 km^2$*day) at 55 hPa.

|     | Year    | 2010/11 | 2012/13 | 2013/14 | 2015/16 |
|-----|---------|---------|---------|---------|---------|
| ICE | Interim | 24      | 3       | 9       | 100     |
|     | Min     | 1       | 0       | 0       | 23      |
|     | Max     | 78      | 33      | 47      | 183     |
| NAT | Interim | 680     | 280     | 590     | 760     |
|     | Min     | 490     | 210     | 400     | 630     |
|     | Max     | 730     | 310     | 670     | 810     |
| STS | Interim | 1830    | 946     | 2110    | 2030    |
|     | Min     | 1770    | 850     | 1900    | 1890    |
|     | Max     | 2010    | 990     | 2230    | 2110    |

Figure 4 and 5 show PSC type 2 (ICE) and PSC type 1 a (STS) and b (NAT) areas at 55 hPa (about 20 km) in the Arctic polar vortex for winters 2010/11, 2012/13, 2013/14 and 2015/16. The area was calculated by summing the areas of model grid boxes containing PSCs. There is no formation temperature for STS, but they are formed gradually as water and $HNO_3$ dissolves into binary aerosols with decreasing temperature. Here we have defined STS based on how much $HNO_3$ has dissolved into

the liquid phase, we set the limit at $0.6\,pptv$ liquid $HNO_3$ for any given gridpoint. The limit approximately corresponds to a formation threshold temperature of 202K for STS.

In the winter 2010/11 the polar vortex was stable and cold, but not extremely cold. The ICE PSC area (Fig. 4) in the Interim simulation was mostly moderate except for a period in late January with cold temperatures and large ICE PSC areas. The ICE PSCs lasted longer in the spring than in other winters. It is unusual that ICE PSC:s occur after January, but in 2011 ICE PSCs were seen through February even in the Interim simulation. In the Max simulation the ICE PSCs lasted until mid March. Also CALIPSO (Pitts et al., 2007, 2018; Spang et al., 2018) observed PSCs in the 2010/11 winter. The observed ICE PSC areas are comparable to the FinROSE modelled ICE PSC areas (Thölix et al., 2016). Also the duration of ICE clouds being present is comparable. However, Khosrawi et al. (2018) reported that the comparison between the PSC volume densities simulated with the EMAC model were several orders of magnitude smaller than the ones observed with MIPAS.

In the winter 2012/13 the polar vortex was very cold in December, and some ICE PSCs were simulated. However, after the SSW in early January no ICE PSCs were simulated, not even in the Max simulation. The 2013/14 winter was moderately cold with some ICE PSC occurrence in late January. The winter 2015/16 started as being very cold in December and January, and the ICE PSC area was large throughout January. The maximum ICE PSC areas in the Interim simulation were about $70\,\%$ larger compared to the other cold winter 2010/11. Dörnbrack et al. (2017) and Khosrawi et al. (2017) also reported unprecedented and widespread ICE PSC formation seen in CALIPSO observations in 2015/16. This was also the only winter with a significant ICE PSC area in the Min simulation, with a water vapour concentration of less than $4\,ppmv$.

The water vapour concentration has a strong effect on the ICE PSC formation: in the Max simulations the ICE PSC area increases significantly in all winters. For instance in 2010/11 the largest PSC area is more than twice as large in the Max as in the Interim simulation. In the warm winters (2012/13 and 2013/14) the relative increase in the ICE PSC area due to additional water vapour was even larger than in the cold winters (2010/11 and 2015/16). The amount of water vapour was an important factor for the extent of ICE PSC occurrence also in winter 2015/16, however, the relative increase between the Interim and Max simulation was smaller than in other studied winters, that were warmer. PSCs start to form about two weeks earlier in the Max simulation compared to the Interim simulation. In the Min simulation the stratosphere is too dry for ICE PSC formation in nearly all years.

Figure 5 shows the area of NAT and STS PSCs in the Arctic vortex. Both NAT and STS areas are always significantly larger than the ICE areas because type 1 PSCs form at warmer temperatures than ICE PSCs. Type 1 PSCs typically start to form in early November and ICE PSCs in mid to late December. The simulated peak values in the STS area range from 18 to 24 million $km^2$, NAT area range from 11 to 16 million $km^2$ while the ICE area peaks range from 1.5 to 5.5 million $km^2$ in the Interim simulation. As expected the type 1 PSCs occur later in the spring than ICE PSCs, e.g. in the winter 2010/11 both NAT and STS PSCs were simulated until late April, more than a month later than the ICE PSCs.

In the cold winter 2010/11 STS PSCs persisted for more than five months, and NAT almost five months, from December to mid April. An increase in moisture (Max simulation) had only a minor effect on the NAT and STS areas. In the Min simulation the maximum STS (and NAT) area was about 2 million $km^2$ smaller than in the Interim simulation, while the difference between the Max and Interim simulations was much smaller. In the 2012/13 winter both NAT and STS PSCs were simulated

only in the beginning of the winter and by early January all the NAT PSCs disappeared due to warm conditions. The maximum value of STS area in the 2012/13 winter was the largest among the simulated years, 24 million $km^2$. The maximum NAT area was also the largest, about 13 million $km^2$. The increase of water vapour in the Max simulation did not change the PSC area as much as the decrease of water vapour in the Min simulation. In the early 2012/13 winter the NAT and STS areas were even

larger than in 2010/11, but warmer temperatures in the vortex in February caused the PSCs to diminish more rapidly. The effect of water vapour in 2012/13 was the largest among the simulated years, the increase in NAT and STS areas between the Min and Interim simulations were about 3 million $km^2$. In winter 2013/14 both NAT and STS maximum areas were nearly as high as in 2012/13 (about 23 million $km^2$ and 11 million $km^2$, respectively), but large areas of PSCs persist much longer than in winter 2012/13. The maximum difference between Interim and Min simulation was about 2 million $km^2$. Also in the 2015/16

winter the NAT and STS areas were larger than in 2010/11, but the PSCs did not persisted as late as in 2010/11. The increase in type 1 PSC area due to increased water vapour was smaller than in 2013/14, about 1.5 million $km^2$.

Table 1 shows cumulative ICE, NAT and STS PSC areas at $55\,hPa$. The largest cumulative ICE areas are always seen in the Max simulations. In the Min simulations there are only very small or no ICE PSC areas, with the exception of the winter 2015/16, when considerable ICE PSC area was present in all runs. However, in 2015/16 the ICE PSCs occurred mainly in

December and January, while in 2010/11 the ICE PSCs occurred from January until the end of February. The timing of the PSCs is important for chlorine activation and ozone loss as discussed later. The effect of water vapour on the cumulative ICE PSC area is larger in warm years than in cold years. The NAT and STS areas also strongly depend on winter temperatures – the maximum NAT and STS areas are simulated in the coldest winter 2015/16 in every simulation, while the smallest areas are simulated during the warmest winter 2012/13. However, unlike ICE PSC, the NAT and STS clouds are formed in every winter.

The formation of type 1 PSCs is less sensitive to changes in water vapour concentration than the ICE PSCs. The relatively large changes in water vapour between different simulations results in relatively small changes in the cumulative NAT and STS area.

### 3.3   Chlorine activation

In early winter chlorine is present as reservoir compounds (HCl and $ClONO_2$), which do not destroy ozone. In the cold conditions within the polar vortex the chlorine species are transformed through heterogeneous reactions, into intermediate

species such as $Cl_2$. When sunlight reaches the polar vortex these species are easily dissociated to form active chlorine species that participate in the catalytic ozone depletion cycles, i.e. $ClO_x$ ($ClO$, $Cl_2O_2$ and $Cl$). Active chlorine is transformed back to reservoir species through reactions with $NO_2$ and $CH_4$, however if PSCs are present the regeneration of $ClO_x$ is sustained.

Figure 6 shows the fraction of reservoir, intermediate and active chlorine species at $55\,hPa$ in the Min and Max simulations. The results from the Max simulation are represented by the upper limit for the intermediate (magenta) and active species

(green), and by the lower limit for the reservoir species (black). The chlorine fractions from the Interim simulation always fit within the range from the Min and Max simulations. The timing of the changes in the partitioning of the chlorine species correlates well with the occurrence of PSCs, e.g. the chlorine reservoir species start to transform into intermediate species when the STS PSCs appear (Fig. 5).

**Table 2.** Vortex-mean mixing ratio of $ClO_x$ integrated over the whole winter (ppbv*day) and as monthly mean concentration (ppbv) in the Min, Max and Interim simulations. Percentage in parentheses indicate the effect of SWV concentration changes compared to the Interim simulation.

| | Year | 2010/11 (Cold) | 2012/13 (Warm) | 2013/14 (Intermediate) | 2015/16 (Cold) |
|---|---|---|---|---|---|
| Winter | Interim | 152 | 73 | 113 | 119 |
| | Min | 145 (-5 %) | 72 (-1 %) | 101 (-11 %) | 117 (-2 %) |
| | Max | 161 (+6 %) | 76 (+4 %) | 120 (+6 %) | 123 (+3 %) |
| November | Interim | 0.12 | 0.12 | 0.11 | 0.12 |
| | Min | 0.14 | 0.13 | 0.12 | 0.14 |
| | Max | 0.13 | 0.13 | 0.12 | 0.13 |
| December | Interim | 0.35 | 0.74 | 0.28 | 0.43 |
| | Min | 0.41 | 0.79 | 0.38 | 0.48 |
| | Max | 0.36 | 0.72 | 0.34 | 0.43 |
| January | Interim | 1.4 | 0.94 | 1.28 | 1.59 |
| | Min | 1.49 | 1 | 1.43 | 1.61 |
| | Max | 1.36 | 0.98 | 1.34 | 1.55 |
| February | Interim | 1.5 | 0.22 | 1.03 | 1.23 |
| | Min | 1.59 | 0.23 | 1.22 | 1.28 |
| | Max | 1.56 | 0.23 | 1.17 | 1.25 |
| March | Interim | 1.21 | 0.17 | 0.52 | 0.41 |
| | Min | 1.35 | 0.18 | 0.71 | 0.48 |
| | Max | 1.31 | 0.18 | 0.66 | 0.46 |
| April | Interim | 0.26 | 0.14 | 0.14 | 0.052 |
| | Min | 0.38 | 0.15 | 0.14 | 0.054 |
| | Max | 0.34 | 0.14 | 0.14 | 0.053 |

In the 2010/11 winter chlorine activation starts in the latter half of December and the fraction of $ClO_x$ is large through the January–Mach period, reaching a maximum of about 85 %. The active chlorine starts to transform back to reservoir species (mainly $ClONO_2$) in the beginning of March. In early April when the PSCs disappear the active chlorine rapidly decreases to background values.

In the 2012/13 winter chlorine activation starts slightly earlier than in the other years, but already in the beginning of February most of the chlorine has converted back to the reservoir species due to a SSW. The maximum fraction of $ClO_x$ is about 75 %,

and is reached already in the end of December. The active chlorine decreased during January, and in the beginning of February the concentration reached nearly background values.

The beginning of the winter 2013/14 winter was very cold, the chlorine activation started in mid December, and the maximum chlorine activation was reached already in the end of January. After that the vortex temperature increased and chlorine transformed back to reservoir species. The maximum fraction of $ClO_x$ was slightly lower than in cold winters, about 70 %.

The 2015/16 winter started similar to the cold winter 2010/11 and nearly all of the chlorine was activated at the beginning of January, but the deactivation started already in the end of January making the period with high $ClO_x$ shorter than in winter 2010/11. In the end of February the vortex temperature increased and chlorine transformed back to reservoir species. The maximum fraction of the activated chlorine of about 80 % was reached by the beginning of January.

The water vapour concentration seems to strongly affect the transformation of chlorine from the reservoir species to the intermediate ones in the beginning of the Arctic winter. The fractions of intermediate and reservoir chlorine species change significantly with water vapour concentration during November and December. The water vapour concentration affects the composition of binary aerosols and STS. When more water condenses to the particles the uptake coefficients for the heterogeneous reactions and the surface area increase, i.e. the reaction rates increase. The difference between Min and Max simulations can be up to 30 % when about half of the reservoir chlorine have transformed to intermediate species, just before the concentration of active chlorine species starts to increase. The difference in concentration of active chlorine and reservoir species between the Min and Max simulations are smallest during the cold periods, due to heterogeneous chemistry on the PSCs (Fig. 6). The cold winter 2015/16 shows a very small range, and the intermediately cold winter 2013/14 a wider range in concentrations. The water vapour content has less effect on the chlorine partitioning in cold winters. In the cold spring 2011 the difference in chlorine activation between Min and Max simulations was about 5 % on average, it reached nearly 20 % in the beginning of April, when the chlorine deactivation was fast. In the warm winter 2012/13 the change was less significant, about 5 % during the winter. In the 2013/14 winter the difference in chlorine activation between Min and Max simulations reached 10 % in the latter half of January, from mid February to mid March the difference was 15–18 %. The chlorine activation in 2015/16 winter seemed to be less dependent on water vapour content, probably due to the ICE PSCs that appear in all simulations. Therefore, the conditions were favourable for high chlorine activation in all simulations. The difference in the fraction of activated chlorine between simulations is only a few percents, only when the deactivation starts (in the end of February) the difference is more than 5 %.

The effect of increased water vapour seems to be large in moderately cold years, i.e. when the chlorine activation is not so complete. The start and end of chlorine activation correlate with the appearance of STS and NAT. The ICE PSCs did not significantly increase the chlorine activation. For example in 2012/13 and 2013/14 winters there were no ICE PSC in the Min simulation, but the chlorine activation was nearly as high as in the Max simulation, which had ICE PSCs.

Table 2 shows the vortex averaged $ClO_x$ as a cumulative sum over the whole winter, and as monthly mean concentration. The sums are integrated from November to April. The cumulative sum provides information about both the duration of the chlorine activation period and the concentration of $ClO_x$, while the monthly average concentration shows the timing of chlorine activation. The cumulative chlorine activation was largest in winter 2010/11 and smallest in 2012/13. The activation started

in November every year, but remained small until December. The winter 2010/11 differs from the others, with high chlorine activation from January to March, giving the largest cumulative sum of the studied winters. Even in April the $ClO_x$ concentration remains elevated. The warm winter 2012/13 had the smallest cumulative chlorine activation, significant chlorine activation was seen only in December and January. The changes in water vapour between the Min/Interim/Max simulations had the largest
effect on the cumulative $ClO_x$ in moderately cold winters (2010/11 and 2013/14), where the increase of $ClO_x$ from Interim to Max was 3 to 6 % and change of $ClO_x$ from Interim to Min was -1 to -11 %. In the cold winter 2015/16 the respective changes were only 3 and -2 %, and in the warm winter 2012/13 the changes were 4 and -1 %.

## 3.4    Ozone loss

Figure 7 shows the mean chemical total ozone loss within the polar vortex for all the studied winters 2010/11, 2012/13, 2013/14
and 2015/16. The total column chemical ozone loss was calculated by subtracting the passive transported total ozone from the modelled total ozone. In Fig. 7 the polar vortex is defined using the potential vorticity limit 36 PVU only at the 475 K level. The figure shows the chemical ozone depletion in the Interim, Min and Max simulations as well as the difference in the loss between the Min and Max simulations. The passive ozone tracer was initialized every year on December $1^{st}$, when it was set equal to the ozone distribution in the model on that day. Chemical processes start to reduce ozone already in December, but
they have minor effect on the total wintertime ozone loss. In January the chemical processes become more intensive, when the chlorine activation increases (see Fig. 6).

In general the ozone loss is larger in cold years. The largest ozone loss was simulated in the beginning of April 2011 when about 90 DU ozone had been destroyed according to our model. FinROSE seems to underestimate the ozone loss; for example Sinnhuber et al. (2011) and Manney et al. (2011) simulated 120 DU ozone loss and Pommereau et al. (2013) even 170 DU in
winter 2010/11. If we look at maximum ozone losses instead of the polar vortex mean losses, then the numbers are larger. The maximum ozone loss in 2010/11 within the polar vortex was 128 DU, which is comparable to the value given by Sinnhuber et al. (2011).

The ozone loss in the warm winter 2012/13 differs from the loss in colder winters (2010/11 and 2015/16). The maximum average ozone loss in the polar vortex in the 2012/13 winter was only 23 DU, because the polar vortex was unstable and small.
By the mid of April 2014 the simulated vortex mean ozone loss was 79 DU in the Interim simulation. Before mid February, i.e. during the coldest period, there was very little effect from the changes in SWV. A relatively small ozone loss of 56 DU was simulated in 2015/16, which was due to the unstable polar vortex, which split and warmed, stopping the catalytic ozone cycles and ozone loss in the beginning of March, i.e. earlier than in 2010/11 and 2013/14.

Figure 7 also shows the difference of polar vortex averaged chemical ozone loss between Min and Max simulation. It tells
how much the water vapour concentration change affects the ozone loss. The difference is largest (about 15 DU) in 2013/14, a moderately cold winter, with significant ozone depletion. Another winter, 2010/11, with significant ozone loss and cold, but not extremely cold conditions showed the second largest effect from addition of water vapour, about 10 DU. In 2012/13, when the ozone loss stopped very early, the difference between Min and Max simulations was about 5 DU. The 2015/16 winter started as very cold, but was ended early by a SSW. The difference in ozone loss between the simulations remained very small up to mid

February, by mid March the difference was about 7.5 DU. A reduction in the water vapour decreased ozone loss in every winter. In the Interim simulation the deepest ozone losses were about 2–9 DU (6–11 %) deeper than in the Min-simulation. The effect from an increase in water vapour from Interim to Max was about the same. In the 2010/11 winter the loss increased by about 7 %, while Sinnhuber et al. (2011) and Vogel et al. (2011) reported an increase of ozone loss by 20 % and 10 % (respectively) with a water vapour increase of about the same magnitude as considered here. Thus, our estimates are slightly smaller than those by Vogel et al. (2011). An additional sensitivity experiment showed that the difference compared to other studies can be, at least partly, due to the coarse horizontal resolution in FinROSE ($3° \times 6°$), which is not sufficient to fully capture the deepest ozone loss. Specifically, repeating Interim simulation for winter 2010/11 with higher resolution ($1.5° \times 3°$) than in the original simulation showed larger ozone loss by 15 DU.

The changes in the amount of water vapour are in the range that was tested here and are not very important for ozone loss in cold years. In the 2010/11 winter the chlorine activation was nearly complete in the Arctic polar vortex, and additional water vapour did not increase chlorine activation and and thus did not increase the ozone depletion. Increasing water vapour concentration (compared the Interim simulation) strengthens ozone loss at least by 4 DU for other winters except for the 2011 winter when the increase was not significant.

**Table 3.** Maximum polar vortex-mean ozone loss produced by full chemistry, heterogeneous chemistry and separately by the STS, NAT and ICE part in Min, Max and Interim simulations ( DU). Percentages show the fraction due to each part relative to the full chemistry.

|  | Year | 2010/11 (Cold) | 2012/13 (Warm) | 2013/14 (Intermediate) | 2015/16 (Cold) |
|---|---|---|---|---|---|
| Min | Full chemistry | 84 | 21 | 70 | 53 |
|  | Heterogeneous part | 50 (60 %) | 13 (62 %) | 32 (45 %) | 33 (62 %) |
|  | STS, NAT and ICE | 20 (24 %) | 3 (15 %) | 13 (19 %) | 11 (21 %) |
| Interim | Full chemistry | 90 | 23 | 79 | 56 |
|  | Heterogeneous part | 56 (62 %) | 14 (63 %) | 40 (51 %) | 35 (63 %) |
|  | STS, NAT and ICE | 30 (33 %) | 5 (24 %) | 23 (30 %) | 17 (30 %) |
| Max | Full chemistry | 91 | 25 | 85 | 59 |
|  | Heterogeneous part | 56 (62 %) | 15 (62 %) | 45 (53 %) | 37 (63 %) |
|  | STS, NAT and ICE | 34 (37 %) | 7 (28 %) | 30 (35 %) | 19 (33 %) |

To better understand the mechanism of SWV influence on ozone loss, simulations without heterogeneous chemistry were performed. From those simulations ozone loss caused by heterogeneous chemistry can be separated by subtracting the total ozone simulated without heterogeneous chemistry from that simulated in the full chemistry run. Two different set-ups were used for testing the effect of the heterogeneous chemistry. In the first gas-phase chemistry simulation the heterogeneous chemistry was not included at all. In the second simulation the formation of PSCs was limited by setting the air temperature passed to the heterogeneous chemistry module to 200 K, similarly to what was done in Karpechko et al. (2013). The increase of water

vapour (Max simulation) did not increase the ozone loss, but in the Min simulation ozone depletion was reduced by 6 DU. This setting allows some heterogeneous processing on binary aerosols and some STS that are very dilute in $HNO_3$, and due to the temperature limit the surface area densities will remain quite small. Table 3 summarises ozone loss characteristics during the studied years and shows the loss produced by full chemistry, heterogeneous chemistry and separately by the NAT, STS and
ICE PSCs in the Interim, Min and Max simulations.

In the Interim simulation with full chemistry in 2010/11 about 90 DU ozone was depleted, of which the heterogeneous chemistry caused depletion of 56 DU, i.e. about 62 % of the total ozone loss. Heterogeneous chemistry due to PSCs destroyed 30 DU ozone, which was about 33 % of the total loss. The result indicates that chlorine activation on the binary aerosols has a significant role in ozone depletion. Some studies suggest that binary aerosols are more important for chlorine activation
than PSCs (e.g., Drdla and Müller, 2012; Wohltmann et al., 2013; Kirner et al., 2015). The increase of water vapour (Max simulation) did not increase the ozone loss, but in the Min simulation there was 6 DU less ozone depletion. This is consistent with the results by Kirner et al. (2015), who argue that the contribution of ICE PSCs to the ozone loss is always less than 5 % in the Antarctic spring, where the chlorine activation is nearly complete.

In the warm 2012/13 winter the ozone loss is only 23 DU, and the heterogeneous part is 63 % of it. NAT, STS and ICE PSCs
caused only a small part of the total heterogeneous chemistry driven ozone loss (24 %). The loss caused by heterogeneous chemistry increased with increasing water vapour, but remained small even in the Max simulation. In 2013/14 the heterogeneous chemistry caused about 40 DU ozone destruction, which is about 51 % of the ozone loss and NAT, STS and ICE about 23 DU (30 %), when the total ozone loss was 79 DU in the Interim run. The increase in SWV from Interim to Max increased the ozone loss by about 6 DU and the decrease in SWV from Min to Interim decreased the ozone loss by 9 DU. Thus, water
vapour changes have a larger effect on ozone loss in moderately cold years than in cold ones. The ozone depletion due to heterogeneous chemistry increased with water vapour, even though the fraction due to heterogeneous chemistry was smaller than in 2010/11 and 2015/16.

In the 2015/16 winter the heterogeneous contribution was largest when compared to other simulated years, reaching even 63 % of the ozone loss, and also the STS, NAT and ICE part was large, 30 %. The total ozone loss is however only 56 DU.
When the water vapour content was increased from Interim to Max simulation, the fraction due to the heterogeneous chemistry remained the same, but the fraction due to STS, NAT and ICE PSCs increased.

Based on the results in Table 3 it can be concluded that nearly all SWV impact on ozone loss is due to heterogeneous chemistry. For example in 2010/11 the ozone loss without heterogeneous chemistry was 34 DU in Interim, 34 DU in Min and 35 DU in Max simulation and only the heterogeneous part changed from model run to model run. In 2013/14 the non-
heterogeneous contribution is about 39 DU, and in 2015/16 about 21 DU, i.e. in warm years it is larger than in cold years. In 2012/13 the non-heterogeneous contribution was only 9 DU, but also the total ozone loss is very small. However, the heterogeneous contribution is about 62 % of the total ozone loss in all other winters than 2013/14 and also with both increased and decreased water vapour. The fraction of STS, NAT and ICE driven chemistry changes with water vapour concentration.

Finally we analyse the vertical distribution of the ozone loss and the effect of SWV on ozone loss, which is shown in Fig. 8.
The largest ozone loss was simulated in 2010/11, when the ozone destruction in the Interim run with normal SWV was about

1.4 ppmv between 60–30 hPa. The ozone depletion increased by 0.2 ppmv between the Min and Max simulations. In 2012/13 the maximum ozone reduction was almost the same as in 2010/11, but it occurred at higher altitude and lasts for shorter period than in 2011. The effect of the increase in water vapour from Min to Max simulation had only a minor effect on the ozone depletion in 2012/13. The heterogeneous chemistry and chlorine activation did not have an important role in the warm

conditions and the ozone loss between 60–30 hPa remains very weak. In winters when the polar vortex is unstable and small or disturbed the Brewer–Dobson circulation brings more $NO_x$-rich air into the polar vortex than usual. Hence the ozone loss in the 2012/13 winter was produced mostly by $NO_x$ chemistry as shown previously by e.g., Sagi et al. (2017), and can be seen from FinROSE results by comparing the simulations with and without the heterogeneous chemistry. The total ozone column loss in this winter remained smaller than in cold years, when the ozone depletion is driven by halogens. In the 2014 spring the

conditions in the polar vortex remained favourable, but the temperature was not as low as in 2011. The ozone loss developed steadily, but remained moderate. The two winters 2010/11 and 2013/14 with the most favourable conditions for halogen driven ozone depletion showed the largest increase in ozone loss with water vapour. The effect was more pronounced in 2013/14, which was the warmer of the two winters. The winters 2010/11 and 2015/16 look similar during January–February, but the ozone loss became much more severe in 2010/11 due to favourable conditions in March–April. In 2015/16 there was a very

cold period, but it occurred too early to have a large impact on the ozone depletion, and therefore the water vapour increase had only a moderate effect later in the spring. In 2013/14 the largest ozone loss was about 1.1 ppmv between 60 and 30 hPa while in 2015/16 it was about 1 ppmv at the same altitude. Livesey et al. (2015) and Sagi et al. (2017) showed results from 450 K level, and their ozone losses were about 2 ppmv in winter 2011. In winter 2013 Sagi et al. (2017) had about 1.5 ppmv ozone loss, which is about the same what we found.

**4   Discussion and conclusions**

Khosrawi et al. (2016) showed that an increase in SWV and a cooling of the stratospheric temperature enhance each other, so that the area of PSCs increases and that these can last longer in the vortex. The ozone loss can thus increase although the halogen loading has been decreased. In this study, rather than artificially changing the temperature we used meteorological fields from seven winters during the period 2010–2016 with different temperatures and dynamical conditions in the stratosphere. We

changed the water vapour content in the tropical tropopause region according to the CCMVal-2 simulations. The water vapour entry concentration is controlled by the cold point in the TTL, and the distribution of SWV is largely determined by this entry concentration together with the transport and the contribution from methane oxidation. Our results show that, as expected, a wetter/drier tropical tropopause leads to a wetter/drier Arctic polar vortex and also the size of polar ozone depletion changes when the water vapour changes, which affects also the Arctic ozone loss.

30       A reduction in SWV decreases the ozone loss due to heterogeneous processes by decreasing the PSC formation. An increase in SWV instead makes the heterogeneous chemistry more important. As expected, heterogeneous chemistry is less important in warm winters. I.e. in 2012/13 only 14 DU ozone depletion was initiated by heterogeneous chemistry in the Interim simulation. The corresponding loss in the winter 2010/11, with persistent cold conditions, was 56 DU. The increase in loss with water

vapour was small, i.e. the loss increased from 50 to 56 DU from the Min to the Max simulation. In the winter 2012/13 the corresponding losses were 32 and 45 DU, i.e. the water vapour concentration had the largest impact in moderately cold winters. If the winter is cold enough, the increase is less important, because the PSCs may form even at low water vapour concentrations, and the chlorine activation is already nearly complete in Arctic vortex, therefore the water vapour increase is less important.

Winters in the polar stratosphere are often divided into cold, or dynamically inactive, and warm, or dynamically active. In the cold Arctic winters the polar vortex is stable and more PSCs are formed and halogens can destroy ozone. Warm conditions in the polar winter stratosphere are often due to SSW, which allows $NO_x$-rich air masses from the mesosphere to enter the vortex and take part in the ozone depletion (Sagi et al., 2017). Cold polar stratospheric winters differ from the warm winters regarding the ozone loss and the fraction of ozone loss initiated by heterogeneous chemistry, during cold stratospheric winters

the PSC areas are larger and thus chlorine activation within the polar vortex is more complete. A lack of water leads to less ICE PSCs, and therefore to less $ClO_x$. However, the ICE PSC area is not the only explaining factor for ozone loss. The type 1 PSCs that form at higher temperatures are responsible for a large fraction of the chlorine activation. The formation of STS and NAT is limited by the partial pressure of nitric acid, sulfuric acid and water and hence the concentration of water vapour is not the only thing affecting the NAT and STS areas. However, the dry conditions in the Min simulations have some limiting effect

on the peak NAT and STS areas.

    The cold Arctic winter 2010/11 differs from the others by an especially long chlorine activation period, which lead to large ozone depletion. In the warm Arctic winter 2012/13 the polar vortex was weak; however it was shifted to south where it was exposed to sun-light earlier than usually, and thus ozone loss could start earlier. The ozone loss was however weak because chlorine activation remained very low. The ozone depletion in 2012/13 occurred at higher altitudes than in the other years,

because of the $NO_x$ induced ozone loss. The 2013/14 Arctic winter was moderately cold, and the ozone depletion was second largest among the considered winters. In this winter the effect of water vapour changes on ozone loss was the largest across the studied winters. The Arctic winter 2015/16 started as extremely cold (Matthias et al., 2016; Manney et al., 2016), but the polar stratosphere warmed early terminating chlorine activation and leaving ozone loss relatively low, despite the fact that the cumulative ICE areas were extremely large.

Chemical ozone destruction inside the Arctic vortex varied between 23 and 90 DU in the Interim-simulations, 25 and 91 DU in the Max-simulations and 21 and 84 DU in the Min simulations. We find that the meteorological conditions are more important for the ozone depletion than the concentration of water vapour. Also the fraction of heterogeneous chemistry in the ozone loss is more dependent on the temperature than on the water content. Livesey et al. (2015) came to a similar conclusion, when investigating ozone loss based on the MLS observations.

MacKenzie and Harwood (2004) showed from their chemistry–climate model simulations, that the increase of water vapour increases the area of PSCs both by microphysical effects and due to lowering the stratospheric temperatures. The microphysical processes cover about 20 % of the increase and the rest is due to cooling of the polar stratosphere. In our study we only changed the water vapour concentration. However, the temperature effect can be seen by investigating different years. The cumulative ICE areas between studied years varied by a factor of 30, and in cumulative NAT and STS by 2.7 and 2.2, respectively.

MacKenzie and Harwood (2004) got about 15 % more ozone loss at 465 K level with less than 1 ppmv additional water vapour

without changing temperature. In our study the ozone loss increased by 1 DU (1 %) in 2010/2011, 2 DU (9 %) in 2012/2013, 6 DU (8 %) in 2013/2014 and 3 DU (5 %) in 2015/2016 when the water vapour concentration was increased by about 2 ppmv. When the water vapour was instead decreased by about 1.5 ppmv, the ozone loss decreased by 6 DU (7 %), 2 DU (9 %), 9 DU (11 %) and 3 DU (5 %), respectively. The small contribution due to water vapour increase in The Arctic winter 2010/11 can

be compared to the results of MacKenzie and Harwood (2004) in the Antarctic vortex. There the chlorine activation is nearly complete in every winter. Also in the Arctic winter 2010/11 nearly all chlorine in the polar vortex was activated, and additional water vapour did not change the activation and, thus not either the ozone depletion.

Note that effects of changing water vapour concentration on air temperature, not accounted for here, would probably have increased the the impact of water vapour on ozone loss. The indirect impact comes through water vapour radiative impact on

stratospheric temperatures. Tian et al. (2016) estimates that a 2 ppmv increase of water vapour would cool the polar stratosphere by approximately 2 K, while Rex et al. (2004) estimates that a 1 K cooling could increase ozone loss in the Arctic by 15 DU. Thus based on these estimates a water vapour increase of 2 ppmv, similar to the difference between Interim and Max runs, could result in up to 30 DU additional ozone loss. This estimate suggests that the direct water vapour impact on ozone loss quantified in our experiments may account for only about one fifth of the total ozone loss, but in order to confirm this estimation

a designed experiment with a chemistry–climate model would be needed.

In summary, we find that variability of stratospheric water vapour of 3.5 ppmv, comparable in magnitude to uncertainty in simulated water vapour concentration near the tropical tropopause, results in differences in simulated Arctic ozone loss up to 15 DU, i.e. more than 15 % of the total chemical ozone loss in the Arctic vortex. Better understanding of tropical processes contributing to the stratospheric water vapour concentration, and thus constraining stratospheric water vapour, would therefore

reduce the uncertainty in Arctic ozone loss and improve future projections of ozone layer recovery.

*Competing interests.*   The authors declare that they have no conflict of interest.

*Data availability.*   Data from the FinROSE simulations is available from the authors upon request. MLS data is available at https://mls.jpl.nasa.gov.

*Acknowledgements.*   We acknowledge ECMWF for providing us the ERA-Interim reanalysis data. We also want to thank the UARS reference atmosphere project and MLS/Aura teams for water vapour data. The MLS data were obtained through the Aura MLS website (http://mirador.

gsfc.nasa.gov/). Funding by the Academy of Finland through the UTLS project (140408) and by the EU through the project GAIA-CLIM is gratefully acknowledged.

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

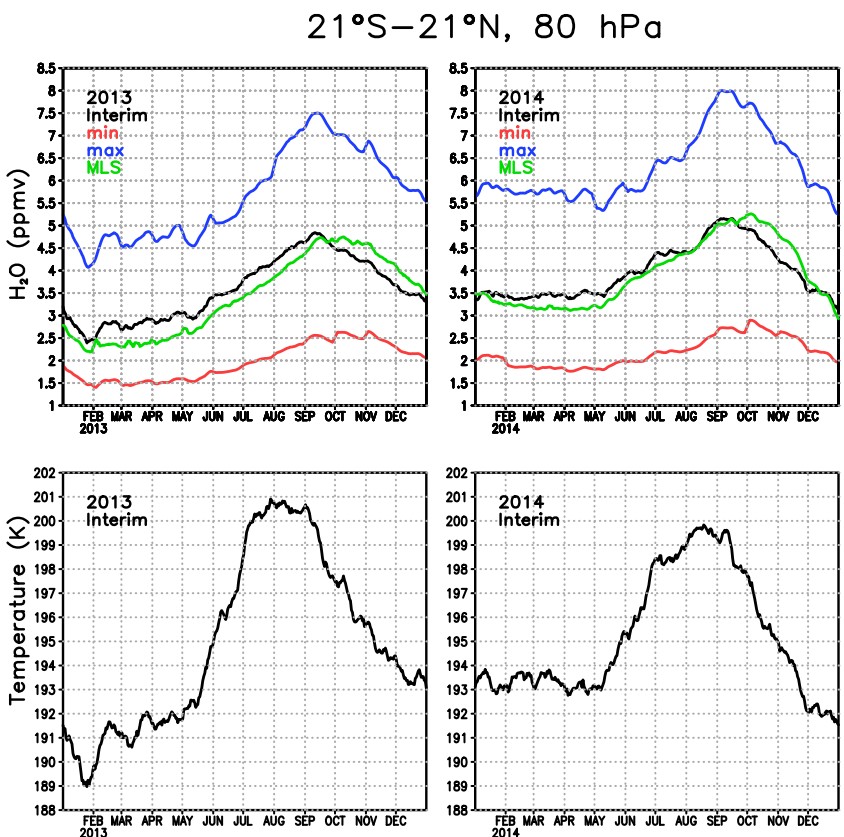

**Figure 1.** Water vapour and temperature around the tropical tropopause between $21^\circ$ S and $21^\circ$ N at level $80\,\mathrm{hPa}$ in 2013 and 2014. Green line is Interim, blue Max, red Min simulation and black is MLS.

# Temperature (K) in vortex

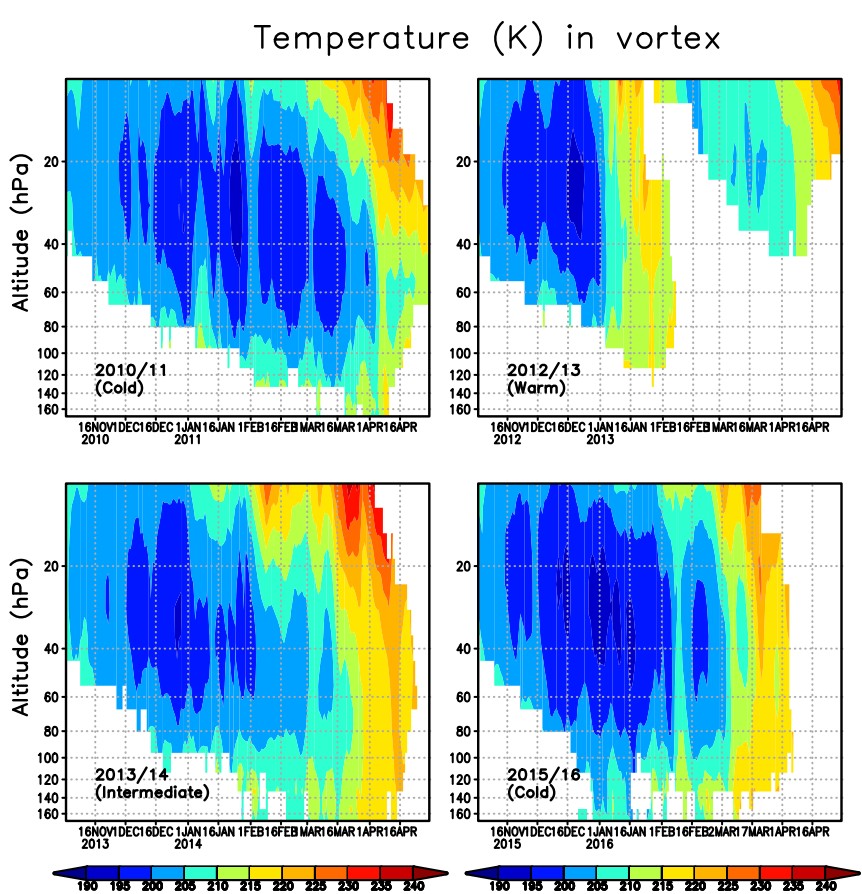

**Figure 2.** Vortex average temperature within the Arctic polar vortex between altitudes 170 and 10 hPa.

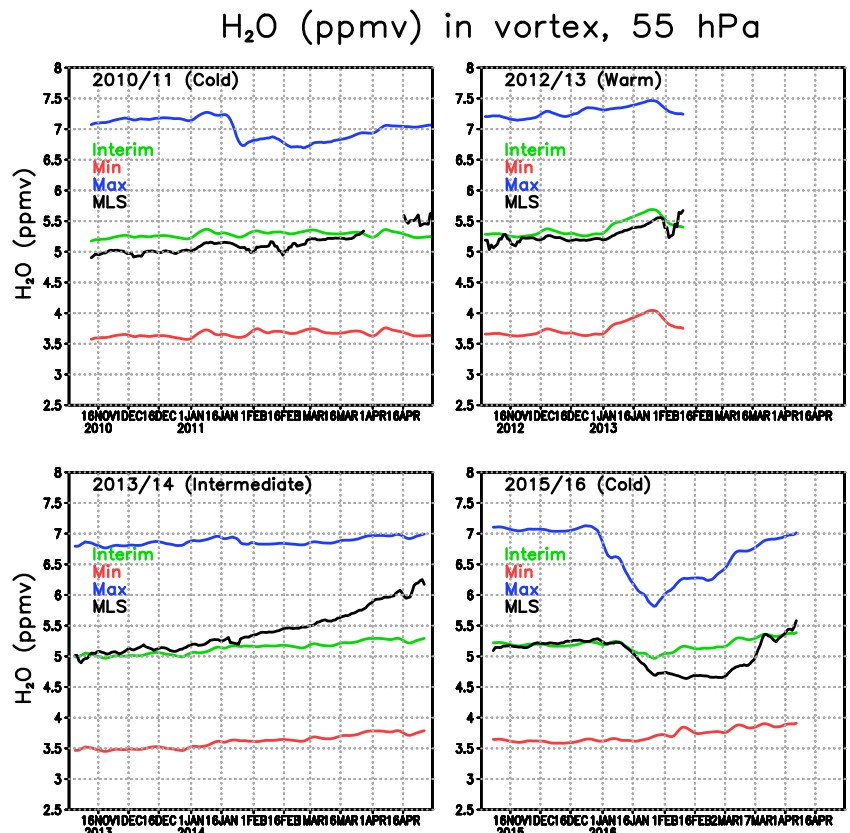

**Figure 3.** Water vapour concentration (ppmv) at 55 hPa within the Arctic polar vortex. Green line is Interim, blue Max, red Min simulation and black is MLS.

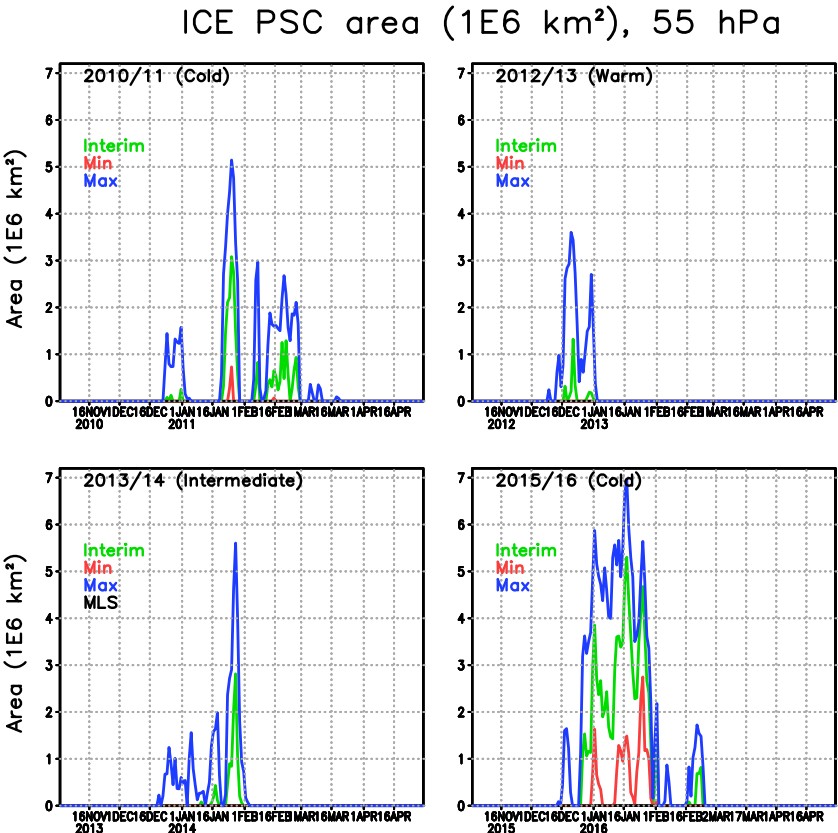

**Figure 4.** The areas of ICE PSCs ($10^6 \mathrm{km}^2$) within the Arctic polar vortex in the FinROSE simulations at 55 hPa. Green line is Interim, blue Max and red Min simulation.

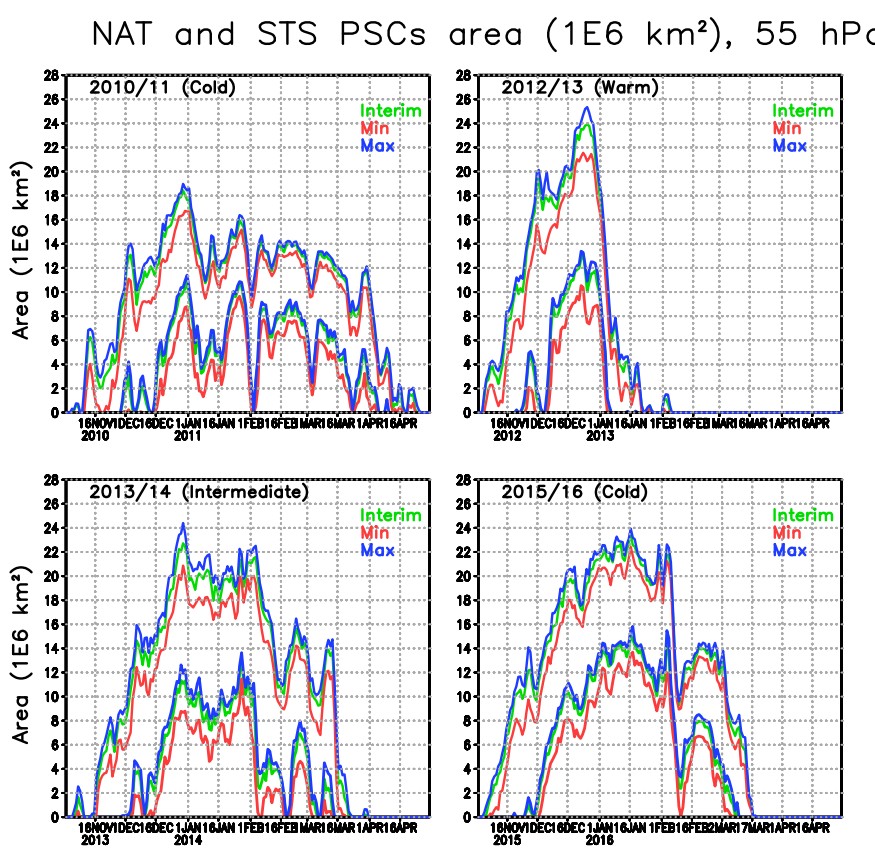

**Figure 5.** The area of NAT (thick lines) and STS PSCs (thin lines) $(10^6\,\text{km}^2)$ within the Arctic polar vortex in the simulations at $55\,\text{hPa}$. Green line is Interim, blue Max and red Min simulation.

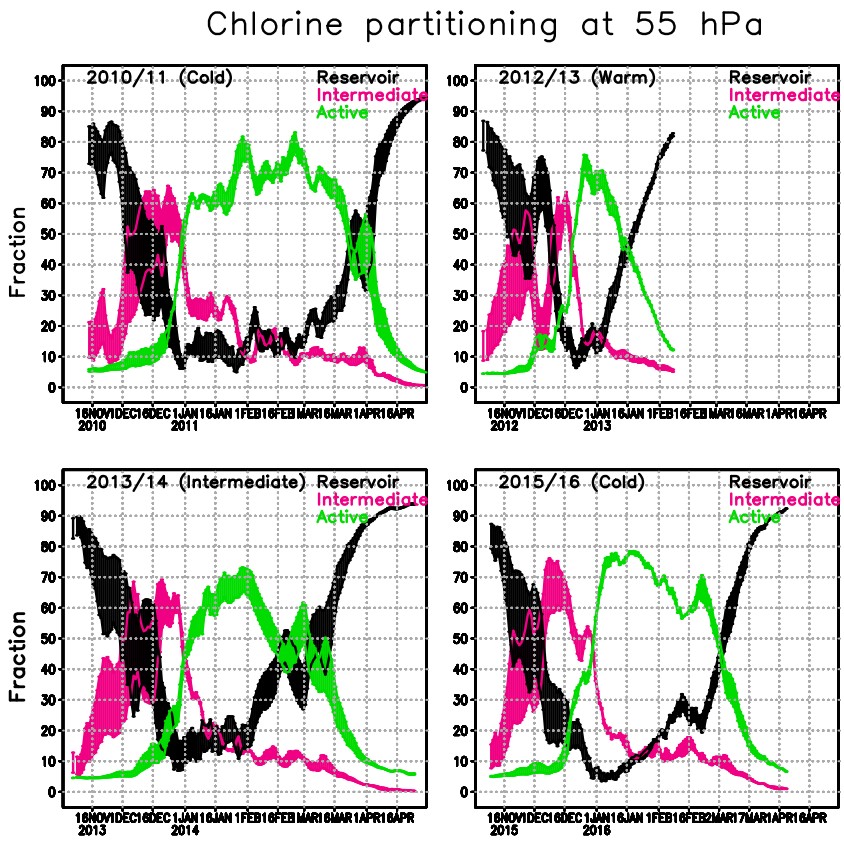

**Figure 6.** Chlorine partitioning (%) within the Arctic polar vortex at 55 hPa in the Min and Max simulations. Active form (green) is $Cl+ClO+2*Cl_2O_2$. Intermediate (magenta) contains $2*Cl_2+HOCl+OClO+BrCl+ClNO_2$ and reservoir chlorine (black) $HCl+ClONO_2$.

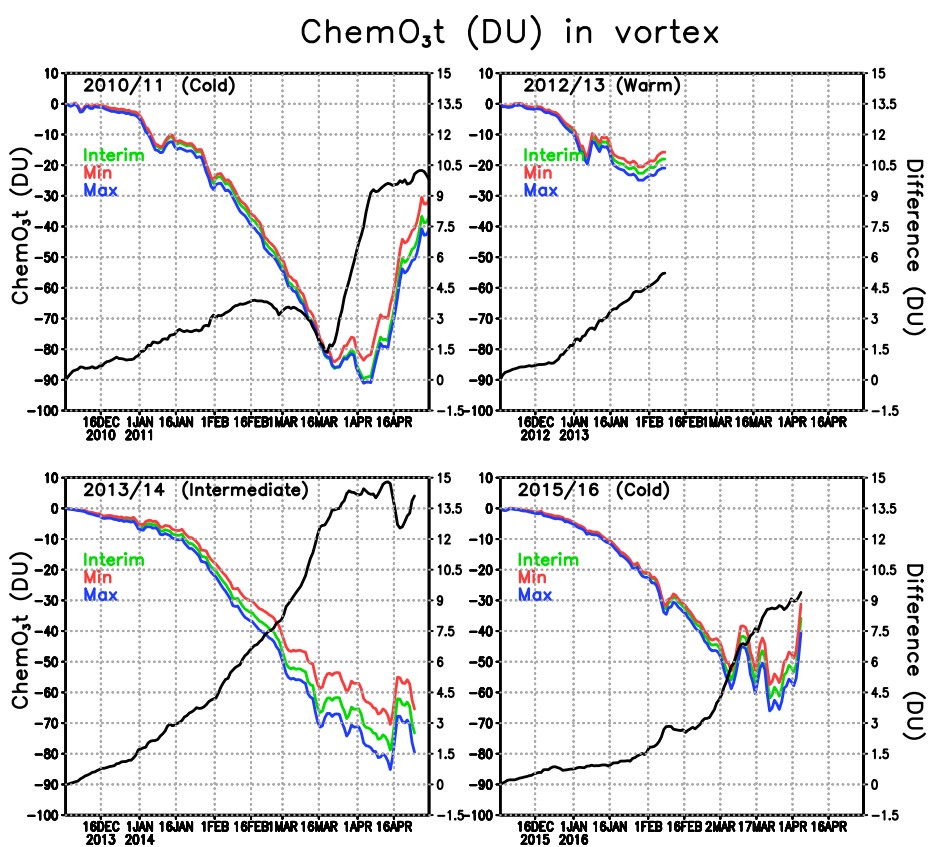

**Figure 7.** Chemical total ozone loss (DU) and difference between ozone loss in the Min and Max simulations within the Arctic polar vortex. Green line is Interim, blue Max and red Min simulation. The difference is in black.

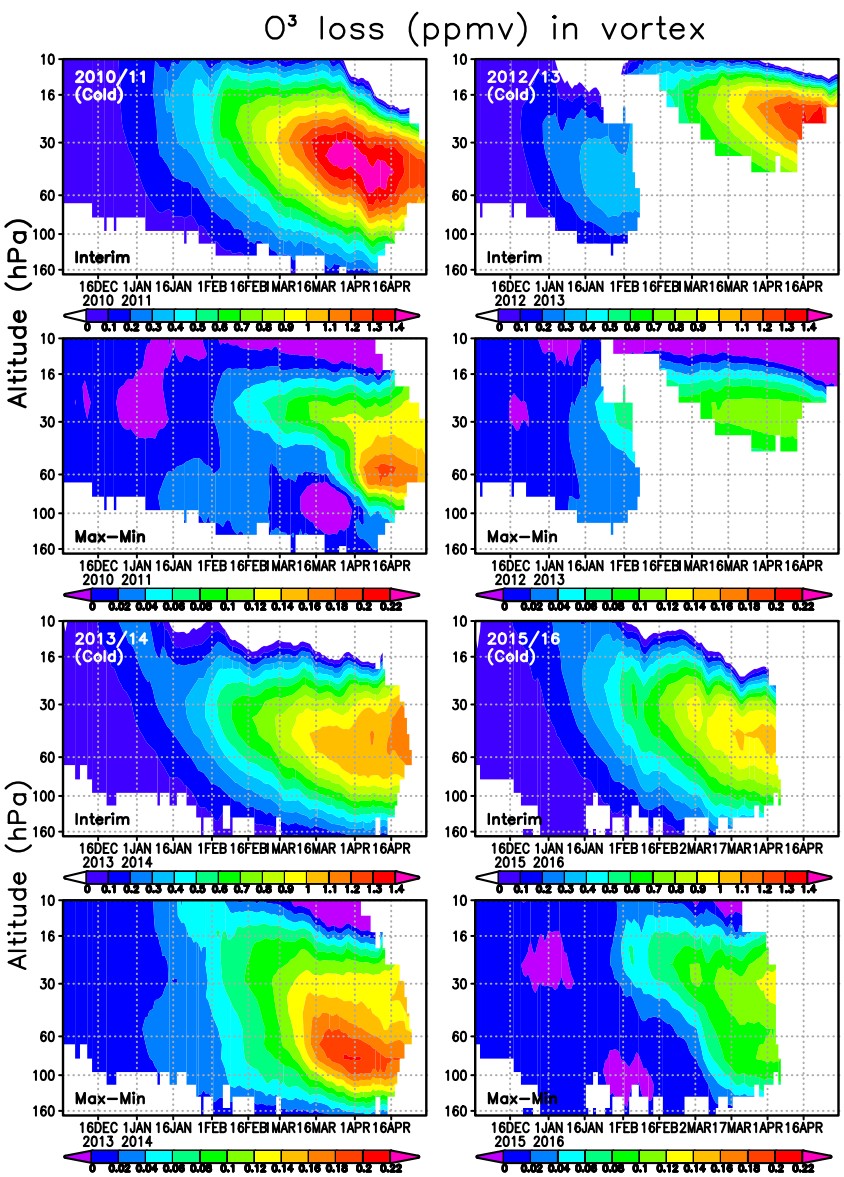

**Figure 8.** Averaged chemical ozone loss (ppmv) in the Interim simulation (upper panels) and the difference between Max and Min simulations (lower panels) within the Arctic polar vortex.