# Peer review of "Linking uncertainty in simulated Arctic ozone loss to uncertainties in modelled tropical stratospheric water vapour"

_Atmospheric Chemistry and Physics, 2018_

## Referee Comment (RC1) · Anonymous Referee #2 · 3 May 2018

**General**

The authors investigate the sensitivity of modelled Arctic ozone loss to the water vapour mixing ratio entering the tropical stratosphere in a chemical transport model. They guide the reader well step by step through the causal chain water vapour concentration → PSC volume → chlorine activation → ozone loss. The authors clearly state that the investigated question is different from investigating the effect of water vapour changes due to climate change (which would occur on such a timescale that also the concentration of chlorine- and bromine-containing species changes considerably). They also clearly state that they investigate only one aspect of the above-mentioned question,

namely the effect of water vapour on the surface area density of Polar Stratospheric Clouds (PSCs), neglecting the (probably stronger) effect of water vapour changes on ozone loss via changes of stratospheric temperature.

**Comments**

- An increase of water vapour may enhance heterogeneous chemistry by enlarging the air volume in which PSCs are formed (shown in Figs. 4 and 5) or via enlarging the surface area of existing particles (not shown). The authors seem to assume that the first effect is the dominant one. A discussion of this topic would help to complete the logic of the paper.

- The authors claim an important role of $NO_x$ chemistry in warm winters. I would appreciate plots showing this, e.g., altitude-time plots of vortex-averaged $NO_x$ or / and altitude-time plots of the corresponding reaction rates ($NO_2 + O \rightarrow ...$ for $NO_x$ chemistry, and, for comparison, $Cl_2O_2 + h\nu \rightarrow ...$ and perhaps $ClO + O \rightarrow ...$ for $ClO_x$ chemistry).

- The authors do not discuss the influence of heterogeneous $NO_x$ chemistry on ozone. Can the model results be used to answer the question whether the heterogeneous reaction $N_2O_5 + H_2O \rightarrow 2\,HNO_3$ reduces $NO_x$ and thus $NO_x$-driven ozone loss in cold years (compared to other years)?

**Minor comments**

- 1/17: "2-7% more ozone loss than in colder winters" $\Rightarrow$ Does this mean "2-7% stronger increase of ozone loss than in colder winters"?

- 2/1 and 14/3: "warms the climate" means "warms the troposphere"?

- 3/2-3: "cooling stratosphere ... slowing down some gas-phase reactions": Which gas-phase reactions are meant? In the polar stratosphere (during winter / spring) an important reaction is the three-body reaction $ClO + ClO + M \rightarrow Cl_2O_2$, the rate-constant of which **increases** for decreasing temperature.

- 5/3: "around 80 hPa": In view of the discussion in 5/29-31 this should be formulated more precisely, e.g. "at the cold point, which lies approximately x hPa below 80 hPa"

- 5/29-31: "leads ... by 3-4 weeks ... Brewer-Dobson circulation ... too fast": Does this mean that between the cold point and 80 hPa the ERA-Interim circulation takes 3-4 weeks less than the real circulation. Is the distance between the cold point and 80 hPa large enough to gain such a difference?

- 6/3: "gains a small amount of water": How, by horizontal mixing?

- 8/4: "For example too dry models may not be able to simulate a large Arctic ozone loss such as of 2010/11": How does this sentence relate to the preceding sentence?

- 9/15: "in spring": and also during southward excursions of air masses during winter

- 9/16: "PSCs sustain the regeneration of $ClO_x$": This is only possible if both reaction partners for a heterogeneous reaction are still present.

- 10/9: "rather short": The green curve in Fig. 6d lies above 60% for about 2 months. Is this meant by "rather short"?

- 10/17: "differs significantly ... during the period with high $ClO_x$": The difference is mostly less than 10%. Is this meant by "differs significantly"?

- 10/18: What exactly is meant by "chlorine activation period": the time when most chlorine exists in active form?

- 11/4: "... only at the 475 K level": Does this mean that the 475 K level is used for the definition of the whole vortex? If so, this should be mentioned (and perhaps be discussed) already in 6/7.

- 11/11: "FinROSE seems to underestimate the ozone loss, possibly due to a general 10% negative bias in total ozone": Why does an underestimation of the ozone concentration lead to a significant underestimation of the ozone loss? The rate of ozone loss in polar winter/spring is largely determined by the rates of reactions like $ClO + ClO + M \rightarrow Cl_2O_2 + M$ and $BrO + ClO \rightarrow$ ... .

- 11/23: "stopping the catalytic ozone cycles and ozone loss early": The ozone loss stops around the beginning of March (Fig. 7d). Is this meant by "early"?

- 13/4: "the heterogeneous chemistry destroyed about 36 DU of ozone": In fact, ozone is destroyed by gas-phase reactions. Heterogeneous chemistry produces (some of) the corresponding reactants.

- 14/24: "in higher level": Does this mean at higher altitudes?

- Tables 1, 2, 3: As a "service" for the reader the character of the years (warm, intermediate, cold) might be added (as was done in the figures). Perhaps the winter 2015/16 might be called "initially cold", in order to distinguish it from 2010/11.

**Technical details**

- Please check the "s" of the plural of substantives or singular of verbs (1/13, 2/3, 10/12, 10/16-17, 14/4, 15/29).

- 1/22: "processes. Especially" ⇒ "processes, especially"

- 2/7: "(2002)" ⇒ ", 2002"

- 4/9: "lagrangian" ⇒ "Lagrangian"

- 6/27: The abbreviation "BD" has not been defined before (and is used only once).

- 6/33: "(Fig. 3)" ⇒ Fig. 3

- 7/18: Really "0.3 ppt" (or 0.3 ppb)?

- Table 1: Please show only 2-3 non-zero digits.

- Table 2: "mixing ratio integrated over the winter of activated chlorine" ⇒ "mixing ratio of activated chlorine integrated over the winter"

---

## Referee Comment (RC2) · Anonymous Referee #1 · 26 May 2018

**General**

The topic of this paper is Arctic ozone loss in general and the sensitivity of Arctic ozone loss on the polar water vapour concentrations in particular. If the authors agree, they need to remove all the discussion of mid-latitude ozone loss and the related cycles. Further, discussion/citation of upper stratospheric ozone loss is not helpful.

The topic of the paper (water vapour sensitivity) is of importance and the the approach using a CTM to focus on the impact of water vapour on PSCs on heterogeneous chemistry is good. On the other hand, the results on the sensitivity will gain on impact if the representation of PSCs in the model, the simulation of chlorine activation and ozone

loss are clearly demonstrated in the paper. I do not think that this is the case in the present version of the paper (the cited papers are not sufficient, see below).

For example, the impact of humidity on heterogeneous chemistry should be different for different types of PSCs. If in this model the PSC types are dominated by a different type that in reality (or even in different models) carrying over the information on sensitivity deduced here can be problematic. There is some comparison with ERA-interim, but this is not the real world. Also there i.e. very little information on chlorine chemistry (except in Fig.6, where there is no distinction between HCl and ClONO2 for example, see also below).

The paper addresses a second case, namely a warm and dynamically unstable winter in the Arctic stratosphere. In such a case, in contrast to the halogen induced ozone loss in cold Arctic winters, there is little ozone loss below 500 K and ozone loss in the middle stratosphere (NOx-induced destruction) becomes important (e.g., Konopka et al., 2007; Sagi et al., 2017). However, this is a completely different chemical mechanism, which will have a very different dependence on water vapour (clearly no impact of heterogeneous chemistry). Therefore this second case needs to be very clearly discriminated from the "halogen" case throughout the paper.

In summary, I recommend focusing the paper and a providing a better basis and justification for the work presented. I also suggest a better balance of the cited literature (as stated in the quick review, I am not suggesting to cite all the references mentioned here). I am sorry for the many critical remarks but I believe that a revised paper taking into account these comments would be much stronger than the present version.

**General**

The discussion of the HOx chemistry in mid-latitudes and in the upper stratosphere (and the associated references) in the introduction is confusing. In the Arctic, in cold winters, an increase in water vapour is expected to enhance ozone loss (assuming that

a substantial amount of stratospheric chlorine is present) by enhancing heterogeneous reactions (Shi et al., 2001; Drdla and Müller, 2012). The situation can actually be more complicated as there is no simple argument that "more PSCs" means "more ozone loss". This is also a finding of this paper, if one considers the case of the cold Arctic winter 2010/11. The latter point should be more emphasized in the paper I suggest.

Recently, climatologies of PSC occurrence have become available from observations (Spang et al., 2017; Pitts et al., 2018). The question how important the discrimination between different PSC types in a model is for a successful simulation of polar ozone loss is not trivial. This paper could contribute substantially to this issue; perhaps more than visible in the present version of this paper. This aspect could be very relevant for the discussion of the sensitivities on water vapour (which will be different for different types of PSCS). Finally, models might misrepresent PSC volume density compared to observations; Khosrawi et al. (2018) report that the comparison between the PSC volume density as simulated with EMAC and the one derived from Envisat/MIPAS observations shows that the simulated PSC volume densities are several orders of magnitude smaller than the observed ones.

I also think that the demonstration that FinROSE is successful in modelling polar ozone loss is not sufficient. Very little information is given in this paper. Fig. 6 only give a summary (e.g. I cannot judge whether or not the recovery of active chlorine into ClONO2 is convincing), Other models have done comparisons with observations in details (see e.g., Wohltmann et al., 2017, and references therein). Here the reference to Karpetchko et al. (2013) is given, but in this study FinROSE is used with PSCs "switched off", so this paper is not valid to support the performance of the model for Arctic ozone loss in cold winters (see also below). In section two of the paper a description of the initialization of the model should be given. How is total chlorine (Cly) initialized? What ware the initial values for HCl, ClONO2, N2O5 etc – this information would be helpful to interpret the results of the paper. How good is ERA-I ozone in comparison to MLS ozone? How well does FinROSE simulate downward transport in the Arctic (compare to MLS N2O?).

One important driver of chemical ozone loss in cold Arctic winters is the stratospheric chlorine loading. I suggest making this point and briefly discuss the temporal development of stratospheric chlorine (e.g., Engel et al., 2018).

**Details**

- title: the title is too general, is sounds like *the uncertainty* in simulated Arctic ozone is due to water vapour. But there are more reasons for uncertainties

- abstract, l. 8: the point here is that there is a one-to-one correspondence between entry value and polar water vapour conditions – correct? This point could be made more clearly.

- abstract, l 13-15: I think this is an important finding of this paper.

- abstract, l 17: 2–7% more: this is a very important result of this paper. But can be anything worked out what the mechanism is? It would be important to state this finding in the abstract.

- p.2, l. 1: Temperature may show a warming in the troposphere but a significant cooling near the tropopause in connection with deep convection (Kim et al., 2018), so the connection between climate change and tropopause temperature may not as straightforward as suggested here. I suggest more discussion of this point which is important for this paper.

- p.2, l. 3: "affect chlorine partitioning" but how? Is this relevant for polar chemistry?

- p.2, l. 7: this citation is for the upper stratosphere; I do not think it is appropriate here.

- p.2, l. 12: One could also mention the observations discussed by Kuttippurath and Nair (2017) here.

- p. 2, l. 14: do you have references to more recent studies to back up this statement on recovery?

- p. 2, l. 17: remember that the focus here is on *Arctic* ozone.

- p. 3, l. 3: I do not think this statement is correct. Please consider the temperature dependence of the main polar ozone loss cycles (von Clarmann, 2013; Canty et al., 2016; Wohltmann et al., 2017). I you do not agree, please specify "some".

- p.3, l 17/18: this is more than "likely" if temperatures are high, there are no PSCs and thus there is no impact of water vapour on PSCs and heterogeneous reactions. Under these conditions NOx chemistry indeed is relevant (but we are missing a discussion here of the impact – if any – of water vapour on the NOx chemistry in question). In any even (see also other points in this review) these two cases must be clearly separated. You are looking at different processes here.

- p. 3, l 28: the idea of a "controlled" experiment is good! also change to "impact of . . . on ozone loss".

- p. 4., l 9: this is not an important point, but is has recently been argued that the reaction $CH_3O_2 + ClO$ is important for polar ozone loss (Müller et al., 2018); is this reaction taken into account in the chemical scheme used here?

- p. 4, l. 16/17: As it is very important for this paper: describe here how the dependence on water vapour of the heterogeneous reactions (i.e. the $\gamma$) is determined.

- p. 4, l. 19: As it is very important for chemistry simulations at the poles: describe here if(how) spherical geometry is properly taken into account in the photolysis code.

- p. 4, l. 29: the reference to Karpetchko et al. (2013) is important here as this is the only citation give to support the performance of FinROSE in simulation polar ozone loss. However, in Karpetchko et al. (2013) FinROSE is used with PSCs "switched off", so this paper is not valid to support the performance of the model for the issues discussed in this paper.

- p. 5, top: It is not clear to me what was done exactly here: what is "Interim(MAX)"? Why did you not simply shift the water vapour values up and down by some value preserving the variability?

- p 5, l 9: The comparison to MLS ozone would be important. It should be shown and discussed in the paper in detail. Further, MLS HCl (and possibly other measurements of Cl-species could be helpful to validate the model).

- p. 6, l 7: which reference theta level was used for mPV? How is 36 PVU chosen?

- p. 6, l. 23: Here (and perhaps elsewhere) the question arises if downward transport in the FinROSE model in the Arctic polar vortex is simulated appropriately. This issue has an impact of water vapour in the polar vortex. A comparison of simulated N2O with observations would be helpful here.

- p. 7., l 5: when dehydration occurs the initial water vapour and thus the Max/Min scenarios will not be relevant any more, as the polar water vapour is set to the equilibrium value over ice – correct? This issue should be discussed in the paper.

- p. 7, l 15: what is meant by NAT/STS volume, Just adding both PSC types? But the impact of increased water vapour on NAT and STS will be different. So why has this been done; I suggest separating NAT and STS.

- p. 7, l. 22: is there some impact of sedimentation on the duration of ice PSCs?

- p. 7, l. 23/24: Note that the Calipso PSC product has more recent information now (Spang et al., 2017; Pitts et al., 2018); I suggest using the most recent information. For example the estimated ice area might change.

- p. 8, l 4: "too dry models" this point sounds very speculative

- p 8, l. 25: One question that arises here how well the model simulates the size of the vortex – as this point might be relevant for assessing the PSC area.

- p 9, l 10: small twice

- p. 9, l 14: chlorine activation does not require PSCs, it starts on cold binary aerosols, but is also humidity dependent (e.g. Solomon, 1999; Wegner et al., 2012).

- p 9, l 25: which reservoir species?

- p 10, top: the start of activation is one thing, but not really what determines how much ozone loss happens in a particular winter.

- p 10, l 12: which process is responsible here?

- p 10., l 23: but why?

- p 10., l 26: I am not convinced that this statement is correct, is this really a cause and effect relation?

- p 10., l 30: I do not think that table 2 is a good summary of the chlorine activation simulation in the model run, too many important details are missing. (See also other comments in this review).

- p 11, sec 3.4: it would helpful to have more comparisons to ozone loss from simulations of other models.

- p 11, l 18: is this statement on NOx also true for FinROSE? If yes, what is the evidence from the model simulation for this statement?

- p 12, l 2: this might be true, but this is an example where speculation could be replaced by statements based on the actual simulations. Has a simulation been attempted with a better resolution?

- p 12., l 7: "without het. chemistry": it should be clearly stated what has been assumed regarding the heterogeneous reaction N2O5 + H2O, which is not temperature dependent but would be important here.

- p 12., l 13: this is not true for the het. reaction N2O5 + H2O

- p 12., l 16-18: this statement is really confusing: if I understand correctly, only 30 DU ozone loss is caused by NAT/STS/ICE PSCs in Arctic winter 2010/11: This is in contrast to statements elsewhere in the paper and also to literature and our general understanding of Arctic ozone loss. rent types of PSCs are for the results of this section.

- p. 14., section 4: My suggestion would be to not combine discussions and conclusions. Have a separate discussion section to focus on the relations of the results of this study with what is available in the relevant literature and a separate conclusion section to focus on the main conclusions of *this study*. But this is up to the authors.

- p 14., l 15: this is an example where the analysis of the paper could be more focused and more detailed. What means "larger"? It will be important to which altitude regions and how much towards the vortex edge the PSC area extends. If these details are analyzed more can be learned about the processes responsible for the model results.

- Fig. 6: In general I think that there is not enough information on chlorine species in this paper. Specifically: 2012/13: why is there a range in reservoir species in early winter in this case for Min/Max? Can this be explained/understood? In 2010/11 there seems to be full depletion of HCl. It is not obvious that this can be easily achieved in models. Is it understood how this happens in FinROSE (reformation of ClONO2 and HOCl is necessary for this). What is the role of HOCl in chlorine activation through heterogeneous reactions?

- Fig. 7: Is there really an established Arctic (!) vortex (not only small remainders of the vortex) for the entire period shown here? Until mid-April! Show a time series of the size of the vortex at least in the reply or in a supplement.

**References**

Canty, T. P., Salawitch, R. J., and Wilmouth, D. M.: The kinetics of the ClOOCl catalytic cycle, J. Geophys. Res., 121, 13 768–13 783, doi:10.1002/2016JD025710, 2016.

Drdla, K. and Müller, R.: Temperature thresholds for chlorine activation and ozone loss in the polar stratosphere, Ann. Geophys., 30, 1055–1073, doi:10.5194/angeo-30-1-2012, 2012.

Engel, A., Bönisch, H., Ostermöller, J., Chipperfield, M. P., Dhomse, S., and Jöckel, P.: A refined method for calculating equivalent effective stratospheric chlorine, Atmos. Chem. Phys., 18, 601–619, doi:10.5194/acp-18-601-2018, 2018.

Khosrawi, F., Kirner, O., Stiller, G., Höpfner, M., Santee, L., M., Kellmann, S., and Braesicke, P.: Comparison of ECHAM5/MESSy Atmospheric Chemistry (EMAC) Simulations of the Arctic winter 2009/2010 and 2010/2011 with Envisat/MIPAS and Aura/MLS Observations, Atmos. Chem. Phys. Discuss., in review, 2018.

Kim, J., Randel, W. J., and Birner, T.: Convectively Driven Tropopause-Level Cooling and Its Influences on Stratospheric Moisture, J. Geophys. Res., 123, 590–606, doi:10.1002/2017JD027080, 2018.

Konopka, P., Engel, A., Funke, B., Müller, R., Grooß, J.-U., Günther, G., Wetter, T., Stiller, G., von Clarmann, T., Glatthor, N., Oelhaf, H., Wetzel, G., López-Puertas, M., Pirre, M., Huret, N., and Riese, M.: Ozone loss driven by nitrogen oxides and triggered by stratospheric

warmings may outweigh the effect of halogens, J. Geophys. Res., 112, D05105, doi:10.1029/2006JD007064, 2007.

Kuttippurath, J. and Nair, P. J.: The signs of Antarctic ozone hole recovery, Sci. Rep., 7, 585, doi:10.1038/s41598-017-00722-7, 2017.

Müller, R., Grooß, J.-U., Zafar, A. M., and Lehmann, R.: The maintenance of elevated active chlorine levels in the Antarctic lower stratosphere through HCl null cycles, Atmos. Chem. Phys., 18, 2985–2997, doi:10.5194/acp-18-2985-2018, 2018.

Pitts, M. C., Poole, L. R., and Gonzalez, R.: Polar stratospheric cloud climatology based on CALIPSO spaceborne lidar measurements from 2006–2017, Atmos. Chem. Phys. Discuss., 2018, 1–54, doi:10.5194/acp-2018-234, https://www.atmos-chem-phys-discuss.net/acp-2018-234/, 2018.

Sagi, K., Pérot, K., Murtagh, D., and Orsolini, Y.: Two mechanisms of stratospheric ozone loss in the Northern Hemisphere, studied using data assimilation of Odin/SMR atmospheric observations, Atmos. Chem. Phys., 17, 1791–1803, doi:10.5194/acp-17-1791-2017, https://www.atmos-chem-phys.net/17/1791/2017/, 2017.

Shi, Q., Jayne, J. T., Kolb, C. E., Worsnop, D. R., and Davidovits, P.: Kinetic model for reaction of $ClONO_2$ with $H_2O$ and HCl and HOCl with HCl in sulfuric acid solutions, J. Geophys. Res., 106, 24 259–24 274, doi:10.1029/2000JD000181, 2001.

Solomon, S.: Stratospheric ozone depletion: A review of concepts and history, Rev. Geophys., 37, 275–316, doi:10.1029/1999RG900008, 1999.

Spang, R., Hoffmann, L., Müller, R., Grooß, J.-U., Tritscher, I., Höpfner, M., Pitts, M., Orr, A., and Riese, M.: A climatology of polar stratospheric cloud composition between 2002 and 2012 based on MIPAS/Envisat observations, Atmos. Chem. Phys. Discuss., 2017, 1–44, doi:10.5194/acp-2017-898, https://www.atmos-chem-phys-discuss.net/acp-2017-898/, 2017.

von Clarmann, T.: Chlorine in the stratosphere, Atmósfera, 26, 415–458, 2013.

Wegner, T., Grooß, J.-U., von Hobe, M., Stroh, F., Sumińska-Ebersoldt, O., Volk, C. M., Hösen, E., Mitev, V., Shur, G., and Müller, R.: Heterogeneous chlorine activation on stratospheric aerosols and clouds in the Arctic polar vortex, Atmos. Chem. Phys., 12, 11 095–11 106, doi:10.5194/acp-12-11095-2012, 2012.

Wohltmann, I., Lehmann, R., and Rex, M.: A quantitative analysis of the reactions involved in stratospheric ozone depletion in the polar vortex core, Atmos. Chem. Phys., 17, 10 535–10 563, doi:10.5194/acp-17-10535-2017, https://www.atmos-chem-phys.net/17/10535/2017/, 2017.

---

## Author Comment (AC1) · 22 Aug 2018

The topic of this paper is Arctic ozone loss in general and the sensitivity of Arctic ozone loss on the polar water vapour concentrations in particular. If the authors agree, they need to to remove all the discussion of mid-latitude ozone loss and the related cycles. Further, discussion/citation of upper stratospheric ozone loss is not helpful.

The topic of the paper (water vapour sensitivity) is of importance and the the approach using a CTM to focus on the impact of water vapour on PSCs on heterogeneous chemistry is good. On the other hand, the results on the sensitivity will gain on impact if the representation of PSCs in the model, the simulation of chlorine activation and ozone loss are clearly demonstrated in the paper. I do not think that this is the case in the present version of the paper (the cited papers are not sufficient, see below).

For example, the impact of humidity on heterogeneous chemistry should be different for different types of PSCs. If in this model the PSC types are dominated by a different type that in reality (or even in different models) carrying over the information on sensitivity deduced here can be problematic. There is some comparison with ERA-interim, but this is not the real world. Also there i.e. very little information on chlorine chemistry (except in Fig.6, where there is no distinction between HCl and ClONO2 for example, see also below).

The paper addresses a second case, namely a warm and dynamically unstable winter in the Arctic stratosphere. In such a case, in contrast to the halogen induced ozone loss in cold Arctic winters, there is little ozone loss below 500 K and ozone loss in the middle stratosphere (NOx-induced destruction) becomes important (e.g., Konopka et al., 2007; Sagi et al., 2017). However, this is a completely different chemical mechanism, which will have a very different dependence on water vapour (clearly no impact of heterogeneous chemistry). Therefore this second case needs to be very clearly discriminated from the "halogen" case throughout the paper.

In summary, I recommend focusing the paper and a providing a better basis and justification for the work presented. I also suggest a better balance of the cited literature (as stated in the quick review, I am not suggesting to cite all the references mentioned here). I am sorry for the many critical remarks but I believe that a revised paper taking into account these comments would be much stronger than the present version.

General

The discussion of the HOx chemistry in mid-latitudes and in the upper stratosphere (and the associated references) in the introduction is confusing. In the Arctic, in cold winters, an increase in water vapour is expected to enhance ozone loss (assuming that a substantial amount of stratospheric chlorine is present) by enhancing heterogeneous reactions (Shi et al., 2001; Drdla and Müller, 2012). The situation can actually be more complicated as there is no simple argument that "more PSCs" means "more ozone loss". This is also a finding of this paper, if one considers the case of the cold Arctic winter 2010/11. The latter point should be more emphasized in the paper I suggest.

Recently, climatologies of PSC occurrence have become available from observations (Spang et al., 2017; Pitts et al., 2018). The question how important the discrimination between different PSC types in a model is for a successful simulation of polar ozone loss is not trivial. This paper could contribute substantially to this issue; perhaps more than visible in the present version of this paper. This aspect could be very relevant for the discussion of the sensitivities on water vapour (which will be different for different types of PSCS). Finally, models might misrepresent PSC volume density compared to observations; Khosrawi et al. (2018) report that the comparison between the PSC volume density as simulated with EMAC and the one derived from Envisat/MIPAS observations

shows that the simulated PSC volume densities are several orders of magnitude smaller than the observed ones.

I also think that the demonstration that FinROSE is successful in modelling polar ozone loss is not sufficient. Very little information is given in this paper. Fig. 6 only give a summary (e.g. I cannot judge whether or not the recovery of active chlorine into ClONO2 is convincing), Other models have done comparisons with observations in details (see e.g., Wohltmann et al., 2017, and references therein). Here the reference to Karpetchko et al. (2013) is given, but in this study FinROSE is used with PSCs "switched off", so this paper is not valid to support the performance of the model for Arctic ozone loss in cold winters (see also below). In section two of the paper a description of the initialization of the model should be given. How is total chlorine (Cly) initialized? What ware the initial values for HCl, ClONO2, N2O5 etc – this information would be helpful to interpret the results of the paper. How good is ERA-I ozone in comparison to MLS ozone? How well does FinROSE simulate downward transport in the Arctic (compare to MLS N2O?).

One important driver of chemical ozone loss in cold Arctic winters is the stratospheric chlorine loading. I suggest making this point and briefly discuss the temporal development of stratospheric chlorine (e.g., Engel et al., 2018).

We thank the Referee #1 for the thorough and valuable comments. We think that the issues raised here in the introduction has been addressed in the detailed comments below, except for the discussion related to the temporal evolution of stratospheric chlorine. A discussion was added to the Introduction in the manuscript and a reference to Engel et al. (2018) was added. In addition, the references were checked and corrected and new references were added.

The discussion of the ozone loss results was changed in order to more clearly discriminate the loss caused by NOx from the loss related to halogens.

The PSC volume was changed to PSC area, which dind't change the results and conclusions, but the text was altered and some of the figures were replotted.

The definition of the start and end of the polar vortex season was defined in more detail, which changed the appearance of some of the figures, but did not change the results or conclusions.

The validation of the FinROSE model has been done earlier, therefore we didn't include validation in this study, e.g. in Karpechko et al. (2013) we have compared the full chemistry simulation to satellite observations. The ERA-Inteim ozone have been validated by e.g. Dragani (2011).

Details

• title: the title is too general, is sounds like the uncertainty in simulated Arctic ozone is due to water vapour. But there are more reasons for uncertainties
We changed the title, it now reads:

"Linking the Uncertainty of Simulated Arctic Ozone Losses to Modelling Uncertainties in the Tropical Stratospheric Water Vapour "

• abstract, l. 8: the point here is that there is a one-to-one correspondence between entry value and polar water vapour conditions – correct? This point could be made more clearly.

The change in water vapour in the tropical tropopause region was seen nearly as a on-to-one correspondence in the Arctic polar vortex. We highlighted this finding in the text.

• abstract, l 13-15: I think this is an important finding of this paper.
We further streamlined the text tio highlight this finding.

"If the cold conditions persist long enough (e.g. as in 2010/11), the chlorine activation is nearly complete. In this case addition of water vapour to the stratosphere does not increase the chlorine activation and ozone destruction significantly."

• abstract, l 17: 2–7% more: this is a very important result of this paper. But can be anything worked out what the mechanism is? It would be important to state this finding in the abstract.
We made some changes to the changed the Abstract, related also the the previous comment:

"We found that the impact of water vapour changes on ozone loss in the Arctic polar vortex depends on the meteorological conditions. The strongest effect was in intermediately cold conditions, such as 2013/14, when added water vapour resulted in 2-7% more ozone loss due to the additional polar stratospheric clouds (PSC) and associated chlorine activation on their surface, leading to ozone loss. The effect was less pronounced in cold winters such as 2010/11 because cold conditions persisted long enough for a nearly complete chlorine activation even in simulations with observed water vapour. In this case addition of water vapour to the stratosphere led to increased area of ice PSC but it could not increase the chlorine activation and ozone destruction significantly. In the warm winter 2012/13 the impact of water vapour concentration on ozone loss was small, because the ozone loss was mainly NOx induced."

• p.2, l. 1: Temperature may show a warming in the troposphere but a significant cooling near the tropopause in connection with deep convection (Kim et al., 2018), so the connection between climate change and tropopause temperature may not as straightforward as suggested here. I suggest more discussion of this point which is important for this paper.
We specified the text and added the reference:

"A warmer climate in the troposphere is suggested to increase stratospheric water vapour (SWV) through increases in the water vapour entering through the tropopause, which would further warm the troposphere below (Dessler et al., 2013). However tropospheric warming may also lead to a significant cooling near the tropopause in connection with deep convection (Kim et al., 2018), so that the link between warming climate and tropopause temperature is not straightforward."

• p.2, l. 3: "affect chlorine partitioning" but how? Is this relevant for polar chemistry?
We removed some of the discussion that was not relevant to the polar chemistry.

• p.2, l. 7: this citation is for the upper stratosphere; I do not think it is appropriate here.
The citations were checked and changed (Solomon 1999 and Khosrawi 2016)

 p.2, l. 12: One could also mention the observations discussed by Kuttippurath and Nair (2017) here.
The reference and discussion was added.

• p. 2, l. 14: do you have references to more recent studies to back up this statement on recovery?
We added reference to a recent study of Dhomse 2018, and changed the recovery dates according to their results. Also references to WMO 2014, and Morgernstren 2017 were added.

• p. 2, l. 17: remember that the focus here is on Arctic ozone.
References to Smalley 2017 and Rosenlof 2001 were removed from here.

• p. 3, l. 3: I do not think this statement is correct. Please consider the temperature dependence of the main polar ozone loss cycles (von Clarmann, 2013; Canty et al., 2016; Wohltmann et al., 2017). I you do not agree, please specify "some".

We think the text is correct in general, but it is true that the offset would probably come from slower ozone loss reactions in areas outside the vortex, which could contribute to a so called super recovery of the ozone layer. We changed the text accordingly.

• p.3, l 17/18: this is more than "likely" if temperatures are high, there are no PSCs and thus there is no impact of water vapour on PSCs and heterogeneous reactions. Under these conditions NOx chemistry indeed is relevant (but we are missing a discussion here of the impact – if any – of water vapour on the NOx chemistry in question). In any even (see also other points in this review) these two cases must be clearly separated. You are looking at different processes here.

We modified the text to more clearly separate the discussion of the NOx and chlorine caused ozone loss. We were not able to attribute any change in NOx chemistry to changes in water vapour.

• p. 3, l 28: the idea of a "controlled" experiment is good! also change to "impact of . . . on ozone loss".

Sentence was changed to:

"Therefore a more controlled experiment is needed in order to assess impact of these SWV changes on ozone loss."

• p. 4., l 9: this is not an important point, but is has recently been argued that the reaction CH3O2 + ClO is important for polar ozone loss (Müller et al., 2018); is this reaction taken into account in the chemical scheme used here?

CH3O2 is not included in our chemistry scheme, and therefore we are unfortunately not able to discuss the importance of the CH3O2 + ClO reaction in the Arctic polar vortex.

• p. 4, l. 16/17: As it is very important for this paper: describe here how the dependence on water vapour of the heterogeneous reactions (i.e. the γ) is determined.

The following text was added to the description of the chemistry scheme.

"The reaction rates on NAT and ICE PSCs are not directly affected by the water vapour concentration except through the available surface area, i.e. the uptake coefficients are constant. In the case of binary aerosols and STS PSCs the uptake coefficients of some reactions depend on the composition of the droplets, i.e the hydrolysis reactions of ClONO2, BrONO2 and N2O5, as well as the reaction of HCl between ClONO2 and HOCl (Sander et al., 2011)."

• p. 4, l. 19: As it is very important for chemistry simulations at the poles: describe here if(how) spherical geometry is properly taken into account in the photolysis code.

We changed the text and added references to Stamnes (1988) and Dahlback and Stamnes, (1991) where the pseudo-spherical approximation is discussed.

"Look-up-tables of photodissociation coefficients were pre-calculated using the PHODIS model (Kylling et al., 1995). Within PHODIS the radiative transfer equation is solved by the discrete ordinate algorithm (Stamnes et al.,1988). This algorithm has been modified to account for the spherical shape of the atmosphere using the pseudo-spherical approximation (Dahlback and Stamnes, 1991)."

• p. 4, l. 29: the reference to Karpetchko et al. (2013) is important here as this is the only citation give to support the performance of FinROSE in simulation polar ozone loss. However, in

Karpetchko et al. (2013) FinROSE is used with PSCs "switched off", so this paper is not valid to support the performance of the model for the issues discussed in this paper.
Please note that Karpechko et al have both simulations, with and without heterogeneous chemistry.

• p. 5, top: It is not clear to me what was done exactly here: what is "Interim(MAX)"? Why did you not simply shift the water vapour values up and down by some value preserving the variability?
Please note that the variability is not preserved across CCMs either and our approach was chosen to replicate simulations by CCM. In our approach we replicate changes in amplitude as well as shift in mean water vapour. The description was improved.

"The simulations differed from each other by the prescribed water vapour concentration in the tropical tropopause region (stratosphere between 21S–21N, below 80 hPa), where it was prescribed as follows: (1) water vapour taken from ERA-Interim (Interim simulation), (2) increased water vapour (Max simulation), and (3) decreased water vapour (Min simulation). Specifically, the SWV lower boundary conditions for Min and Max simulations were obtained by multiplying values from ERA-Interim between tropopause and 80 hPa, and between 21 S–21 N by monthly coefficients ranging between 1.46-1.7 (Max) and between 0.5-0.63 (min), so that they approximately correspond to the driest and wettest CCM, as determined by SWV values at the tropical tropopause, across models analyzed by Gettleman et al. (2010)."

• p 5, l 9: The comparison to MLS ozone would be important. It should be shown and discussed in the paper in detail. Further, MLS HCl (and possibly other measurements of Cl-species could be helpful to validate the model).
Please note that validation of FinROSE-ctm have been done earlier (Damski et al 2007, and Thölix et al. 2016) and therefore it was not included in this paper. In the revised version we provide references to our earlier studies where such validation was done.

• p. 6, l 7: which reference theta level was used for mPV? How is 36 PVU chosen?
The reference theta level is 475K. The limit 36 PVU was chosen based on publications by Rex at al. (1999) and Streibel et al. (2006), who have shown that 36 PVU is a good approximation for the vortex edge. The text was modified as follows:

"The polar vortex was identified using the modified potential vorticity (mPV) (Lait, 1994), with the 475 K potential temperature as reference level. Here the polar vortex is defined as the area enclosed by the 36 PVU isoline separately for every model level. The 36 PVU contour approximately correspond to the region of maximum PV gradient, i.e the polar vortex edge (Rex et al, 1999, Streibel et al, 2006)."

• p. 6, l. 23: Here (and perhaps elsewhere) the question arises if downward transport in the FinROSE model in the Arctic polar vortex is simulated appropriately. This issue has an impact of water vapour in the polar vortex. A comparison of simulated N2O with observations would be helpful here.
Please note that validation of FinROSE-ctm have been done earlier (Damski et al 2007, and Thölix et al. 2016) and therefore it was not included in this paper. In the revised version we provide references to our earlier studies where such validation was done.

• p. 7., l 5: when dehydration occurs the initial water vapour and thus the Max/Min scenarios will not be relevant any more, as the polar water vapour is set to the equilibrium value over ice – correct? This issue should be discussed in the paper.
The dehydration is calculated based on the settling velocity of ICE PSC particles. In the Arctic the dehydration is typically never complete, at least not on large areas. The not enough time for all

particles to sediment out of a layer at a given grid point during the time the ICE PSCs persist, and the area cold enough for formation of ICE PSCs is typically small compared to the Arctic polar vortex. Therefore the differences in water vapour concentration persists also after dehydration has occurred. The following text was added to the Modelling and data section:

"The sedimentation of PSC particles, which can lead to denitrification and dehydration, is calculated based on the settling velocity, which takes into account the PSC particle size."

• p. 7, l 15: what is meant by NAT/STS volume, Just adding both PSC types? But the impact of increased water vapour on NAT and STS will be different. So why has this been done; I suggest separating NAT and STS.
Yes, the volume of the grid points containing either NAT or STS (or both) were summed. To better separate the impact of water vapour on the heterogeneous chemistry we now separated the NAT and STS volume. We now also use area instead of volume. The text was altered accordingly.

• p. 7, l. 22: is there some impact of sedimentation on the duration of ice PSCs?
Yes, the dehydration deepens with time. The longer the ICE PSCs persist the more of the particles have time to settle out of a given layer. See comment above related to dehydration.

• p. 7, l. 23/24: Note that the Calipso PSC product has more recent information now (Spang et al., 2017; Pitts et al., 2018); I suggest using the most recent information. For example the estimated ice area might change.
References to Spang et al., (2018) and Pitts et al., 2018 were added. The PSC volumes have changed to PSC areas at 20 km level, because of easier comparison to observations. In general the timing of PSC in our model is good, but our ICE area is somewhat overestimated compared to Calipso data. The text was modified according the new figures.

• p. 8, l 4: "too dry models" this point sounds very speculative
The sentence "For example too dry models may not be able to simulate a large Arctic ozone loss such as of 2010/11." was removed from the results section and the discussion was moved to the Discussion and conclusions section.

 p 8, l. 25: One question that arises here how well the model simulates the size of the vortex – as this point might be relevant for assessing the PSC area.
The polar vortex is calculated using ERA-Interim data, and not altered by the model. The vortex limit has been discussed in the reply to question of p. 6, l 7.

• p 9, l 10: small twice
Corrected.

• p. 9, l 14: chlorine activation does not require PSCs, it starts on cold binary aerosols, but is also humidity dependent (e.g. Solomon, 1999; Wegner et al., 2012).
It is true that the activation starts on binary aerosols. However, there is no hard limit between binary aerosols and PSCs. In FinROSE the switchover from binary aerosols to STS is done at 215 K. At that temperature the STS is practically a binary aerosol, the large influence from dissolved water and HNO3 happens at even lower temperatures. The humidity is taken into account through the composition and uptake coefficient, as described in a previous comment (p. 4, l. 16/17).  The text was modified as follows:

"In the cold conditions within the polar vortex the chlorine species are transformed, through heterogeneous reactions, into intermediate species such as Cl2."

• p 9, l 25: which reservoir species?
In early March mostly ClONO2 is formed and a few weeks later HCl. This was added to the text.

• p 10, top: the start of activation is one thing, but not really what determines how much ozone loss happens in a particular winter.
The purpose here is only to describe the evolution of active chlorine for different winters.

• p 10, l 12: which process is responsible here?
The reason is probably heterogeneous reactions that occur in binary aerosols and STS, the added water vapour increases the uptake coefficients and to some extent also the surface area. The text was modified as follows:

"The water vapour concentration seem to strongly affect the transformation of chlorine from the reservoir species to the intermediate ones in the beginning of Arctic winter. The fractions of intermediate and reservoir chlorine species change significantly with water vapour concentration in November and December, when STS PSCs start to form. The water vapour concentration affects the composition of binary aerosols an STS. When more water condenses to the particles the uptake coefficients for the heterogeneous reactions increase, i.e. the reaction rate increases."

• p 10., l 23: but why?
During the coldest period the difference in chlorine activation is small probably due to the ICE PSCs that appear in all simulations. The conditions were favourable for high chlorine activation in all simulations and therefore the differences were small. A clarification was added to the text.

• p 10., l 26: I am not convinced that this statement is correct, is this really a cause and effect relation?
We agree with the referee in the sense that the statement "The amount of chlorine activation correlate with the volume of NAT/STS PSCs." is probably too broad. The correlation is more complex, we altered the text to reflect this. We now deal with the STS and NAT separately, as suggested earlier.

• p 10., l 30: I do not think that table 2 is a good summary of the chlorine activation simulation in the model run, too many important details are missing. (See also other comments in this review).
We added monthly sums to the table, and now the timing of the chlorine activation in different winters can be seen from the table. The text modified according the new table values.

• p 11, sec 3.4: it would helpful to have more comparisons to ozone loss from simulations of other models.
We were not able to find additional references than Vogel (2011), Sinnhuber (2011) ja Pommereau et al. (2013), related to the effect of water vapour on Arctic ozone depletion.

• p 11, l 18: is this statement on NOx also true for FinROSE? If yes, what is the evidence from the model simulation for this statement?
The text was rearranged, and the NOx-part was moved to the discussion of the ozone profile figure (Fig 8). It can be seen from the no-hetero run that the ozone loss in winter 2012/13 is gas phase chemistry driven. The text now reads:

"In winters when the polar vortex is unstable and small or disturbed the Brewer–Dobson circulation brings more NO x -rich air to the polar vortex than usual. Hence the ozone loss in the 2012/13 winter was produced mostly by NO x chemistry as shown previously by e.g., Sagi et al. (2017), and can be seen from FinROSE results by comparing the simulations with and without the heterogeneous chemistry."

• p 12, l 2: this might be true, but this is an example where speculation could be replaced by statements based on the actual simulations. Has a simulation been attempted with a better resolution?

We did an additional experiment and it confirmed our assumption. The ozone loss in 2010/11 with higher resolution (1.5 x 3) were about 10 DU larger than with FinROSE's resolution. Unfortunately, redoing all experiments with increased resolution are currently not feasible for technical reasons, therefore we report in the manuscript the results based on the original resolution.

• p 12., l 7: "without het. chemistry": it should be clearly stated what has been assumed regarding the heterogeneous reaction N2O5 + H2O, which is not temperature dependent but would be important here.

No separate assumptions have been made for the N2O5 + H2O reaction. There were two different setups used, one where no heterogeneous reactions were included, and another one where the temperature for the aerosol/PSC scheme was limited to 200K. The one with the 200K limit allows some heterogeneous processing on binary aerosols and some STS that are very dilute in HNO3. The N2O5 + H2O reaction is practically not temperature dependent, but the rate will be affected by the smaller surface area available. The possible effect on e.g. NOx induced loss was not considered here. We altered the description of the setup for the no-hetero simulations.

• p 12., l 13: this is not true for the het. reaction N2O5 + H2O

The idea to limit the temperature to 200K was to allow reactions on binary aerosols, but not on PSCs. It is true that a small amount of STS can form, but due to the temperature limit the surface area densities will remain quite small.

• p 12., l 16-18: this statement is really confusing: if I understand correctly, only 30 DU ozone loss is caused by NAT/STS/ICE PSCs in Arctic winter 2010/11: This is in contrast to statements elsewhere in the paper and also to literature and our general understanding of Arctic ozone loss. rent types of PSCs are for the results of this section.

We believe that the result shows that the binary aerosols and STS have an effect. There are studies that suggest that binary aerosols are more important for chlorine activation than PSCs (e.g. Drdla and Müller,2012; Wohltmann et al., 2013; Kirner et al., 2015). However, the contribution of PSCs are also important (e.g. Wegner et al., 2016). We added discussion on these results.

• p. 14., section 4: My suggestion would be to not combine discussions and conclusions. Have a separate discussion section to focus on the relations of the results of this study with what is available in the relevant literature and a separate conclusion section to focus on the main conclusions of this study. But this is up to the authors.

We did consider this suggestion, but decided not to change the structure.

• p 14., l 15: this is an example where the analysis of the paper could be more focused and more detailed. What means "larger"? It will be important to which altitude regions and how much towards the vortex edge the PSC area extends. If these details are analyzed more can be learned about the processes responsible for the model results.

The main finding here was that if the winter is cold, the PSCs may form even at low water vapour concentrations, therefore the water vapour increase is less important.

• Fig. 6: In general I think that there is not enough information on chlorine species in this paper. Specifically:
2012/13: why is there a range in reservoir species in early winter in this case for Min/Max? Can this be explained/understood?

The range can at least partly be explained by differences in the PSCs, i.e. temperature. We added discussion of the early chlorine partitioning.

• Fig. 7: Is there really an established Arctic (!) vortex (not only small remainders of the vortex) for the entire period shown here? Until mid-April! Show a time series of the size of the vortex at least in the reply or in a supplement.

We believe that in cold winters a well developed vortex can last until mid-April and, in extreme cases such as 1990 and 1997, potentially even until early May. See for example study by Waugh et al. (1999). In the revised version we consider vortex to be established when its area (defined by 36 PVU contour) exceeds that corresponding to 80° equivalent latitude, following Waugh et al. (1999). This new definition results in only minor changes with respect to our original version. The size of the vortex can be seen in Fig 1.

[Figure]

Figure 1. Vortex area (Milion sq meters).

References

Canty, T. P., Salawitch, R. J., and Wilmouth, D. M.: The kinetics of the ClOOCl catalytic cycle, J. Geophys. Res., 121, 13 768–13 783, doi:10.1002/2016JD025710, 2016.

Dahlback, A. and Stamnes, K.: A new spherical model for computing the radiation field available for photolysis and heating at twilight, Planet. Space Sci., 39, 671–683, 1991.

Damski, J., Thölix, L., Backman, L., Kaurola, J., Taalas, P., Austin, J., Butchart, N., Kulmala, M.: A chemistry-transport model simulation of middle atmospheric ozone from 1980 to 2019 using coupled chemistry GCM winds and temperatures, Atmos. Chem. Phys., 7, 2165–2181, 2007.

Dhomse, S. S., Kinnison, D., Chipperfield, M. P., Salawitch, R. J., Cionni, I., Hegglin, M. I., Abraham, N. L., Akiyoshi, H., Archibald, A. T., Bednarz, E. M., Bekki, S., Braesicke, P., Butchart, N., Dameris, M., Deushi, M., Frith, S., Hardiman, S. C., Hassler, B., Horowitz, L. W., Hu, R.-M., Jöckel, P., Josse, B., Kirner, O., Kremser, S., Langematz, U., Lewis, J., Marchand, M., Lin, M., Mancini, E., Marécal, V., Michou, M., Morgenstern, O., O'Connor, F. M., Oman, L., Pitari, G., Plummer, D. A., Pyle, J. A., Revell, L. E., Rozanov, E., Schofield, R., Stenke, A., Stone, K., Sudo, K., Tilmes, S., Visioni, D., Yamashita, Y., and Zeng, G.: Estimates of ozone return dates from Chemistry-Climate Model Initiative simulations, Atmos. Chem. Phys., 18, 8409-8438, https://doi.org/10.5194/acp-18-8409-2018, 2018.

Dragani, R.: On the quality of the ERA-Interim ozone reanalyses: comparisons with satellite data, Q. J. Roy. Meteor. Soc., 137, 1312–1326, https://doi.org/10.1002/qj.821, 2011.

Drdla, K. and Müller, R.: Temperature thresholds for chlorine activation and ozone loss in the polar stratosphere, Ann. Geophys., 30, 1055–1073, doi:10.5194/angeo-30-1-2012, 2012.

Engel, A., Bönisch, H., Ostermöller, J., Chipperfield, M. P., Dhomse, S., and Jöckel, P.: A refined method for calculating equivalent effective stratospheric chlorine, Atmos. Chem. Phys., 18, 601–619, doi:10.5194/acp-18-601-2018, 2018.

Karpechko, A. Yu., et al. (2013), The link between springtime total ozone and summer UV radiation in Northern Hemisphere extratropics, J. Geophys. Res. Atmos. 118, 8649–8661, doi:10.1002/jgrd.50601.

Khosrawi, F., Kirner, O., Stiller, G., Höpfner, M., Santee, M. L., Kellmann, S., and Braesicke, P.: Comparison of ECHAM5/MESSy Atmospheric Chemistry (EMAC) simulations of the Arctic winter 2009/2010 and 2010/2011 with Envisat/MIPAS and Aura/MLS observations, Atmos. Chem. Phys., 18, 8873-8892, https://doi.org/10.5194/acp-18-8873-2018, 2018.

Kim, J., Randel, W. J., and Birner, T.: Convectively Driven Tropopause-Level Cooling and Its Influences on Stratospheric Moisture, J. Geophys. Res., 123, 590–606, doi:10.1002/2017JD027080, 2018.

Kirner, O., Müller, R., Ruhnke, R., and Fischer, H.: Contribution of liquid, NAT and ice particles to chlorine activation and ozone depletion in Antarctic winter and spring, Atmos. Chem. Phys., 15, 2019–2030, doi:10.5194/acp-15-2019-2015, 2015.

Konopka, P., Engel, A., Funke, B., Müller, R., Grooß, J.-U., Günther, G., Wetter, T., Stiller, G., von Clarmann, T., Glatthor, N., Oelhaf, H., Wetzel, G., López-Puertas, M., Pirre, M., Huret, N., and Riese, M.: Ozone loss driven by nitrogen oxides and triggered by stratospheric warmings may

outweigh the effect of halogens, J. Geophys. Res., 112, D05105, doi:10.1029/ 2006JD007064, 2007.

Kuttippurath, J. and Nair, P. J.: The signs of Antarctic ozone hole recovery, Sci. Rep., 7, 585, doi:10.1038/s41598-017-00722-7, 2017.

Morgenstern, O., Hegglin, M. I., Rozanov, E., O'Connor, F. M., Abraham, N. L., Akiyoshi, H., Archibald, A. T., Bekki, S., Butchart, N., Chipperfield, M. P., Deushi, M., Dhomse, S. S., Garcia, R. R., Hardiman, S. C., Horowitz, L. W., Jöckel, P., Josse, B., Kinnison, D., Lin, M., Mancini, E., Manyin, M. E., Marchand, M., Marécal, V., Michou, M., Oman, L. D., Pitari, G., Plummer, D. A., Revell, L. E., Saint-Martin, D., Schofield, R., Stenke, A., Stone, K., Sudo, K., Tanaka, T. Y., Tilmes, S., Yamashita, Y., Yoshida, K., and Zeng, G.: Review of the global models used within phase 1 of the Chemistry–Climate Model Initiative (CCMI), Geosci. Model Dev., 10, 639-671, doi:10.5194/gmd-10-639-2017, 2017.

Müller, R., Grooß, J.-U., Zafar, A. M., and Lehmann, R.: The maintenance of elevated active chlorine levels in the Antarctic lower stratosphere through HCl null cycles, Atmos. Chem. Phys., 18, 2985–2997, doi:10.5194/acp-18-2985-2018, 2018.

Pitts, M. C., Poole, L. R., and Gonzalez, R.: Polar stratospheric cloud climatology based on CALIPSO spaceborne lidar measurements from 2006 to 2017, Atmos. Chem. Phys., 18, 10881-10913, doi:10.5194/acp-18-10881-2018, 2018.

Rex, M., Von Der Gathen, P., Braathen, G. et al. Journal of Atmospheric Chemistry, 32: 35. https://doi.org/10.1023/A:1006093826861, 1999.

Sagi, K., Pérot, K., Murtagh, D., and Orsolini, Y.: Two mechanisms of stratospheric ozone loss in the Northern Hemisphere, studied using data assimilation of Odin/SMR atmospheric observations, Atmos. Chem. Phys., 17, 1791–1803, doi:10.5194/acp-17-1791-2017, https://www.atmos-chem-phys.net/17/1791/2017/, 2017.

Shi, Q., Jayne, J. T., Kolb, C. E., Worsnop, D. R., and Davidovits, P.: Kinetic model for reaction of ClONO 2 with H 2 O and HCl and HOCl with HCl in sulfuric acid solutions, J. Geophys. Res., 106, 24 259–24 274, doi:10.1029/2000JD000181, 2001.

Solomon, S.: Stratospheric ozone depletion: A review of concepts and history, Rev. Geophys., 37, 275–316, doi:10.1029/1999RG900008, 1999.

Spang, R., Hoffmann, L., Müller, R., Grooß, J.-U., Tritscher, I., Höpfner, M., Pitts, M., Orr, A., and Riese, M.: A climatology of polar stratospheric cloud composition between 2002 and 2012 based on MIPAS/Envisat observations, Atmos. Chem. Phys., 18, 5089-5113, https://doi.org/10.5194/acp-18-5089-2018, 2018.

Stamnes, K., Tsay, S.-C., Wiscombe, W., and Jayaweera, K.: Numerically stable algorithm for discrete-ordinate-method radiative transfer in multiple scattering and emitting layered media, Appl. Opt., 27, 2502–2509, 1988.

Streibel et al. Chemical ozone loss in the Arctic winter 2002/2003 determined with Match, Atmos. Chem. Phys., 6, 2783–2792, 2006.

Thölix, L., Backman, L., Kivi, R., and Karpechko, A. Yu.: Variability of water vapour in the Arctic stratosphere, Atmos. Chem. Phys., 16, 4307–4321, doi:10.5194/acp-16-4307-2016, 2016.

von Clarmann, T.: Chlorine in the stratosphere, Atmósfera, 26, 415–458, 2013.

Waugh, D. W., William, J. R., Steven, P., Paul, A. N., & Eric, R. N. (1999). Persistence of the lower stratospheric polar vortices. JGR, 104(D22), 27191–27201.

Wegner, T., Pitts, M. C., Poole, L. R., Tritscher, I., Grooß, J.-U., and Nakajima, H.: Vortex-wide chlorine activation by a mesoscale PSC event in the Arctic winter of 2009/10, Atmos. Chem. Phys., 16, 4569-4577, https://doi.org/10.5194/acp-16-4569-2016, 2016.

Wegner, T., Grooß, J.-U., von Hobe, M., Stroh, F., Sumińska-Ebersoldt, O., Volk, C. M., Hösen, E., Mitev, V., Shur, G., and Müller, R.: Heterogeneous chlorine activation on stratospheric aerosols and clouds in the Arctic polar vortex, Atmos. Chem. Phys., 12, 11 095–11 106, doi:10.5194/acp-12-11095-2012, 2012.

WMO (World Meteorological Organization), Scientific Assessment of Ozone Depletion: 2014, World Meteorological Organization, Global Ozone Research and Monitoring Project-Report No. 55, 416 pp., Geneva, Switzerland, 2014.

Wohltmann, I., Wegner, T., Müller, R., Lehmann, R., Rex, M., Manney, G. L., Santee, M. L., Bernath, P., Suminska-Ebersoldt, O., Stroh, F., von Hobe, M., Volk, C. M., Hösen, E., Ravegnani, F., Ulanovsky, A., and Yushkov, V.: Uncertainties in modelling heterogeneous chemistry and Arctic ozone depletion in the winter 2009/2010, Atmos. Chem. Phys., 13, 3909–3929, doi:10.5194/acp-13-3909-2013, 2013.

Wohltmann, I., Lehmann, R., and Rex, M.: A quantitative analysis of the reactions involved in stratospheric ozone depletion in the polar vortex core, Atmos. Chem. Phys., 17, 10 535–10 563, doi:10.5194/acp-17-10535-2017, https://www.atmos-chem-phys.net/17/10535/2017/, 2017.

Anonymous Referee #2

General
The authors investigate the sensitivity of modelled Arctic ozone loss to the water vapour mixing ratio entering the tropical stratosphere in a chemical transport model. They guide the reader well step by step through the causal chain water vapour concentration → PSC volume → chlorine activation → ozone loss. The authors clearly state that the investigated question is different from investigating the effect of water vapour changes due to climate change (which would occur on such a timescale that also the concentration of chlorine- and bromine-containing species changes considerably). They also clearly state that they investigate only one aspect of the above-mentioned question, namely the effect of water vapour on the surface area density of Polar Stratospheric Clouds (PSCs), negelecting the (probably stronger) effect of water vapour changes on ozone loss via changes of stratospheric temperature.

Comments
• An increase of water vapour may enhance heterogeneous chemistry by enlarging the air volume in which PSCs are formed (shown in Figs. 4 and 5) or via enlarging the surface area of existing particles (not shown). The authors seem to assume that the first effect is the dominant one. A discussion of this topic would help to complete the logic of the paper.
It is true that there also the available surface area and the composition of binary aerosols and PSCs can affect the heterogeneous chemistry, in addition to just the existence of PSCs. We therefore

1) improved the description of the chemitry scheme

"The reaction rates on NAT and ICE PSCs are not directly affected by the water vapour concentration except through the available surface area, i.e. the uptake coefficients are constant. In the case of binary aerosols and STS PSCs the uptake coefficients of some reactions depend on the composition of the droplets, i.e the hydrolysis reactions of $ClONO_2$, $BrONO_2$ and $N_2O_5$, as well as the reaction of HCl between $ClONO_2$ and HOCl (Sander et al., 2011)."

2) We added some discussion on the effect of water vapour on heterogeneous binary in/on aerosols and STS.

"The water vapour concentration seem to strongly affect the transformation of chlorine from the reservoir species to the intermediate ones in the beginning of Arctic winter. The fractions of intermediate and reservoir chlorine species change significantly with water vapour concentration in November and December, when STS PSCs start to form. The water vapour concentration affects the composition of binary aerosols an STS. When more water condenses to the particles the uptake coefficients for the heterogeneous reactions increase, i.e. the reaction rate increases."

• The authors claim an important role of NOx chemistry in warm winters. I would appreciate plots showing this, e.g., altitude-time plots of vortex-averaged NOx or / and altitude-time plots of the corresponding reaction rates ($NO_2 + O$ → ... for NOx chemistry, and, for comparison, $Cl_2O_2 + h\nu$ → ... and perhaps $ClO + O$ → ... for ClOx chemistry).
From the difference between the no-hetero and the full chemistry simulations it can be seen that the ozone loss in winter 2012/13 is gas phase chemistry driven. The text was rearranged, and the NOx-part was moved to the discussion of the ozone profile figure (Fig 8). The text now reads:

"In winters when the polar vortex is unstable and small or disturbed the Brewer–Dobson circulation brings more $NO_x$-rich air to the polar vortex than usual. Hence the ozone loss in the 2012/13 winter was produced mostly by $NO_x$ chemistry as shown previously by e.g., Sagi et al. (2017), and

can be seen from FinROSE results by comparing the simulations with and without the heterogeneous chemistry."

• The authors do not discuss the influence of heterogeneous NOx chemistry on ozone. Can the model results be used to answer the question whether the heterogeneous reaction N2O5 + H2O → 2HNO3 reduces NOx and thus NOx -driven ozone loss in cold years (compared to other years)? We were not able to attribute any change in NOx chemistry to changes in water vapour, but we modified the text to more clearly separate the discussion of the NOx and chlorine caused ozone loss.

Minor comments
• 1/17: "2-7% more ozone loss than in colder winters" ⇒ Does this mean "2-7% stronger increase of ozone loss than in colder winters"?
Yes. We changed the text in the abstract:

"We found that the impact of water vapour changes on ozone loss in the Arctic polar vortex depends on the meteorological conditions. The strongest effect was in intermediately cold conditions, such as 2013/14, when added water vapour resulted in 2-7% more ozone loss due to the additional polar stratospheric clouds (PSC) and associated chlorine activation on their surface, leading to ozone loss. The effect was less pronounced in cold winters such as 2010/11 because cold conditions persisted long enough for a nearly complete chlorine activation even in simulations with observed water vapour. In this case addition of water vapour to the stratosphere led to increased volume of ice PSC but it could not increase the chlorine activation and ozone destruction significantly. In the warm winter 2012/13 the impact of water vapour concentration on ozone loss was small, because the ozone loss was mainly $NO_x$ induced."

• 2/1 and 14/3: "warms the climate" means "warms the troposphere"?
We changed the text as suggested.

• 3/2-3: "cooling stratosphere ... slowing down some gas-phase reactions": Which gas-phase reactions are meant? In the polar stratosphere (during winter / spring) an important reaction is the three-body reaction ClO + ClO + M → Cl 2 O 2 , the rate-constant of which increases for decreasing temperature.
It is true that the effect of temperature on future ozone is complex and our statement was too general. We meant the second-order reactions mainly in NOx and HOx ozone loss cycles. This effect is mainly seen outside the polar vortex, while the effect on PSCs from temperature is seen within the vortex. We improved the text to clarify this.

• 5/3: "around 80 hPa": In view of the discussion in 5/29-31 this should be formulated more precisely, e.g. "at the cold point, which lies approximately x hPa below 80 hPa"
We added some details to the description of the lower boundary condition for water vapour.

"The simulations differed from each other by the prescribed water vapour concentration in the tropical tropopause region (stratosphere between 21S–21N, below 80 hPa), where it was prescribed as follows: (1) water vapour taken from ERA-Interim (Interim simulation), (2) increased water vapour (Max simulation), and (3) decreased water vapour (Min simulation). Specifically, the SWV lower boundary conditions for Min and Max simulations were obtained by multiplying values from ERA-Interim between tropopause and 80 hPa, and between 21 S–21  N by monthly coefficients ranging between 1.46-1.7 (Max) and between 0.5-0.63 (min), so that they approximately correspond to the driest and wettest CCM, as determined by SWV values at the tropical tropopause, across models analyzed by Gettleman et al. (2010)."

• 5/29-31: "leads ... by 3-4 weeks ... Brewer-Dobson circulation ... too fast": Does this mean that between the cold point and 80 hPa the ERA-Interim circulation takes 3-4 weeks less than the real circulation. Is the distance between the cold point and 80 hPa large enough to gain such a difference?

Thank you for this comment. We believe that a too fast BD transport in ERA-I likely contributes to the difference but we are not sure if it can indeed explain such a large delay. Therefore we rewrite the text as follows:

"However, Interim variability leads that of MLS by 3–4 weeks. The reason for the time lag between Interim and MLS is not clear although it could at least partly be associated with a too fast Brewer–Dobson circulation in ERA-Interim which is responsible for upward transport of the water vapour anomalies in the tropics (Schoeberl et al., 2012)."

• 6/3: "gains a small amount of water": How, by horizontal mixing?

In the revised version this claim is removed and the sentence is rewritten. In the original version we unfortunately overlooked the fact that in Min simulation the scaling factor was varying with a seasonal cycle between 0.5-0.63, rather than fixed at value 0.5 as was stated in the original text. This point is clarified in the revised version. Therefore the difference between Min and Interim of factor 0.6 in Fig. 1 is consistent with prescribed values and there is no evidence of gained water vapour in Min simulation. The new text reads:

"The Max simulation has 2–3 ppm more water vapour in the tropics than the Interim simulation, while the Min simulation is about 1.5 ppm drier than the Interim simulation. These differences correspond to the ratio between Max/Interim of approximately 1.55-1.6 and about 0.55–0.6 between Min/Interim, i.e. they are consistent with the prescribed boundary conditions."

• 8/4: "For example too dry models may not be able to simulate a large Arctic ozone loss such as of 2010/11": How does this sentence relate to the preceding sentence?

The sentence "For example too dry models may not be able to simulate a large Arctic ozone loss such as of 2010/11." was removed from the results section and the discussion was moved to the Discussion and conclusions section.

• 9/15: "in spring": and also during southward excursions of air masses during winter

Yes, it is more related to availability of sunlight than the time of the year, the text was changed accordingly:

"When sunlight reaches the polar vortex these species are easily dissociated to form active chlorine species that participate in the catalytic ozone depletion cycles, i.e. ClO x (ClO, Cl 2 O 2 and Cl)."

• 9/16: "PSCs sustain the regeneration of ClO x ": This is only possible if both reaction partners for a heterogeneous reaction are still present.

The reservoir species are continuously formed through the ClO+NO2 and Cl+CH4 reactions, but as long as there are PSCs the ClOx is regenerated. We added some text to clarify this part.

"Active chlorine goes back to reservoir species through reactions with NO2 and CH4, however if PSCs are present the regeneration of ClOx is sustained."

• 10/9: "rather short": The green curve in Fig. 6d lies above 60% for about 2 months. Is this meant by "rather short"?

We meant in comparison to the winter 2010/11, the text was corrected:

"The 2015/16 winter started similar to the cold winter 2010/11 and nearly all of the chlorine was activated at the beginning of January, but the deactivation started already in the end of January making the period with high ClOx shorter than in winter 2010/11."

• 10/17: "differs significantly ... during the period with high ClO x ": The difference is mostly less than 10%. Is this meant by "differs significantly"?
The formulation was quite vague, we altered this and the next sentence as follows:

"The difference in concentration of active chlorine and reservoir species between the Min and Max simulations are smallest during the cold periods, due to the effective processing on the PSCs (Fig 6.). The cold winter 2015/16 shows a very small range, and the intermediatly cold winter 2013/14 a wider range in concentrations. The water vapour content has less effect on the chlorine partitioning in cold winters."

• 10/18: What exactly is meant by "chlorine activation period": the time when most chlorine exists in active form?
The chlorine activation period was not defined exactly, it was based on a subjective interpretation of the figures. We made an effort to define it more robustly and altered the text accordingly.

• 11/4: "... only at the 475 K level": Does this mean that the 475 K level is used for the definition of the whole vortex? If so, this should be mentioned (and perhaps be discussed) already in 6/7.
We used the PV on the 475 K level to define the polar vortex only in Fig 7. The text was modified as follows:

"In Fig. 7 the polar vortex is defined using the potential vorticity limit 36 PVU only at the 475 K level."

• 11/11: "FinROSE seems to underestimate the ozone loss, possibly due to a general 10% negative bias in total ozone": Why does an underestimation of the ozone concentration lead to a significant underestimation of the ozone loss? The rate of ozone loss in polar winter/spring is largely determined by the rates of reactions like $ClO + ClO + M \rightarrow Cl_2O_2 + M$ and $BrO + ClO \rightarrow$ ... .
In extreme cases a negative bias could limit the ozone loss in DU, however for Arctic conditions this is generally not the case. We don't have a clear understanding of the reason for the underestimation, and this statement is not valid here. We changed the text accordingly.

• 11/23: "stopping the catalytic ozone cycles and ozone loss early": The ozone loss stops around the beginning of March (Fig. 7d). Is this meant by "early"?
We meant earlier than 2011. The text has changed:

"A relatively small ozone loss of 56 DU was simulated in 2015/16, which was due to the unstable polar vortex, which split and warmed, stopping the catalytic ozone cycles and ozone loss in the beginning of March, i.e. earlier than in 2010/11 and 2013/14."

• 13/4: "the heterogeneous chemistry destroyed about 36 DU of ozone": In fact, ozone is destroyed by gas-phase reactions. Heterogeneous chemistry produces (some of) the corresponding reactants.
Yes, the formulation was missleading, the text now reads:

"In 2013/14 the heterogeneous chemistry caused about 40 DU ozone destruction, which is about 51 % of the ozone loss and ICE and NAT/STS about 23 DU (30 %), when the total ozone loss was 79 DU in the Interim run.

• 14/24: "in higher level": Does this mean at higher altitudes?

No, we meant more complete. The text has changed:
"Cold winters differ from the warm winters regarding the ozone loss and the fraction of ozone loss initiated by heterogeneous chemistry, during cold winters the PSC volumes are larger and thus chlorine activation within the polar vortex is more complete."

• Tables 1, 2, 3: As a "service" for the reader the character of the years (warm, intermediate, cold) might be added (as was done in the figures). Perhaps the winter 2015/16 might be called "initially cold", in order to distinguish it from 2010/11.
The character of the years was added to the tables as well.

**Technical details**
• Please check the "s" of the plural of substantives or singular of verbs
1/13, 2/3, 10/12, 10/16-17, 14/4, 15/29
We made the suggested corrections to the text.

• 1/22: "processes. Especially" ⇒ "processes, especially"
OK
• 2/7: "(2002)" ⇒ ", 2002"
OK
• 4/9: "lagrangian" ⇒ "Lagrangian"
OK
• 6/27: The abbreviation "BD" has not been defined before (and is used only once).
OK
• 6/33: "(Fig. 3)" ⇒ Fig. 3
OK
• 7/18: Really "0.3 ppt" (or 0.3 ppb)?
The limit we used was 0.3 ppt, it seems low but at temperature above 200K only a small fraction of the nitric acid is dissolved in the aerosols. We had a closer look at the limit and we have now increased it to 0,6 ppt, which approximately corresponds to 202 K. We changed the text accordingly.
• Table 1: Please show only 2-3 non-zero digits.
OK
• Table 2: "mixing ratio integrated over the winter of activated chlorine" ⇒ "mixing ratio of activated chlorine integrated over the winter"
OK

---

## Author Response (AR1)

The topic of this paper is Arctic ozone loss in general and the sensitivity of Arctic ozone loss on the polar water vapour concentrations in particular. If the authors agree, they need to to remove all the discussion of mid-latitude ozone loss and the related cycles. Further, discussion/citation of upper stratospheric ozone loss is not helpful.

The topic of the paper (water vapour sensitivity) is of importance and the the approach using a CTM to focus on the impact of water vapour on PSCs on heterogeneous chemistry is good. On the other hand, the results on the sensitivity will gain on impact if the representation of PSCs in the model, the simulation of chlorine activation and ozone loss are clearly demonstrated in the paper. I do not think that this is the case in the present version of the paper (the cited papers are not sufficient, see below).

For example, the impact of humidity on heterogeneous chemistry should be different for different types of PSCs. If in this model the PSC types are dominated by a different type that in reality (or even in different models) carrying over the information on sensitivity deduced here can be problematic. There is some comparison with ERA-interim, but this is not the real world. Also there i.e. very little information on chlorine chemistry (except in Fig.6, where there is no distinction between HCl and ClONO2 for example, see also below).

The paper addresses a second case, namely a warm and dynamically unstable winter in the Arctic stratosphere. In such a case, in contrast to the halogen induced ozone loss in cold Arctic winters, there is little ozone loss below 500 K and ozone loss in the middle stratosphere (NOx-induced destruction) becomes important (e.g., Konopka et al., 2007; Sagi et al., 2017). However, this is a completely different chemical mechanism, which will have a very different dependence on water vapour (clearly no impact of heterogeneous chemistry). Therefore this second case needs to be very clearly discriminated from the "halogen" case throughout the paper.

In summary, I recommend focusing the paper and a providing a better basis and justification for the work presented. I also suggest a better balance of the cited literature (as stated in the quick review, I am not suggesting to cite all the references mentioned here). I am sorry for the many critical remarks but I believe that a revised paper taking into account these comments would be much stronger than the present version.

General

The discussion of the HOx chemistry in mid-latitudes and in the upper stratosphere (and the associated references) in the introduction is confusing. In the Arctic, in cold winters, an increase in water vapour is expected to enhance ozone loss (assuming that a substantial amount of stratospheric chlorine is present) by enhancing heterogeneous reactions (Shi et al., 2001; Drdla and Müller, 2012). The situation can actually be more complicated as there is no simple argument that "more PSCs" means "more ozone loss". This is also a finding of this paper, if one considers the case of the cold Arctic winter 2010/11. The latter point should be more emphasized in the paper I suggest.

Recently, climatologies of PSC occurrence have become available from observations (Spang et al., 2017; Pitts et al., 2018). The question how important the discrimination between different PSC types in a model is for a successful simulation of polar ozone loss is not trivial. This paper could contribute substantially to this issue; perhaps more than visible in the present version of this paper. This aspect could be very relevant for the discussion of the sensitivities on water vapour (which will be different for different types of PSCS). Finally, models might misrepresent PSC volume density compared to observations; Khosrawi et al. (2018) report that the comparison between the PSC volume density as simulated with EMAC and the one derived from Envisat/MIPAS observations

shows that the simulated PSC volume densities are several orders of magnitude smaller than the observed ones.

I also think that the demonstration that FinROSE is successful in modelling polar ozone loss is not sufficient. Very little information is given in this paper. Fig. 6 only give a summary (e.g. I cannot judge whether or not the recovery of active chlorine into ClONO2 is convincing), Other models have done comparisons with observations in details (see e.g., Wohltmann et al., 2017, and references therein). Here the reference to Karpetchko et al. (2013) is given, but in this study FinROSE is used with PSCs "switched off", so this paper is not valid to support the performance of the model for Arctic ozone loss in cold winters (see also below). In section two of the paper a description of the initialization of the model should be given. How is total chlorine (Cly) initialized? What ware the initial values for HCl, ClONO2, N2O5 etc – this information would be helpful to interpret the results of the paper. How good is ERA-I ozone in comparison to MLS ozone? How well does FinROSE simulate downward transport in the Arctic (compare to MLS N2O?).

One important driver of chemical ozone loss in cold Arctic winters is the stratospheric chlorine loading. I suggest making this point and briefly discuss the temporal development of stratospheric chlorine (e.g., Engel et al., 2018).

We thank the Referee #1 for the thorough and valuable comments. We think that the issues raised here in the introduction has been addressed in the detailed comments below, except for the discussion related to the temporal evolution of stratospheric chlorine. A discussion was added to the Introduction in the manuscript and a reference to Engel et al. (2018) was added. In addition, the references were checked and corrected and new references were added.

The discussion of the ozone loss results was changed in order to more clearly discriminate the loss caused by NOx from the loss related to halogens.

The PSC volume was changed to PSC area, which didn't change the results and conclusions, but the text was altered and some of the figures were replotted.

The definition of the start and end of the polar vortex season was defined in more detail, which changed the appearance of some of the figures, but did not change the results or conclusions.

The validation of the FinROSE model has been done earlier, therefore we didn't include validation in this study, e.g. in Karpechko et al. (2013) we have compared the full chemistry simulation to satellite observations. The ERA-Interim ozone have been validated by e.g. Dragani (2011).

Details

• title: the title is too general, is sounds like the uncertainty in simulated Arctic ozone is due to water vapour. But there are more reasons for uncertainties
We changed the title, it now reads:

"Linking uncertainty in simulated Arctic ozone loss to uncertainties in modelled tropical stratospheric water vapour"

• abstract, l. 8: the point here is that there is a one-to-one correspondence between entry value and polar water vapour conditions – correct? This point could be made more clearly.

The change in water vapour in the tropical tropopause region was seen nearly as a one-to-one correspondence in the Arctic polar vortex. We highlighted this finding in the text.

"In this study we investigate the sensitivity of simulated Arctic ozone loss to the amount of water vapour that enters the stratosphere through the tropical tropopause."

"The water vapour changes in the tropical tropopause led to about 1.5 ppm less and 2 ppm more water vapour in the Arctic polar vortex compared to the ERA-Interim, respectively. The change induced in the water vapour concentration in the tropical tropopause region was seen as a nearly one-to-one change in the Arctic polar vortex."

• abstract, l 13-15: I think this is an important finding of this paper.
We further streamlined the text to highlight this finding.

"The effect was less pronounced in cold winters such as 2010/11 because cold conditions persisted long enough for a nearly complete chlorine activation even in simulations with observed water vapour. In this case addition of water vapour to the stratosphere led to increased area of ICE PSC but it could not increase the chlorine activation and ozone destruction significantly.

• abstract, l 17: 2–7% more: this is a very important result of this paper. But can be anything worked out what the mechanism is? It would be important to state this finding in the abstract.
We made some changes to the changed the Abstract, related also the the previous comment:

"We found that the impact of water vapour changes on ozone loss in the Arctic polar vortex depends on the meteorological conditions. The strongest effect was in intermediately cold conditions, such as 2013/14, when added water vapour resulted in 2-7% more ozone loss due to the additional polar stratospheric clouds (PSC) and associated chlorine activation on their surface, leading to ozone loss. The effect was less pronounced in cold winters such as 2010/11 because cold conditions persisted long enough for a nearly complete chlorine activation even in simulations with observed water vapour. In this case addition of water vapour to the stratosphere led to increased area of ice PSC but it could not increase the chlorine activation and ozone destruction significantly. In the warm winter 2012/13 the impact of water vapour concentration on ozone loss was small, because the ozone loss was mainly NOx induced."

• p.2, l. 1: Temperature may show a warming in the troposphere but a significant cooling near the tropopause in connection with deep convection (Kim et al., 2018), so the connection between climate change and tropopause temperature may not as straightforward as suggested here. I suggest more discussion of this point which is important for this paper.
We specified the text and added the reference:

"A warmer climate in the troposphere is suggested to increase stratospheric water vapour (SWV) through increases in the water vapour entering through the tropopause, which would further warm the troposphere below (Dessler et al., 2013). However tropospheric warming may also lead to a significant cooling near the tropopause in connection with deep convection (Kim et al., 2018), so that the link between warming climate and tropopause temperature is not straightforward."

• p.2, l. 3: "affect chlorine partitioning" but how? Is this relevant for polar chemistry?
We removed some of the discussion that was not relevant to the polar chemistry.

• p.2, l. 7: this citation is for the upper stratosphere; I do not think it is appropriate here.
The citations were checked and changed (Solomon 1999 and Khosrawi 2016)

p.2, l. 12: One could also mention the observations discussed by Kuttippurath and Nair (2017) here.
The reference and discussion was added.

"Several studies have discussed Antarctic ozone depletion and its recovery (see e.g. Eyring (2010), Dameris (2014), Solomon et al. (2016), Chipperfield et al. (2017), Kuttippurath and Nair (2017) and Strahan and Douglass (2018). "

"Kuttippurath and Nair (2017) recently showed that Antarctic ozone has begun to recover based on ozone balloon soundings and total ozone data from satellite instruments. Based on profile data from the Aura Microwave Limb Sounder (MLS) of HCl and ozone Strahan and Douglass (2018) showed a decline in lower stratosphere chlorine and a corresponding decline in ozone depletion for the period 2013-2016 compared to the period 2004–2007. However, a recovery of ozone to 1980 ozone levels is projected not to occur until around 2025–2043 in the Arctic and 2055–2066 in the Antarctic (Dhomse et al., 2018)."

• p. 2, l. 14: do you have references to more recent studies to back up this statement on recovery?
We added reference to a recent study of Dhomse 2018, and changed the recovery dates according to their results. Also references to Dameris 2014, Solomon et al. (2016), Chipperfield et al. (2017), Kuttippurath and Nair (2017) and Strahan and Douglass (2018) were added. See also previous comment.

• p. 2, l. 17: remember that the focus here is on Arctic ozone.
References to Smalley 2017 and Rosenlof 2001 were removed from here.

• p. 3, l. 3: I do not think this statement is correct. Please consider the temperature dependence of the main polar ozone loss cycles (von Clarmann, 2013; Canty et al., 2016; Wohltmann et al., 2017). I you do not agree, please specify "some".
We think the text is correct in general, but it is true that the offset would probably come from slower ozone loss reactions in areas outside the vortex, which could contribute to a so called super recovery of the ozone layer. We changed the text accordingly.

"However, cooling of the stratosphere could at least partially offset the effect of the increased PSCs by slowing down the second-order reactions in ozone loss cycles (Rosenfield et al., 2002 and Revell et al., 2012). This effect is mainly seen in NOx and HOx induced loss outside the polar vortex, while the effect on PSCs from temperature is seen within the vortex"

• p.3, l 17/18: this is more than "likely" if temperatures are high, there are no PSCs and thus there is no impact of water vapour on PSCs and heterogeneous reactions. Under these conditions NOx chemistry indeed is relevant (but we are missing a discussion here of the impact – if any – of water vapour on the NOx chemistry in question). In any even (see also other points in this review) these two cases must be clearly separated. You are looking at different processes here.
We modified the text in the Introduction slightly. In addition we reorganised the discussion about NOx induced loss in the Results section to more clearly separate the discussion of the NOx and chlorine caused ozone loss. We were not able to attribute any change in NOx chemistry to changes in water vapour.

"In the winter 2012/13 the polar vortex was more unstable and a vortex split occurred early January due to a sudden stratospheric warming (SSW), and NOx rich air from the mesosphere descended to the upper stratosphere and led to ozone loss there. Thus the effect on Arctic ozone depletion from changes in SWV will depend on the meteorological conditions, and the dynamical stability in a given winter."

• p. 3, l 28: the idea of a "controlled" experiment is good! also change to "impact of . . . on ozone loss".
Sentence was changed to:

"Therefore a more controlled experiment is needed in order to assess impact of these SWV changes on ozone loss."

• p. 4., l 9: this is not an important point, but is has recently been argued that the reaction CH3O2 + ClO is important for polar ozone loss (Müller et al., 2018); is this reaction taken into account in the chemical scheme used here?
CH3O2 is not included in our chemistry scheme, and therefore we are unfortunately not able to discuss the importance of the CH3O2 + ClO reaction in the Arctic polar vortex.

• p. 4, l. 16/17: As it is very important for this paper: describe here how the dependence on water vapour of the heterogeneous reactions (i.e. the γ) is determined.
The following text was added to the description of the chemistry scheme.

"The reaction rates on NAT and ICE PSCs are not directly affected by the water vapour concentration except through the available surface area, i.e. the uptake coefficients are constant. In the case of binary aerosols and STS PSCs the uptake coefficients of some reactions depend on the composition of the droplets, i.e the hydrolysis reactions of ClONO2, BrONO2 and N2O5, as well as the reaction of HCl between ClONO2 and HOCl (Sander et al., 2011)."

• p. 4, l. 19: As it is very important for chemistry simulations at the poles: describe here if(how) spherical geometry is properly taken into account in the photolysis code.
We changed the text and added references to Stamnes (1988) and Dahlback and Stamnes, (1991) where the pseudo-spherical approximation is discussed.

"Look-up-tables of photodissociation coefficients were pre-calculated using the PHODIS model (Kylling et al., 1995). Within PHODIS the radiative transfer equation is solved by the discrete ordinate algorithm (Stamnes et al.,1988). This algorithm has been modified to account for the spherical shape of the atmosphere using the pseudo-spherical approximation (Dahlback and Stamnes, 1991)."

• p. 4, l. 29: the reference to Karpetchko et al. (2013) is important here as this is the only citation give to support the performance of FinROSE in simulation polar ozone loss. However, in Karpetchko et al. (2013) FinROSE is used with PSCs "switched off", so this paper is not valid to support the performance of the model for the issues discussed in this paper.
Please note that Karpechko et al have both simulations, with and without heterogeneous chemistry.

• p. 5, top: It is not clear to me what was done exactly here: what is "Interim(MAX)"? Why did you not simply shift the water vapour values up and down by some value preserving the variability?
Please note that the variability is not preserved across CCMs either and our approach was chosen to replicate simulations by CCM. In our approach we replicate changes in amplitude as well as shift in mean water vapour. The description was improved.

"The simulations differed from each other by the prescribed water vapour concentration in the tropical tropopause region (stratosphere between 21S–21N, below 80 hPa), where it was prescribed as follows: (1) water vapour taken from ERA-Interim (Interim simulation), (2) increased water vapour (Max simulation), and (3) decreased water vapour (Min simulation). Specifically, the SWV lower boundary conditions for Min and Max simulations were obtained by multiplying values from ERA-Interim between tropopause and 80 hPa, and between 21 S–21  N by monthly coefficients

ranging between 1.46-1.7 (Max) and between 0.5-0.63 (min), so that they approximately correspond to the driest and wettest CCM, as determined by SWV values at the tropical tropopause, across models analyzed by Gettleman et al. (2010)."

• p 5, l 9: The comparison to MLS ozone would be important. It should be shown and discussed in the paper in detail. Further, MLS HCl (and possibly other measurements of Cl-species could be helpful to validate the model).
Please note that validation of FinROSE-ctm have been done earlier (e.g., Damski et al 2007, Thölix et al., 2010, Salmi et al., 2011 and Thölix et al. 2016) and therefore it was not included in this paper. In the revised version we provide references to our earlier studies where such validation was done.

"The FinROSE-CTM has previously been used to study the impact of meteorological conditions on water vapour trends (Thölix et al., 2016), ozone and NO x chemistry in the mesosphere (Salmi et al., 2011), Arctic polar ozone loss (Karpechko et al., 2013) and the impact of the driver data on the model transport (Thölix et al., 2010). Long term trends of Arctic and Antarctic ozone losses, past and future, have been investigated by using driving data from a chemistry–climate model (Damski et al., 2007a). The model results showed good agreement with satellite and ground based observations. The FinROSE water vapour was compared to observations of water vapour profiles from the Microwave Limb Sounder (MLS) and frost point hygrometer soundings from Sodankylä Thölix et al. (2016). The extent of ICE PSCs simulated by FinROSE was compared to Cloud-Aerosol Lidar and Infrared Path finder Satellite Observation (CALIPSO) data in Thölix et al. (2016).The total ozone distribution was compared to data from Total Ozone Mapping Spectrometer (TOMS) and the Ozone Monitoring Instrument (OMI) satellite instruments in Damski et al. (2007a), Thölix et al. (2010) and Karpechko et al. (2013). Salmi et al. (2011) compared the NO x and ozone profiles in FinROSE to data from the tmospheric Chemistry Experiment Fourier Transform Spectrometer (ACE-FTS) instrument."

• p. 6, l 7: which reference theta level was used for mPV? How is 36 PVU chosen?
The reference theta level is 475K. The limit 36 PVU was chosen based on publications by Rex at al. (1999) and Streibel et al. (2006), who have shown that 36 PVU is a good approximation for the vortex edge. The text was modified as follows:

"The polar vortex was identified using the modified potential vorticity (mPV) (Lait, 1994), with the 475 K potential temperature as reference level. Here the polar vortex is defined as the area enclosed by the 36 PVU isoline separately for every model level. The 36 PVU contour approximately correspond to the region of maximum PV gradient, i.e the polar vortex edge (Rex et al, 1999, Streibel et al, 2006)."

• p. 6, l. 23: Here (and perhaps elsewhere) the question arises if downward transport in the FinROSE model in the Arctic polar vortex is simulated appropriately. This issue has an impact of water vapour in the polar vortex. A comparison of simulated N2O with observations would be helpful here.
Please note that validation of FinROSE-ctm have been done earlier (e.g., Damski et al 2007, Thölix et al., 2010, Salmi et al., 2011 and Thölix et al. 2016) and therefore it was not included in this paper. In the revised version we provide references to our earlier studies where such validation was done. See also previous cmment.

• p. 7., l 5: when dehydration occurs the initial water vapour and thus the Max/Min scenarios will not be relevant any more, as the polar water vapour is set to the equilibrium value over ice – correct? This issue should be discussed in the paper.

The dehydration is calculated based on the settling velocity of ICE PSC particles. In the Arctic the dehydration is typically never complete, at least not on large areas. The not enough time for all particles to sediment out of a layer at a given grid point during the time the ICE PSCs persist, and the area cold enough for formation of ICE PSCs is typically small compared to the Arctic polar vortex. Therefore the differences in water vapour concentration persists also after dehydration has occurred. The following text was added to the Modelling and data section:

"The sedimentation of PSC particles, which can lead to denitrification and dehydration, is calculated based on the settling velocity, which takes into account the PSC particle size."

• p. 7, l 15: what is meant by NAT/STS volume, Just adding both PSC types? But the impact of increased water vapour on NAT and STS will be different. So why has this been done; I suggest separating NAT and STS.
Yes, the volume of the grid points containing either NAT or STS (or both) were summed. To better separate the impact of water vapour on the heterogeneous chemistry we now separated the NAT and STS area. We now also use area instead of volume. The text was altered accordingly.

• p. 7, l. 22: is there some impact of sedimentation on the duration of ice PSCs?
Yes, the dehydration deepens with time. The longer the ICE PSCs persist the more of the particles have time to settle out of a given layer. See comment above related to dehydration.

• p. 7, l. 23/24: Note that the Calipso PSC product has more recent information now (Spang et al., 2017; Pitts et al., 2018); I suggest using the most recent information. For example the estimated ice area might change.
References to Spang et al., (2018) and Pitts et al., 2018 were added. The PSC volumes have changed to PSC areas at 20 km level, because of easier comparison to observations. In general the timing of PSC in our model is good, but our ICE area is somewhat overestimated compared to Calipso data. The text was modified according the new figures.

• p. 8, l 4: "too dry models" this point sounds very speculative
The sentence "For example too dry models may not be able to simulate a large Arctic ozone loss such as of 2010/11." was removed from the results section and the discussion was moved to the Discussion and conclusions section.

p 8, l. 25: One question that arises here how well the model simulates the size of the vortex – as this point might be relevant for assessing the PSC area.
The polar vortex is calculated using ERA-Interim data, and not altered by the model. The vortex limit has been discussed in the reply to question of p. 6, l 7.

• p 9, l 10: small twice
Corrected.

• p. 9, l 14: chlorine activation does not require PSCs, it starts on cold binary aerosols, but is also humidity dependent (e.g. Solomon, 1999; Wegner et al., 2012).
It is true that the activation starts on binary aerosols. However, there is no hard limit between binary aerosols and PSCs. In FinROSE the switchover from binary aerosols to STS is done at 215 K. At that temperature the STS is practically a binary aerosol, the large influence from dissolved water and HNO3 happens at even lower temperatures. The humidity is taken into account through the composition and uptake coefficient, as described in a previous comment (p. 4, l. 16/17). The text was modified as follows:

"In the cold conditions within the polar vortex the chlorine species are transformed, through heterogeneous reactions, into intermediate species such as Cl2."

• p 9, l 25: which reservoir species?
In early March mostly ClONO2 is formed and a few weeks later HCl. This was added to the text.

• p 10, top: the start of activation is one thing, but not really what determines how much ozone loss happens in a particular winter.
The purpose here is only to describe the evolution of active chlorine for different winters.

• p 10, l 12: which process is responsible here?
The reason is probably heterogeneous reactions that occur in binary aerosols and STS, the added water vapour increases the uptake coefficients and to some extent also the surface area. The text was modified as follows:

"The water vapour concentration seem to strongly affect the transformation of chlorine from the reservoir species to the intermediate ones in the beginning of Arctic winter. The fractions of intermediate and reservoir chlorine species change significantly with water vapour concentration in November and December, when STS PSCs start to form. The water vapour concentration affects the composition of binary aerosols an STS. When more water condenses to the particles the uptake coefficients for the heterogeneous reactions increase, i.e. the reaction rate increases."

• p 10., l 23: but why?
During the coldest period the difference in chlorine activation is small probably due to the ICE PSCs that appear in all simulations. The conditions were favourable for high chlorine activation in all simulations and therefore the differences were small. A clarification was added to the text.

"The chlorine activation in 2015/16 winter seems to be less dependent on water vapour content, probably due to the ICE PSCs that appear in all simulations. Therefore, the conditions were favourable for high chlorine activation in all simulations. The difference in the fraction of activated chlorine between simulations is only few percents, only when the deactivation starts (in the end of February) the difference is more than 5 %. "

• p 10., l 26: I am not convinced that this statement is correct, is this really a cause and effect relation?
We agree with the referee in the sense that the statement "The amount of chlorine activation correlate with the volume of NAT/STS PSCs." is probably too broad. The correlation is more complex, we altered the text to reflect this. We now deal with the STS and NAT separately, as suggested earlier.

"The timing of the changes in the partitioning of the chlorine species correlates well with the occurrence of PSCs, e.g. the chlorine reservoir species start to transform into intermediate species when the STS PSCs appear (Fig. 5)."

• p 10., l 30: I do not think that table 2 is a good summary of the chlorine activation simulation in the model run, too many important details are missing. (See also other comments in this review).
We added monthly means of activated chlorine to the table, and now the timing of the chlorine activation in different winters can be seen from the table. The text was modified according the new table values.

"Table 2 shows the vortex averaged ClO x as a cumulative sum over the whole winter, and as monthly mean concentration. The sums are integrated from November to April. The cumulative sum

has information about both the duration of the chlorine activation period and the concentration of ClO x , while the monthly average concentration shows the timing of chlorine activation. The cumulative chlorine activation was largest in winter 2010/11 and the smallest in 2012/13. The activation started in November every year, but remained small until December. The winter 2010/11 differs from the others, with high chlorine activation from January to March, giving the largest cumulative sum of the studied winters. Even in April the ClO x concentration remains elevated. The warm winter 2012/13 had the smallest cumulative chlorine activation, significant chlorine activation was seen only in December and January. The changes in water vapour between the Min/Interim/Max simulations had the largest effect on the cumulative ClOx in moderately cold winters (2010/11 and 2013/14), where the increase from Interim to Max was 3 to 6 % and change from Interim to Min was -1 to -11 %. In the cold winter 2015/16 the respective changes were only 3 and -2 %, and in the warm winter 2012/13 the changes were 4 and -1 %."

• p 11, sec 3.4: it would helpful to have more comparisons to ozone loss from simulations of other models.
We were not able to find additional references than Vogel (2011), Sinnhuber (2011) and Pommereau et al. (2013), related to the effect of water vapour on Arctic ozone depletion.

• p 11, l 18: is this statement on NOx also true for FinROSE? If yes, what is the evidence from the model simulation for this statement?
The text was rearranged, and the NOx-part was moved to the discussion of the ozone profile figure (Fig 8). It can be seen from the no-hetero run that the ozone loss in winter 2012/13 is gas phase chemistry driven. The text now reads:

"In winters when the polar vortex is unstable and small or disturbed the Brewer–Dobson circulation brings more NOx -rich air to the polar vortex than usual. Hence the ozone loss in the 2012/13 winter was produced mostly by NO x chemistry as shown previously by e.g., Sagi et al. (2017), and can be seen from FinROSE results by comparing the simulations with and without the heterogeneous chemistry."

• p 12, l 2: this might be true, but this is an example where speculation could be replaced by statements based on the actual simulations. Has a simulation been attempted with a better resolution?
We did an additional experiment and it confirmed our assumption. The ozone loss in 2010/11 with higher resolution (1.5 x 3) were about 10 DU larger than with FinROSE's resolution. Unfortunately, redoing all experiments with increased resolution are currently not feasible for technical reasons, therefore we report in the manuscript the results based on the original resolution.

"An additional sensitivity experiment showed that the difference compared to other studies can be, at least partly, due to the coarse horizontal resolution in FinROSE (3x6), which is not sufficient to fully capture the deepest ozone loss. Specifically, repeating Interim simulation for winter 2010/11 with higher resolution (1.5x3) than in the original simulation showed larger ozone loss by 15 DU."

• p 12., l 7: "without het. chemistry": it should be clearly stated what has been assumed regarding the heterogeneous reaction N2O5 + H2O, which is not temperature dependent but would be important here.
No separate assumptions have been made for the N2O5 + H2O reaction. There were two different setups used, one where no heterogeneous reactions were included, and another one where the temperature for the aerosol/PSC scheme was limited to 200K. The one with the 200K limit allows some heterogeneous processing on binary aerosols and some STS that are very dilute in HNO3. The N2O5 + H2O reaction is practically not temperature dependent, but the rate will be affected by the

smaller surface area available. The possible effect on e.g. NOx induced loss was not considered here. We altered the description of the setup for the no-hetero simulations.

"In the second simulation the formation of PSCs was limited by setting the air temperature passed to the heterogeneous chemistry module to 200 K, similarly to what was done in Karpechko (2013). This setting allows some heterogeneous processing on binary aerosols and some STS that are very dilute in HNO3, and due to the temperature limit the surface area densities will remain quite small."

• p 12., l 13: this is not true for the het. reaction N2O5 + H2O
The idea to limit the temperature to 200K was to allow reactions on binary aerosols, but not on PSCs. It is true that a small amount of STS can form, but due to the temperature limit the surface area densities will remain quite small. See also previous comment.

• p 12., l 16-18: this statement is really confusing: if I understand correctly, only 30 DU ozone loss is caused by NAT/STS/ICE PSCs in Arctic winter 2010/11: This is in contrast to statements elsewhere in the paper and also to literature and our general understanding of Arctic ozone loss. rent types of PSCs are for the results of this section.
We believe that the result shows that the binary aerosols and STS have an effect. There are studies that suggest that binary aerosols are more important for chlorine activation than PSCs (e.g. Drdla and Müller,2012; Wohltmann et al., 2013; Kirner et al., 2015). However, the contribution of PSCs are also important (e.g. Wegner et al., 2016). We added discussion on these results.

"In the Interim simulation with full chemistry in 2010/11 about 90 DU ozone was depleted, of which the heterogeneous chemistry caused 56 DU depletion, i.e. about 62 % of the total ozone loss. Heterogeneous chemistry due to PSCs destroyed 30 DU ozone, which was about 33 % of the total loss. The result indicates that chlorine activation on the binary aerosols has a significant role in ozone depletion. Some studies suggest that binary aerosols are more important for chlorine activation than PSCs (e.g. Drdla and Müller,2012; Wohltmann et al., 2013; Kirner et al., 2015). The increase of water vapour (Max simulation) did not increase the ozone loss, but in the Min simulation there was 6 DU less ozone depletion."

• p. 14., section 4: My suggestion would be to not combine discussions and conclusions. Have a separate discussion section to focus on the relations of the results of this study with what is available in the relevant literature and a separate conclusion section to focus on the main conclusions of this study. But this is up to the authors.
We did consider this suggestion, but decided not to change the structure.

• p 14., l 15: this is an example where the analysis of the paper could be more focused and more detailed. What means "larger"? It will be important to which altitude regions and how much towards the vortex edge the PSC area extends. If these details are analyzed more can be learned about the processes responsible for the model results.
The main finding here was that if the winter is cold, the PSCs may form even at low water vapour concentrations, therefore the water vapour increase is less important.

"If the winter is cold enough, the increase is less important, because the PSCs may form even at low water vapour concentrations, and the chlorine activation is already nearly complete in Arctic vortex, therefore the water vapour increase is less important. As expected, heterogeneous chemistry is more important in cold winters."

• Fig. 6: In general I think that there is not enough information on chlorine species in this paper. Specifically:

2012/13: why is there a range in reservoir species in early winter in this case for Min/Max? Can this be explained/understood?

The range in early winter is probably due to the effect of water vapour on the composition of binary aerosols (and STS). A higher water vapour concentration gives a more dilute aerosol and larger surface area, which increase the uptake coefficient and reaction rate. We added discussion of the early chlorine partitioning.

"The water vapour concentration seems to strongly affect the transformation of chlorine from the reservoir species to the intermediate ones in the beginning of Arctic winter. The fractions of intermediate and reservoir chlorine species change significantly with water vapour concentration during November and December. The water vapour concentration affects the composition of binary aerosols an STS. When more water condenses to the particles the uptake coefficients for the heterogeneous reactions and the surface area increase, i.e. the reaction rates increase."

• Fig. 7: Is there really an established Arctic (!) vortex (not only small remainders of the vortex) for the entire period shown here? Until mid-April! Show a time series of the size of the vortex at least in the reply or in a supplement.

We believe that in cold winters a well developed vortex can last until mid-April and, in extreme cases such as 1990 and 1997, potentially even until early May. See for example study by Waugh et al. (1999). In the revised version we consider vortex to be established when its area (defined by 36 PVU contour) exceeds that corresponding to 80° equivalent latitude, following Waugh et al. (1999). This new definition results in only minor changes with respect to our original version. The size of the vortex can be seen in Fig 1.

[Figure]

Figure 1. Vortex area (Million sq meters).

Anonymous Referee #2

General
The authors investigate the sensitivity of modelled Arctic ozone loss to the water vapour mixing ratio entering the tropical stratosphere in a chemical transport model. They guide the reader well step by step through the causal chain water vapour concentration → PSC volume → chlorine activation → ozone loss. The authors clearly state that the investigated question is different from investigating the effect of water vapour changes due to climate change (which would occur on such a timescale that also the concentration of chlorine- and bromine-containing species changes considerably). They also clearly state that they investigate only one aspect of the above-mentioned question, namely the effect of water vapour on the surface area density of Polar Stratospheric Clouds (PSCs), neglecting the (probably stronger) effect of water vapour changes on ozone loss via changes of stratospheric temperature.

Comments
• An increase of water vapour may enhance heterogeneous chemistry by enlarging the air volume in which PSCs are formed (shown in Figs. 4 and 5) or via enlarging the surface area of existing particles (not shown). The authors seem to assume that the first effect is the dominant one. A discussion of this topic would help to complete the logic of the paper.
It is true that there also the available surface area and the composition of binary aerosols and PSCs can affect the heterogeneous chemistry, in addition to just the existence of PSCs. We therefore

1) improved the description of the chemistry scheme

"The reaction rates on NAT and ICE PSCs are not directly affected by the water vapour concentration except through the available surface area, i.e. the uptake coefficients are constant. In the case of binary aerosols and STS PSCs the uptake coefficients of some reactions depend on the composition of the droplets, i.e the hydrolysis reactions of ClONO2, BrONO2 and N2O5, as well as the reaction of HCl between ClONO2 and HOCl (Sander et al., 2011)."

2) We added some discussion on the effect of water vapour on heterogeneous binary in/on aerosols and STS.

"The water vapour concentration seem to strongly affect the transformation of chlorine from the reservoir species to the intermediate ones in the beginning of Arctic winter. The fractions of intermediate and reservoir chlorine species change significantly with water vapour concentration in November and December, when STS PSCs start to form. The water vapour concentration affects the composition of binary aerosols an STS. When more water condenses to the particles the uptake coefficients for the heterogeneous reactions increase, i.e. the reaction rate increases."

• The authors claim an important role of NOx chemistry in warm winters. I would appreciate plots showing this, e.g., altitude-time plots of vortex-averaged NOx or / and altitude-time plots of the corresponding reaction rates (NO2 + O → ... for NOx chemistry, and, for comparison, Cl2O2 + hv → ... and perhaps ClO + O → ... for ClOx chemistry).
From the difference between the no-hetero and the full chemistry simulations it can be seen that the ozone loss in winter 2012/13 is gas phase chemistry driven. The text was rearranged, and the NOx-part was moved to the discussion of the ozone profile figure (Fig 8). The text now reads:

"In winters when the polar vortex is unstable and small or disturbed the Brewer–Dobson circulation brings more NOx -rich air to the polar vortex than usual. Hence the ozone loss in the 2012/13 winter was produced mostly by NO x chemistry as shown previously by e.g., Sagi et al. (2017), and

can be seen from FinROSE results by comparing the simulations with and without the heterogeneous chemistry."

• The authors do not discuss the influence of heterogeneous NOx chemistry on ozone. Can the model results be used to answer the question whether the heterogeneous reaction N2O5 + H2O → 2HNO3 reduces NOx and thus NOx -driven ozone loss in cold years (compared to other years)? We were not able to attribute any change in NOx chemistry to changes in water vapour, but we modified the text to more clearly separate the discussion of the NOx and chlorine caused ozone loss.

Minor comments
• 1/17: "2-7% more ozone loss than in colder winters" ⇒ Does this mean "2-7% stronger increase of ozone loss than in colder winters"?
Yes. We changed the text in the abstract:

"We found that the impact of water vapour changes on ozone loss in the Arctic polar vortex depends on the meteorological conditions. The strongest effect was in intermediately cold conditions, such as 2013/14, when added water vapour resulted in 2-7% more ozone loss due to the additional polar stratospheric clouds (PSC) and associated chlorine activation on their surface, leading to ozone loss. The effect was less pronounced in cold winters such as 2010/11 because cold conditions persisted long enough for a nearly complete chlorine activation even in simulations with observed water vapour. In this case addition of water vapour to the stratosphere led to increased area of ice PSC but it could not increase the chlorine activation and ozone destruction significantly. In the warm winter 2012/13 the impact of water vapour concentration on ozone loss was small, because the ozone loss was mainly NOx induced."

• 2/1 and 14/3: "warms the climate" means "warms the troposphere"?
We changed the text as suggested.

• 3/2-3: "cooling stratosphere ... slowing down some gas-phase reactions": Which gas-phase reactions are meant? In the polar stratosphere (during winter / spring) an important reaction is the three-body reaction ClO + ClO + M → Cl 2 O 2 , the rate-constant of which increases for decreasing temperature.
It is true that the effect of temperature on future ozone is complex and our statement was too general. We meant the second-order reactions mainly in NOx and HOx ozone loss cycles. This effect is mainly seen outside the polar vortex, while the effect on PSCs from temperature is seen within the vortex. We improved the text to clarify this.

"However, cooling of the stratosphere could at least partially offset the effect of the increased PSCs by slowing down the second-order reactions in ozone loss cycles (Rosenfield et al., 2002 and Revell et al., 2012). This effect is mainly seen in NOx and HOx induced loss outside the polar vortex, while the effect on PSCs from temperature is seen within the vortex."

• 5/3: "around 80 hPa": In view of the discussion in 5/29-31 this should be formulated more precisely, e.g. "at the cold point, which lies approximately x hPa below 80 hPa"
We added some details to the description of the lower boundary condition for water vapour.

"The simulations differed from each other by the prescribed water vapour concentration in the tropical tropopause region (stratosphere between 21S–21N, below 80 hPa), where it was prescribed as follows: (1) water vapour taken from ERA-Interim (Interim simulation), (2) increased water vapour (Max simulation), and (3) decreased water vapour (Min simulation). Specifically, the SWV lower boundary conditions for Min and Max simulations were obtained by multiplying values from

ERA-Interim between tropopause and 80 hPa, and between 21 S–21 N by monthly coefficients ranging between 1.46-1.7 (Max) and between 0.5-0.63 (min), so that they approximately correspond to the driest and wettest CCM, as determined by SWV values at the tropical tropopause, across models analyzed by Gettleman et al. (2010)."

• 5/29-31: "leads ... by 3-4 weeks ... Brewer-Dobson circulation ... too fast": Does this mean that between the cold point and 80 hPa the ERA-Interim circulation takes 3-4 weeks less than the real circulation. Is the distance between the cold point and 80 hPa large enough to gain such a difference?
Thank you for this comment. We believe that a too fast BD transport in ERA-I likely contributes to the difference but we are not sure if it can indeed explain such a large delay. Therefore we rewrite the text as follows:

"However, Interim variability leads that of MLS by 3–4 weeks. The reason for the time lag between Interim and MLS is not clear although it could at least partly be associated with a too fast Brewer–Dobson circulation in ERA-Interim which is responsible for upward transport of the water vapour anomalies in the tropics (Schoeberl et al., 2012)."

• 6/3: "gains a small amount of water": How, by horizontal mixing?
In the revised version this claim is removed and the sentence is rewritten. In the original version we unfortunately overlooked the fact that in Min simulation the scaling factor was varying with a seasonal cycle between 0.5-0.63, rather than fixed at value 0.5 as was stated in the original text. This point is clarified in the revised version. Therefore the difference between Min and Interim of factor 0.6 in Fig. 1 is consistent with prescribed values and there is no evidence of gained water vapour in Min simulation. The new text reads:

"The Max simulation has 2–3 ppm more water vapour in the tropics than the Interim simulation, while the Min simulation is about 1.5 ppm drier than the Interim simulation. These differences correspond to the ratio between Max/Interim of approximately 1.55-1.6 and about 0.55–0.6 between Min/Interim, i.e. they are consistent with the prescribed boundary conditions."

• 8/4: "For example too dry models may not be able to simulate a large Arctic ozone loss such as of 2010/11": How does this sentence relate to the preceding sentence?
The sentence "For example too dry models may not be able to simulate a large Arctic ozone loss such as of 2010/11." was removed from the results section and the discussion was moved to the Discussion and conclusions section.

• 9/15: "in spring": and also during southward excursions of air masses during winter
Yes, it is more related to availability of sunlight than the time of the year, the text was changed accordingly:

"When sunlight reaches the polar vortex these species are easily dissociated to form active chlorine species that participate in the catalytic ozone depletion cycles, i.e. ClO x (ClO, Cl 2 O 2 and Cl)."

• 9/16: "PSCs sustain the regeneration of ClO x ": This is only possible if both reaction partners for a heterogeneous reaction are still present.
The reservoir species are continuously formed through the ClO+NO2 and Cl+CH4 reactions, but as long as there are PSCs the ClOx is regenerated. We added some text to clarify this part.

"Active chlorine goes back to reservoir species through reactions with NO2 and CH4, however if PSCs are present the regeneration of ClOx is sustained."

• 10/9: "rather short": The green curve in Fig. 6d lies above 60% for about 2 months. Is this meant by "rather short"?

We meant in comparison to the winter 2010/11, the text was corrected:

"The 2015/16 winter started similar to the cold winter 2010/11 and nearly all of the chlorine was activated at the beginning of January, but the deactivation started already in the end of January making the period with high ClOx shorter than in winter 2010/11."

• 10/17: "differs significantly ... during the period with high ClO x ": The difference is mostly less than 10%. Is this meant by "differs significantly"?

The formulation was quite vague, we altered this and the next sentence as follows:

"The difference in concentration of active chlorine and reservoir species between the Min and Max simulations are smallest during the cold periods, due to the effective processing on the PSCs (Fig 6.). The cold winter 2015/16 shows a very small range, and the intermediatly cold winter 2013/14 a wider range in concentrations. The water vapour content has less effect on the chlorine partitioning in cold winters."

• 10/18: What exactly is meant by "chlorine activation period": the time when most chlorine exists in active form?

The chlorine activation period was not defined exactly, it was based on a subjective interpretation of the figures. It is difficult to find a robust definition and we therefore removed the phrase. See also previous comment.

"The cold winter 2015/16 shows a very small range, and the intermediately cold winter 2013/14 a wider range in concentrations. The water vapour content has less effect on the chlorine partitioning in cold winters."

• 11/4: "... only at the 475 K level": Does this mean that the 475 K level is used for the definition of the whole vortex? If so, this should be mentioned (and perhaps be discussed) already in 6/7.

We used the PV on the 475 K level to define the polar vortex only in Fig 7. The text was modified as follows:

"In Fig. 7 the polar vortex is defined using the potential vorticity limit 36 PVU only at the 475 K level."

• 11/11: "FinROSE seems to underestimate the ozone loss, possibly due to a general 10% negative bias in total ozone": Why does an underestimation of the ozone concentration lead to a significant underestimation of the ozone loss? The rate of ozone loss in polar winter/spring is largely determined by the rates of reactions like ClO + ClO + M → Cl 2 O 2 + M and BrO + ClO → ... .

In extreme cases a negative bias could limit the ozone loss, however for Arctic conditions this is generally not the case. We don't have a clear understanding of the reason for the underestimation, and this statement is not valid here. We changed the text accordingly.

"FinROSE seems to underestimate the ozone loss; for example Sinnhuber et al. (2011) and Manney et al. (2011) simulated 120 DU ozone loss and Pommereau et al. (2013) even 170 DU in winter 2010/11."

• 11/23: "stopping the catalytic ozone cycles and ozone loss early": The ozone loss stops around the beginning of March (Fig. 7d). Is this meant by "early"?

We meant earlier than 2011. The text has changed:

"A relatively small ozone loss of 56 DU was simulated in 2015/16, which was due to the unstable polar vortex, which split and warmed, stopping the catalytic ozone cycles and ozone loss in the beginning of March, i.e. earlier than in 2010/11 and 2013/14."

• 13/4: "the heterogeneous chemistry destroyed about 36 DU of ozone": In fact, ozone is destroyed by gas-phase reactions. Heterogeneous chemistry produces (some of) the corresponding reactants. Yes, the formulation was missleading, the text now reads:

"In 2013/14 the heterogeneous chemistry caused about 40 DU ozone destruction, which is about 51 % of the ozone loss and ICE and NAT/STS about 23 DU (30 %), when the total ozone loss was 79 DU in the Interim run.

• 14/24: "in higher level": Does this mean at higher altitudes?
No, we meant more complete. The text has changed:
"Cold winters differ from the warm winters regarding the ozone loss and the fraction of ozone loss initiated by heterogeneous chemistry, during cold winters the PSC volumes are larger and thus chlorine activation within the polar vortex is more complete."

• Tables 1, 2, 3: As a "service" for the reader the character of the years (warm, intermediate, cold) might be added (as was done in the figures). Perhaps the winter 2015/16 might be called "initially cold", in order to distinguish it from 2010/11.
The character of the years was added to the tables as well.

**Technical details**
• Please check the "s" of the plural of substantives or singular of verbs
1/13, 2/3, 10/12, 10/16-17, 14/4, 15/29
We made the suggested corrections to the text.

• 1/22: "processes. Especially" ⇒ "processes, especially"
OK
• 2/7: "(2002)" ⇒ ", 2002"
OK
• 4/9: "lagrangian" ⇒ "Lagrangian"
OK
• 6/27: The abbreviation "BD" has not been defined before (and is used only once).
OK
• 6/33: "(Fig. 3)" ⇒ Fig. 3
OK
• 7/18: Really "0.3 ppt" (or 0.3 ppb)?
The limit we used was 0.3 ppt, it seems low but at temperatures above 200K only a small fraction of the nitric acid is dissolved in the aerosols. We had a closer look at the limit and we have now increased it to 0.6 ppt, which 
[revised manuscript text omitted]

---

## Editor Decision (ED1)

**Editor comments on Manuscript. No: ACP-2018-310**

Thölix et al., Linking uncertainty on simulated Arctic ozone loss to uncertainties in modelled tropical stratospheric water vapour

P1, L2: Since you are referring here to simulations I would suggest to write either at the begin of the sentence "The simulated amount of water vapour" or at the end of the sentence "between simulations from chemistry-climate models.

P1, L6: Similar here. Since you refer to model simulations "amount" should be replaced by "simulated amount".

P1, L14: add "stratospheric winter" so that it reads "in cold stratospheric winters"

P1, L14: add "winter" after 2013/14.

P1, L15: add "formation of", so that it reads "due to the additional formation of polar stratospheric clouds".

P1, L16: be more precise, thus change "such as 2010/11" to "such as the 2010/11 winter".

P1, L17: Be more precise, thus change with observed water vapour to without changing the prescribed water vapour amount

P1, L18: it should read either "an area" or "areas".

P1, L18: could not → did not.

P1, L22: Rephrase "of ozone layer recovery" either to "ozone recovery" or "of the recovery of the ozone layer".

P2, L5-6: This sentence is not clear. Do you mean that there will be an additional warming of the troposphere due to higher water vapour? Does this not rather come from a future cooling of the stratosphere? Please clarify and rephrase the sentence.

P2, L8: Add comma after "However".

P2, L12: The references of Solomon et al., 1999 and Khosrawi et al. (2016) are not adequate in this context. I would suggest to cite here the paper by Hamill et al. (1997) or Peter (1997). If you prefer a newer reference you alternatively could cite the book chapter of Peter and Grooss (2012).

P2, L15: add "that are rarely reached in the Arctic" after (below about 195 K) and add "there" after place and remove "in the Arctic" at the end of the sentence, so that it reads: "Since the formation of PSCs requires very low temperatures (below about 195 K) that are rarely reached in the Arctic, significant polar ozone depletion takes place there only occasionally."

P2, L24-25: Change sentence as follows: "Kuttipurath and Nair (1997) showed based on ozone balloon soundings and total ozone data from satellite instruments that ozone has begun to recover.

P2, L25: remove "profile", so that it just reads "data".

P2, L26: showed → could show

P3, L14: "of the increase due to the increases"? Is here one of the "increases" obsolete?

P3, L20: Also Revell et al. (2016) → Revell et al. (2016) also

P3, L26: smallest → smallest was

P3, L28: skip "atmospheric".

P3, L33: early → in early

P4, L8-9: Rephrase sentence as follows: "One may wonder what the implications of these discrepancies for stratospheric ozoone losses simulated by CCMs are."

P4, L9: Do you refer here to simulated transport or the real transport? Please be more precise and rephrase sentence accordingly.

P4, L18: "as the" appears twice, thus one is obsolete.

P4, L32: remove "chemistry" and add "are used" or "applied" after Atkinson et al. (2007b).

P5, L3: What exactly is meant with composition? The amount of H2SO4, H2O or HNO3 in the droplets or do you mean the kind of PSC/strat aerosol particles as binary, STS etc?

P5, L24: Reference of Thölix et al. should be given here in parantheses.

P6, L2: The reference of Gettelmann et al. 2010 should be given without parantheses.

P6, L5: considered spinup → considered as spin-up.

P6, L6: add "amount" or "concentration" after "water vapour".

P6, L19: minimum of what? Be more precise.

P6, L21: occur → occurs

P6, L26: What do you mean with lead? Is ahead, thus happening earlier? IN that case I would replace "leads" by "is ahead".

P7, L5: correspond → corresponds

P7, L7: sufficient for → sufficient for the

P7, L7: though → throughout

P7, L13: inside → inside the

P7; L13: close to → close to the

P7, L20: due to → due to an

P7, L23: Change sentence as follows: Typically, there is a stronger increase in water vapour towards spring in the MLS observations compared to the FinRose simulations.

P7, L32: as seen → as can be seen

P8; L7: corresponded to formation → corresponds to the formation

P8; L16: Corresponding to how much HNO3 in weight percent would that be?

P9, L2: PSC → ICE PSC ? I guess you mean here specifically ice PSCs or am I wrong?

P9, L3: Add "being present" so that it reads: "Also the duration of the ICE clouds being present is comparable".

P9, L4: add "the EMAC" so that it reads "with the EMAC model"

P9, L4: are → were

P9, L4. Change "than the observed ones" to "the ones observed with MIPAS."

Note: In Khosrawi et al., the comparison was performed for the total PSC volume. Thus, this does not mean that ice is not correctly simulated with the EMAC model. The STS PSCs have the largest volume and our conclusion was that STS is significantly underestimated, but the other PSCs may be correct or (in case of NAT) be rather overestimated.

P9, L7: as very cold → as being very cold

P9, L8: Through → throughout

P9, L12: had → has

P9, L13: in Max → in the Max

P9, L17: PSC starts → PSCs start

P9, L28: in Interim simulation → in the Interim simulations

P9, L28: add "the" so that it reads "between the Max and……."

P9, L29: add also here simulation, so that it reads Interim simulations

P9, L31: add "the" before 2012/13 and change were to was

P9, L32: as the decrease → decrease of what. Please clarify and rephrase accordingly.

P9, L25: add "the", so that it reads "the Min and Interim simulations"

P10, L2: persist → persisted

P10, L3: add "winter" after 2015/16

P10, L3: area → areas

P10, L6: remove "altitude", so that it reads "at 55 hPa".

P10, L18: comma obsolete

P10, L20: goes back → is transformed back

P10, L29: reservoir → reservoir species

P10, L33: to → to the

P11, Table 2 caption: change → changes

P11, L1: remove "it"

P11, L3: add "the" before "chlorine activation"

P11, L4: warmed → temperature increased

P12, L3: "warmed up" should be rephrased

P12, L6: add "the" so that it reads "the Arctic winter".

P12, L8: an STS → and STS

P12, L12: what exactly to you mean with effective processing on PSCs? Please clarify and rephrase accordingly.

P12, L21: only few percent → only a few percent

P12, L27: replace "has" by " provides" or "contains"

P12, L29: remove "the" before smallest.

P12, L34: increase of what? ClOx?

P13, L1: change of what? Of ClOx?

P13, L9: add distribution so that it reads "ozone distribution". It would also be worth to add "on that day" to be more precise.

P13, L16-17: delete "value" at the end of the sentence and add "value given by" before Sinnhuber et al. (2011).

P13, L20: "mid April" or "by the mid of April".

P13, L29: replace "warmed early" by " was ended early by a SSW".

P13, L32: was about same → was about the same

P13, L32: add "the" before 2010/11

P13, L34:  change sentence to "with a water vapour increase of about the same magnitude as considered here".

P14, L5: is → are and change sentence as follows: The changes in the amount of water vapour are in the range that was testes here and are not very important for ozone loss in cold years."

P14, L7: change sentence as follows: "and thus did not increase the ozone depletion."

P14, L8: Strengthen → strengthens

P14, L8: at least → at least by

P14, L8: at → for

P14, L8: for 2011 → for the 2011 winter

P14, L8: is → was

P14, Table 3 caption: seperately → separately by

P14, L15: rephrase to "….ozone depletion was reduced by 6 DU."

P14, L18: separately → separately by

P14, L19: in Interim → in the Interim

P15, L2: rephrase "depletion of 56DU."

P15, L14: So → Thus

P15, L15: …..role than in colder years? What do you mean exactly? Have larger role in warmer years than in colder years? Please rephrase.

P15, L17: "part"? Do you mean contribution?

P15, L21: through → due to

P15, L25: is → was

P15, L25: "part"? Do you mean contribution?

P15, L27: eater → water

P15, L31: is → was

P15, L31: occur → occurred

P15, L31: to → into

P16, L6: clear → pronounced

P16, L10 and 11: is → was

P16, L13: the same as we found → the same as what we found

P16, L15: What exactly do you mean here with warmer climate? An increase in temperatures? This sentence is not clear and thus should be rephrased.

P16, L17: change to "enhance each other, so that  the area of PSCs increases and that these can last longer in the vortex."

P16, L23: Results →  "The results" or "Our results"

P16, L23: in both occasions "a" should be added, so that it reads "a wetter/drier"

P16, L24: along → when

P17, L1-35: Here in case of a specific winter it should be written "the Arctic winter". When you refer to the stratosphere in general it should be explicitly written "polar stratosphere".

P17, L23: arrived to → came to a

P17, L1: Change sentence as follows: "Also in the Arctic winter 2010/11 chlorine activation in the vortex was nearly……..

P17, L1: what do you mean exactly with "complete"? This should be rephrased.

P17, L4: change to "the impact of water vapour on ozone loss".

P27, Figure 4 caption: ares → areas

Figures: Why are there different pressure levels used which are almost the same (55 and 54 hPa)?

Text: Why are mixing ratios given in ppm? Shouldn't it be ppmv?

**References:**
Hamill, P., E.J. Jensen, P. B. Russell and J. J. Bauman, The life cycle of stratospheric aerosols, Bulletin of the American Meteorological Society, 78, 1395-1410, 1997.

Peter, T., Microphysics and heterogeneous chemistry of  polar stratospheric clouds, Annu. Rev. Phys. Chem., 48, 785–822, 1997.

Peter, T. and Grooss, J.-U., Chapter 4: Polar stratospheric clouds and sulfate aerosol particles: Microphysics, denitrification and heterogeneous chemistry, in: Stratospheric Ozone Depletion and Climate Change, edited by: Müller, R., RSC Publishing, Cambridge, 108–144, 2012.

---

## Author Response (AR2)

Editor comments on Manuscript. No: ACP-2018-310

Thölix et al., Linking uncertainty on simulated Arctic ozone loss to uncertainties in modelled tropical stratospheric water vapour

P1, L2: Since you are referring here to simulations I would suggest to write either at the begin of the sentence "The simulated amount of water vapour" or at the end of the sentence "between simulations from chemistry-climate models.
OK  "between simulations from chemistry-climate models."

P1, L6: Similar here. Since you refer to model simulations "amount" should be replaced by "simulated amount". OK

P1, L14: add "stratospheric winter" so that it reads "in cold stratospheric winters"
We didn't find the right place, but we changed the text as follows:
"The strongest effect was in intermediately cold stratospheric winters, such as 2013/14 winter, when added water vapour resulted in 2-7 % more ozone loss due to the additional formation of polar stratospheric clouds (PSC) and associated chlorine activation on their surface, leading to ozone loss.

P1, L14: add "winter" after 2013/14. OK

P1, L15: add "formation of", so that it reads "due to the additional formation of polar stratospheric clouds". OK

P1, L16: be more precise, thus change "such as 2010/11" to "such as the 2010/11 winter". OK

P1, L17: Be more precise, thus change with observed water vapour to without changing the prescribed water vapour amount
Corrected as follows:
"The effect was less pronounced in cold winters such as the 2010/11 winter  because cold conditions persisted long enough for a nearly complete chlorine activation even in simulations with prescribed stratospheric water vapour amount corresponding to the observed values."

P1, L18: it should read either "an area" or "areas". OK

P1, L18: could not → did not. OK

P1, L22: Rephrase "of ozone layer recovery" either to "ozone recovery" or "of the recovery of the ozone layer". OK

P2, L5-6: This sentence is not clear. Do you mean that there will be an additional warming of the troposphere due to higher water vapour? Does this not rather come from a future cooling of the stratosphere? Please clarify and rephrase the sentence.
Yes, this is what we mean here and this statement is based on the results by Desler et al. We rephrased the sentence as follows to make the point clearer:
"Dessler et al., (2013) suggested that, as a result of global warming, tropopause temperature will increase leading to an increase in stratospheric water vapour (SWV). Since water vapour is a greenhouse gas, future increases in SWV will provide a positive feedback further warming the troposphere below."

P2, L8: Add comma after "However". OK

P2, L12: The references of Solomon et al., 1999 and Khosrawi et al. (2016) are not adequate in this context. I would suggest to cite here the paper by Hamill et al. (1997) or Peter (1997). If you prefer a newer reference you alternatively could cite the book chapter of Peter and Grooss (2012).
We changed the reference to Hamill et al. (1997)

P2, L15: add "that are rarely reached in the Arctic" after (below about 195 K) and add "there" after place and remove "in the Arctic" at the end of the sentence, so that it reads: "Since the formation of PSCs requires very low temperatures (below about 195 K) that are rarely reached in the Arctic, significant polar ozone depletion takes place there only occasionally." OK

P2, L24-25: Change sentence as follows: "Kuttipurath and Nair (1997) showed based on ozone balloon soundings and total ozone data from satellite instruments that ozone has begun to recover.
Corrected as follows:
"Kuttipurath and Nair (1997) showed based on ozone balloon soundings and total ozone data from satellite instruments that Antarctic ozone has begun to recover."

P2, L25: remove "profile", so that it just reads "data". OK

P2, L26: showed → could show OK

P3, L14: "of the increase due to the increases"? Is here one of the "increases" obsolete? OK

P3, L20: Also Revell et al. (2016) → Revell et al. (2016) also OK

P3, L26: smallest → smallest was OK

P3, L28: skip "atmospheric". OK

P3, L33: early → in early OK

P4, L8-9: Rephrase sentence as follows: "One may wonder what the implications of these discrepancies for stratospheric ozone losses simulated by CCMs are." OK

P4, L9: Do you refer here to simulated transport or the real transport? Please be more precise and rephrase sentence accordingly.
Yes, we refer to simulated transport. This is specified it in the revised text:
"This question is difficult to address by analyzing CCM outputs because there are other differences between the models which affect simulated ozone losses, such as differences in simulated transport."

P4, L18: "as the" appears twice, thus one is obsolete. OK

P4, L32: remove "chemistry" and add "are used" or "applied" after Atkinson et al. (2007b). OK

P5, L3: What exactly is meant with composition? The amount of $H_2SO_4$, $H_2O$ or $HNO_3$ in the droplets or do you mean the kind of PSC/strat aerosol particles as binary, STS etc?
We meant the composition of the liquid phase, i.e. the concentration of $H_2SO_4$, $H_2O$ or $HNO_3$.
"Water vapour and $HNO_3$ condenses onto the binary aerosol and STS droplets making them more dilute, which increases the uptake coefficients of some reactions, i.e the hydrolysis reactions of $ClONO_2$, $BrONO_2$ and $N_2O_5$, as well as the reaction of HCl between $ClONO_2$ and $HOCl$ (Sander et al., 2011)."

P5, L24: Reference of Thölix et al. should be given here in parantheses. OK

P6, L2: The reference of Gettelmann et al. 2010 should be given without parantheses. OK

P6, L5: considered spinup → considered as spin-up. OK

P6, L6: add "amount" or "concentration" after "water vapour". OK

P6, L19: minimum of what? Be more precise.
We mean minimum temperature. This is specified in the revised text:
"The temperature shows the typical annual cycle with minimum temperature in northern hemisphere (NH) winter and maximum temperature in NH summer."

P6, L21: occur → occurs OK

P6, L26: What do you mean with lead? Is ahead, thus happening earlier? IN that case I would replace "leads" by "is ahead". OK

P7, L5: correspond → corresponds OK

P7, L7: sufficient for → sufficient for the OK

P7, L7: though → throughout OK

P7, L13: inside → inside the OK

P7; L13: close to → close to the OK

P7, L20: due to → due to an OK

P7, L23: Change sentence as follows: Typically, there is a stronger increase in water vapour towards spring in the MLS observations compared to the FinRose simulations. OK

P7, L32: as seen → as can be seen OK

P8; L7: corresponded to formation → corresponds to the formation OK

P8; L16: Corresponding to how much HNO3 in weight percent would that be?
It is not possible to give a single corresponding weight percent limit, the exact weight percent depends on the composition of the aerosol, i.e. the fractions of sulphuric acid and water. The limit of 0.6 pptv liquid HNO3 means that 0.6 pptv has dissolved from the gas phase to the liquid phase in a given gridpoint. The text was altered to give a more detailed description of the assumption.
"There is no formation temperature for STS, but they are formed gradually as water and HNO3 dissolves into binary aerosols with decreasing temperature. Here we have defined STS based on how much HNO3 has dissolved into the liquid phase, we set the limit at 0.6 pptv liquid HNO3 for any given gridpoint. The limit approximately corresponds to a formation threshold temperature of 202 K for STS."

P9, L2: PSC → ICE PSC ? I guess you mean here specifically ice PSCs or am I wrong?
We didn't find the place from line 2, but we changed the text as follows:
"The observed ICE PSC areas are comparable to the FinROSE modelled ICE PSC areas (Thölix et al., 2016)"

P9, L3: Add "being present" so that it reads: "Also the duration of the ICE clouds being present is comparable". OK

P9, L4: add "the EMAC" so that it reads "with the EMAC model" OK

P9, L4: are → were OK

P9, L4. Change "than the observed ones" to "the ones observed with MIPAS." OK

Note: In Khosrawi et al., the comparison was performed for the total PSC volume. Thus, this does not mean that ice is not correctly simulated with the EMAC model. The STS PSCs have the largest volume and our conclusion was that STS is significantly underestimated, but the other PSCs may be correct or (in case of NAT) be rather overestimated.

P9, L7: as very cold → as being very cold OK

P9, L8: Through → throughout OK

P9, L12: had → has OK

P9, L13: in Max → in the Max OK

P9, L17: PSC starts → PSCs start OK

P9, L28: in Interim simulation → in the Interim simulations OK

P9, L28: add "the" so that it reads "between the Max and......." OK

P9, L29: add also here simulation, so that it reads Interim simulations OK

P9, L31: add "the" before 2012/13 and change were to was OK

P9, L32: as the decrease → decrease of what. Please clarify and rephrase accordingly.
The text now reads:
"The increase of water vapour in the Max simulation did not change the PSC area as much as the decrease of water vapour in the Min simulation. "

P9, L35: add "the", so that it reads "the Min and Interim simulations" OK

P10, L2: persist → persisted OK

P10, L3: add "winter" after 2015/16 OK

P10, L3: area → areas OK

P10, L6: remove "altitude", so that it reads "at 55 hPa".OK

P10, L18: comma obsolete OK

P10, L20: goes back → is transformed back OK

P10, L29: reservoir → reservoir species OK

P10, L33: to → to the OK

P11, Table 2 caption: change → changes OK

P11, L1: remove "it" OK

P11, L3: add "the" before "chlorine activation" OK

P11, L4: warmed → temperature increased OK

P12, L3: "warmed up" should be rephrased
The text now reads:
"In the end of February the vortex temperature increased and chlorine transformed back to reservoir species."

P12, L6: add "the" so that it reads "the Arctic winter". OK

P12, L8: an STS → and STS OK

P12, L12: what exactly to you mean with effective processing on PSCs? Please clarify and rephrase accordingly.
The text now reads:
"The difference in concentration of active chlorine and reservoir species between the Min and Max simulations are smallest during the cold periods, due to heterogeneous chemistry on the PSCs (Fig. 6)."

P12, L21: only few percent → only a few percent OK

P12, L27: replace "has" by " provides" or "contains" OK

P12, L29: remove "the" before smallest. OK

P12, L34: increase of what? ClOx? OK

P13, L1: change of what? Of ClOx? OK

P13, L9: add distribution so that it reads "ozone distribution". It would also be worth to add "on that day" to be more precise. OK

P13, L16-17: delete "value" at the end of the sentence and add "value given by" before Sinnhuber et al. (2011). OK

P13, L20: "mid April" or "by the mid of April". OK

P13, L29: replace "warmed early" by " was ended early by a SSW". OK

P13, L32: was about same → was about the same OK
P13, L32: add "the" before 2010/11 OK

P13, L34: change sentence to "with a water vapour increase of about the same magnitude as considered here". OK

P14, L5: is → are and change sentence as follows: The changes in the amount of water vapour are in the range that was tested here and are not very important for ozone loss in cold years."  OK

P14, L7: change sentence as follows: "and thus did not increase the ozone depletion." OK

P14, L8: Strengthen → strengthens OK
P14, L8: at least → at least by OK

P14, L8: at → for OK

P14, L8: for 2011 → for the 2011 winter OK

P14, L8: is → was OK

P14, Table 3 caption: seperately → separately by OK

P14, L15: rephrase to "....ozone depletion was reduced by 6 DU." OK

P14, L18: separately → separately by OK

P14, L19: in Interim → in the Interim  OK

P15, L2: rephrase "depletion of 56DU." OK

P15, L14: So → Thus OK

P15, L15: .....role than in colder years? What do you mean exactly? Have larger role in warmer years than in colder years? Please rephrase.
The text now reads:
"Thus, water vapour changes have a larger effect on ozone loss in moderately cold years than in cold ones."

P15, L17: "part"? Do you mean contribution? OK

P15, L21: through → due to OK

P15, L25: is → was OK

P15, L25: "part"? Do you mean contribution? OK

P15, L27: eater → water OK

P15, L31: is → was OK

P15, L31: occur → occurred OK

P15, L35: to → into OK

P16, L6: clear → pronounced OK

P16, L10 and 11: is → was OK

P16, L13: the same as we found → the same as what we found OK

P16, L15: What exactly do you mean here with warmer climate? An increase in temperatures? This sentence is not clear and thus should be rephrased.
This statement has already been made in the introduction. Therefore we decided not to repeat it here and just remove.

P16, L17: change to "enhance each other, so that the area of PSCs increases and that these can last longer in the vortex." OK

P16, L23: Results → "The results" or "Our results" OK

P16, L23: in both occasions "a" should be added, so that it reads "a wetter/drier" OK

P16, L24: along → when OK

P17, L1-35: Here in case of a specific winter it should be written "the Arctic winter". When you refer to the stratosphere in general it should be explicitly written "polar stratosphere".
Corrected

P17, L23: arrived to → came to a OK

P18, L1: Change sentence as follows: "Also in the Arctic winter 2010/11 chlorine activation in the vortex was nearly........ OK

P18, L1: what do you mean exactly with "complete"? This should be rephrased.
The text now reads:
"Also in the Arctic winter 2010/11 nearly all chlorine in the polar vortex was activated, and additional water vapour did not lead to additional activation, nor to additional ozone depletion."

P18, L4: change to "the impact of water vapour on ozone loss". OK

P27, Figure 4 caption: ares → areas OK

Figures: Why are there different pressure levels used which are almost the same (55 and 54 hPa)?
It should be 55 everywhere, and is now corrected

Text: Why are mixing ratios given in ppm? Shouldn't it be ppmv?
Yes, it should be ppmv, and is now corrected to the text and figures.

[revised manuscript text omitted]